

# Butterflies (Lepidoptera: Papilionoidea) of Georgia (Caucasus): annotated review of regional butterfly fauna with vernacular names index, notes on distribution and phenology

Iwona Słowińska[1] and Krzysztof Jonko[2]

[1] Department of Invertebrate Zoology and Hydrobiology, Faculty of Biology and Environmental Protection, University of Lodz, Łódź, Poland
[2] Łódź, Poland

## ABSTRACT

**Background:** It is well known that butterflies are valuable indicators of environmental quality, given their existence in various habitats. Collecting and regularly updating data on species richness, abundance, and distribution of all butterfly species in the country is crucial for effective monitoring and conservation efforts, which can ultimately help minimise biodiversity losses. Since the last publication of the Georgian butterfly list, there have been numerous reports registering taxonomic revisions, nomenclatural changes or providing several new butterfly "cryptic species" based on genetic research. In the following article, based on a review of various sources of data, including existing literature and new, unpublished data, we present an annotated regional checklist of butterflies of Georgia, a country that is a part of the Caucasus ecoregion representing one of 36 biodiversity "hotspots".

**Methods:** A database with all reported species for Georgia was created by compiling information from critical reviews of all available literature reports, records submitted by contributors of three websites dedicated to butterfly fauna, and data deposited in the Global Biodiversity Information Facility (GBIF) database. Various specialised sources were used to extract vernacular species names (Georgian, Russian and English).

**Results:** The updated list of butterflies from Georgia includes 244 species of the superfamily Papilionoidea recorded from almost 600 different locations. Nearly 25% of the 244 species were considered rare and extremely rare, while at the same time being at risk of potential extinction. For each species, we present brief phenological information, distribution in Georgia/Caucasus, occurrence status in each region of the country, thumbnails (ventral and dorsal view), as well as a list of vernacular names in Georgian, Russian and English. Regarding species that are reported in the literature as new, uncertain, or questionable in Georgia, we provide the relevant comments. In comparison to the other republics of Transcaucasia (Armenia, Azerbaijan), we noted a similar number of species. Our studies provide a robust baseline of data for further exploration of the Lepidoptera fauna of Georgia. This

Corresponding author
Iwona Słowińska,
iwona.slowinska@biol.uni.lodz.pl

foundation should help to fill in the gaps in knowledge regarding regional species distribution, phenology, and habitat requirements.

## INTRODUCTION

Insect abundance and diversity have declined dramatically during the last decades (*Hallmann et al., 2017*, *2021*; *Leather, 2017*; *Forister, Pelton & Black, 2019*; *Rhodes, 2019*; *Sánchez-Bayo & Wyckhuys, 2019*; *Dalton et al., 2023*). The conservation of biodiversity has emerged as a critical global concern. Data on species distribution and fauna diversity may serve as points of reference for estimating environmental changes and, at the same time, constitute the basis for assessing the conservation status of species and implementing conservation strategies to prevent further loss of insects, including Lepidoptera. Regarding Georgian fauna, *Didmanidze (2004)* is the only work known to the authors with a clear focus to document the lepidopteran fauna of Georgia. It was the first checklist of Georgian butterflies containing all species found in this country known at that time. Unfortunately, many species listed by Didmanidze have been considered doubtful or misidentified by, among others, *Nekrutenko, Korshunov & Effendi (1982a*, *1982b)* and *Tshikolovets & Nekrutenko (2012)* and require independent confirmation. Generally, data on Georgian butterflies are still quite scattered and mostly deal with some separate regions and taxa only.

The Republic of Georgia is situated between two major mountain ranges: the Greater Caucasus and the Lesser Caucasus. Along with Azerbaijan and Armenia, Georgia forms one Transcaucasian region between the Caucasus Mountains and the borders of Iran and Turkey. It is divided into two autonomous republics (Abkhazia, Adjara), one city district (Tbilisi) and nine regions (Fig. 1).

Georgia is located in the Caucasus ecoregion—one of 36 biodiversity hot-spots—which shelters one of the most diverse fauna of the temperate region (*Zazanashvili et al., 2004*). This ecoregion is distinguished for having high levels of diversity and endemism while, at the same time, being seriously threatened by loss of habitats. Although Georgia covers a small part of the Caucasus, it contains the largest part of Caucasian forests, which cover about 40% of its territory. As many as 98% of Georgian forests are growing in the mountains, and only 2% is situated in low-lying areas (*Patarkalashvili, 2017*). The major threats to these habitats include overcutting, overgrazing, forest pests and diseases, fires, invasive alien species, and poor land-use practices (*Akhalkatsi, 2015*; *Zazanashvili et al., 2020*; *Beridze & Dering, 2021*). These factors, along with increasing climate changes intensify the overall loss of habitats and, in consequence, the biodiversity in Georgia. For biodiversity management, a national checklist of insect species is essential. Such initiatives as regularly updatable checklists or databases have already been compiled for many countries. These projects may be an important source of necessary information for a broad

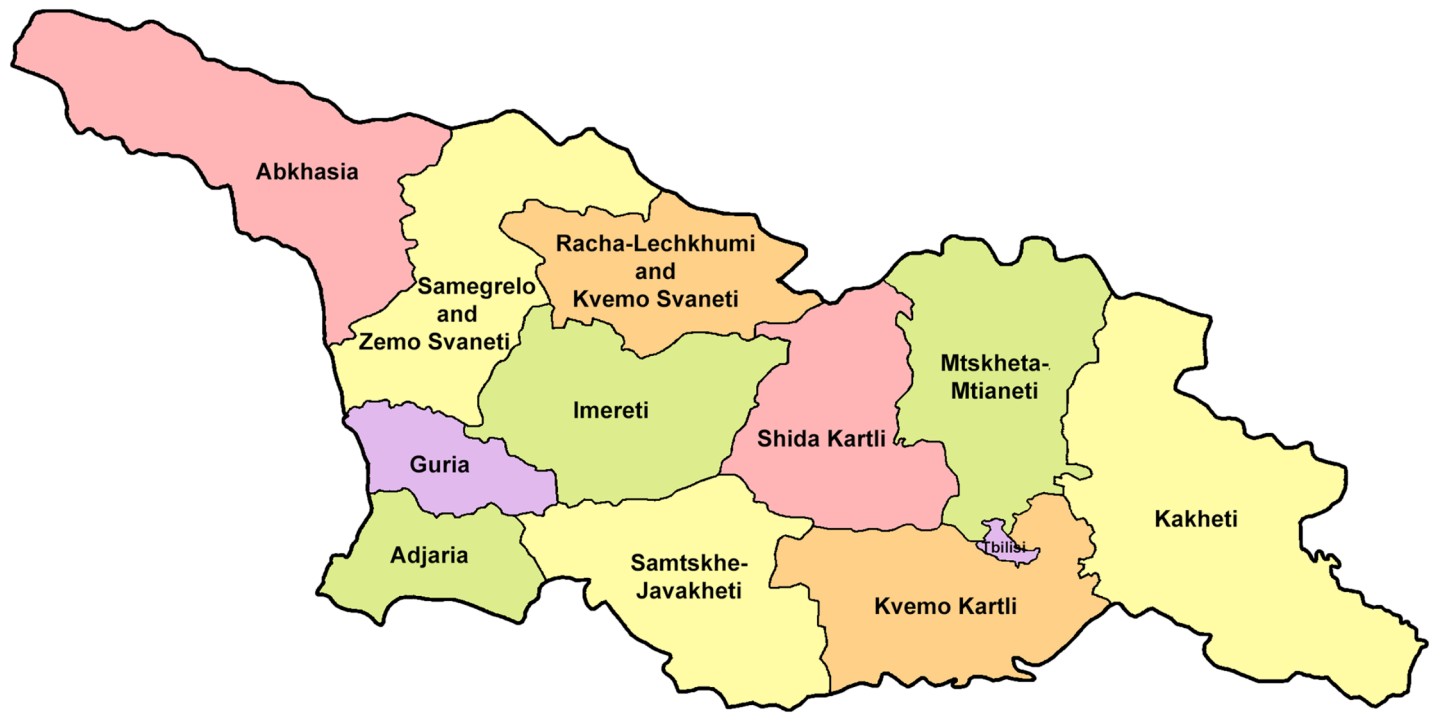

**Figure 1 Map of Georgia with administrative divisions.** Map credit: Krzysztof Jonko.

range of potential users, including conservationists, citizen scientists and scientists that would like to understand patterns of insect species diversity and distribution at a national or regional level.

The history of butterfly research in the region known as present-day Georgia starts in 1832 with a article by *Ménétriés (1832)* who provided the first species list of the Caucasus butterflies, though without precise locality data. For this reason, that publication could not be used as a data source for the regional checklist of butterflies of Georgia. Some time later, *Nordmann (1851)* published the first list of Georgian Lepidoptera species with full locality data. More intensive studies of butterflies from the country began in the second half of the 19th century (*Lederer, 1864, 1870*; *Hedemann, 1876*; *Christoph, 1877, 1881, 1886, 1889*; *Romanoff, 1884, 1885, 1887*; *Shavrov, 1886*; *Radde, 1899*) and continued quite actively in the first half of the 20th century. The period before the end of the Second World War was characterised by many reports of Georgian butterflies (*e.g.*, *Jachontov, 1911*; *Sheljuzhko, 1924*; *Miljanovskiy, 1941, 1947*), but none have been so extensive and comprehensive as publications by *Romanoff (1884, 1885, 1887)*. After the Second World War, many different authors, mainly from the former Soviet Union, published data on butterflies from the territory of Georgia (*e.g.*, *Rjabov, 1958*; *Miljanovskiy, 1964*; *Batiashvili & Didmanidze, 1973*; *Didmanidze, 1975, 1979, 1980*; *Vashakidze, 1976*; *Nekrutenko, 1990*). Studies on Georgian butterflies diminished in the late 20th century and have only recently resumed (*e.g.*, *Didmanidze et al., 2002*; *Tshikolovets, 2003, 2011*; *Didmanidze, 2004, 2005a, 2005b*; *Korb, 2005*; *Morgun, 2010, 2019*; *Tshikolovets & Nekrutenko, 2012*; *Bolshakov & Korb, 2013*; *Stradomsky & Tikhonov, 2015*; *Korb & Bolshakov, 2016*). Currently, lepidopterology

in Georgia faces significant challenges due to the absence of successors to Professor Didmanidze in both scientific institutions and among actively involved enthusiasts. *Mumladze, Japoshvili & Anderson (2019)* indicate that only a few taxa—such as Coleoptera, Hemiptera, Hymenoptera, Acari (Oribatida), and Gastropoda—are actively studied in Georgia. Furthermore, there is a lack of comprehensive data regarding the distribution of taxa, particularly butterflies.

Twenty years have passed since Didmanidze's checklist was published (*Didmanidze, 2004*); therefore, an update is needed, especially since several new investigations have been carried out, recording new species or new localities for species previously noted in Georgia. Intense genetic studies in recent years have resulted in changes in systematics and increased the number of reported species not previously listed. Currently, apart from the information from *Didmanidze (2004)*, *Tshikolovets & Nekrutenko (2012)* and scarce data for individual species from four websites, detailed data on the distribution of most butterfly species in Georgia are not available. Based on the number of recorded taxa, several other species can be expected to occur in Georgia, including those reported in surrounding countries or new to science.

Therefore, in the present article, we provide an updated and annotated regional list of Georgian butterflies with vernacular names, species distribution in Georgia/Caucasus, occurrence status in each region of the country, phenological data and habitat requirements. Moreover, we present additional comments for the species with doubtful presence in Georgia and for the species that have not been previously reported in Georgia. This paper should assist in the understanding of present-day knowledge of Georgia's butterfly fauna and form a starting point for further studies into the biodiversity and conservation of butterflies in that country.

## MATERIALS AND METHODS

### Literature records

We need to point out that the checklist by *Didmanidze (2004)* contains errors such as redundant or missing letters in Latin names, nomenclatural inaccuracies, mistaken authors of species names, omissions, and incomplete or incorrect information about ranges of species. For that reason, we used *Tshikolovets & Nekrutenko (2012)* as our reference data source. We considered that the aforementioned authors summarised the important publications on the butterfly fauna of the Caucasus. Nevertheless, we also critically reviewed all available literature reports to verify original spelling, possible taxonomic changes and precise distributional information in each region (*e.g.*, *Eversmann, 1851*; *Alphéraky, 1876*; *Staudinger & Rebel, 1901*; *Moucha, 1968*; *Batiashvili & Didmanidze, 1973*; *Gross, 1978*; *Tuzov, 1993*; *Tuzov et al., 1997*, *2000*; *Bukhnikashvili et al., 2012*). We did not verify material for misidentified or doubtful species, as it was beyond the scope of the present study. Nonetheless, we have decided to include those records in the checklist. All species with uncertain or questionable occurrence in Georgia are marked with an asterisk (*) placed at the species name in all presented tables. Nevertheless, we included some comments about those species, as well as about some new findings. Due to the lack of data on the number of individuals of particular species, especially in older publications, we

defined species rarity using a criterion that focuses on their frequency of occurrence within the territory of Georgia. If a species was recorded from no more than three localities, we considered it rare; if it was recorded from only one locality, we regarded it as extremely rare within the Georgian fauna.

It is important to note that lepidopteran taxonomy and nomenclature are active fields of study, with ongoing revisions based on new discoveries and genetic analyses. The classification system continues to evolve, and for that reason we decided to base our study on the classification and nomenclature used in the recent publication by *Rajaei et al. (2023)*. The primary source of Russian vernacular names was the publication by *Lvovsky & Morgun (2007)*, while Georgian names were derived from various sources, primarily Georgian publications (including *Bukhnikashvili et al., 2012*) and the portal https:// biodiversity.iliauni.edu.ge./.

## GBIF records

The records obtained from the Global Biodiversity Information Facility (https://www.gbif. org/, accessed on 2 April 2022) have been subjected to a very critical assessment. All records containing obvious errors have been excluded—for example, records collected from New Georgia Island or from the state of Georgia (in the United States of America) which were erroneously assigned (in the GBIF) to Georgia in the Caucasus. The detailed locations of all GBIF records were also verified. All records with missing Global Positioning System (GPS) coordinates or those indicating localities outside the border of present-day Georgia (*i.e.*, localities in the territory of Russia or Turkey) have been excluded. All records described as human observation that did not contain an image confirming such observation were also excluded. All human observations, confirmed by photographs of species that could be confused with other species, were additionally verified.

## Other sources of records

We had access to records submitted by contributors to the Lepidoptera Mundi website (*Jonko & Słowińska, 2023*) and two other websites: the Georgian Biodiversity Database (*Tarkhnishvili & Chaladze, 2023*), Butterflies of the Caucasus and the South of Russia (*Tikhonov et al., 2023*). Additionally, we used data based on the material photographed by us during the Caucasus expeditions in the years 2018–2019 under the project "V4 & Eastern Partnership". All observed individuals were photographed and identified in the field, and then they were included in the general database.

## Database and visualisation

For the purpose of our study, a Microsoft Access 2010 relational database was created locally on the authors' computers and subsequently migrated to the MySQL platform, based on all available data sources. The database consists of several tables containing information about taxonomy, regions, locations, vernacular names, dates of individual records and the period in which the record was noted originally. It includes taxa identified at least to the species level and records with detailed localities or at least region name. The

final checklist has been created for the species level. All taxa within families are arranged in systematic order. The database of faunistic records in SQL/MS Access format is available from the authors upon request.

Visualisation of the regional distribution maps of all species was generated using the GD library and scripts written in PHP. Moreover, the thumbnails of dorsal and ventral views of each species are presented (where available). The main source of the thumbnails is the authors' website Lepidoptera Mundi (*Jonko & Słowińska, 2023*). All of the thumbnails were created from photos of pinned specimens, with the photos edited in Adobe Photoshop CS6. We note that the presented thumbnails are not all generated to the same scale and do not contain a 1 cm reference bar. These images are used only for the depiction of a species and may not represent individuals collected in the Caucasus.

To clearly separate historical and more recent data, we decided to divide all records in the database into three periods: very old (historical) records up to 1945 (end of the Second World War), records collected between the end of the Second World War and Georgia's independence (1945–1991), and records collected since Georgia gained its independence (post-1991). This division was made not for any scientific reasons, but solely based on historical events.

## RESULTS

### List of Georgian butterflies

Based on the most up-to-date butterfly checklist of the Caucasus by *Tshikolovets & Nekrutenko (2012)*, we registered 201 species confirmed in Georgia. The current systematics forced a recalculation of the number of species listed by the aforementioned authors to make it comparable with the current state of knowledge. In summary, prior to the present study, 205 species of butterflies representing five families were recorded in Georgia. In the present article, 39 species were added to this list and, as a result, a total of 244 species belonging to 85 genera are listed for Georgia. The greatest species richness was recorded within the families Nymphalidae (96 species, 34 genera) and Lycaenidae (83 species, 26 genera) (Table 1).

Sixteen species (6.6% of Georgian fauna) were considered Caucasian endemics (see Tables 1 and 2), while four species (*Carterocephalus palaemon*, *Parnassius phoebus*, *Pieris bowdeni*, *Polyommatus admetus*) have not been noted since the end of 19th century. Moreover, for eleven species: *Muschampia baeticus*, *Agriades optilete*, *Polyommatus escheri*, *Tarucus theophrastus*, *Satyrium lunulata*, *Melitaea alatauica*, *M. casta*, *Hyponephele amardaea*, *H. comara*, *Melanargia hylata* and *M. teneates* no information confirming their occurrence in Georgia is available (Table 2) while over twenty other species (recorded mainly by Didmanidze) require confirmation or verification of the identifications.

A complete listing of taxa with known subspecies is available in Table S1 in Supplemental Material.

In total, 7,695 records were found during the study periods, including 3,849 extracted from GBIF. Among them, 7,180 have locality names or geographical coordinates that indicate precise location. In examining the number of records across each period, an

**Table 1 Summary of the number of subfamilies, genera and species within each butterfly family in Georgia.**

| Family | Subfamily | Genera | Species | Endemic* |
|---|---|---|---|---|
| Hesperiidae | 3 | 12 | 29 | 1 |
| Lycaenidae | 3 | 26 | 83 | 6 |
| Nymphalidae | 6 | 34 | 96 | 7 |
| Papilionidae | 2 | 4 | 8 | 1 |
| Pieridae | 3 | 9 | 28 | 1 |
| Total | 17 | 85 | 244 | 16 |

Note:
* Endemic to the Caucasus.

upwards trend was apparent. The number of records was the lowest (822) in the first historical period (up to 1945), while in the second period 2,381 records were reported. The highest number of records (4,492) was noted in the last period (post-1991) (Fig. 2). The increased number of records in the last period was mainly due to large scale user activity on Internet platforms like iNaturalist or GBIF.

## List of comments on new and doubtful species

After an examination of all available publications dealing with Georgian butterflies, we conclude that the presence of several dozen species in Georgia is not certain, while several other species reported earlier need confirmation or further systematic review. Since Didmanidze's checklist (*Didmanidze, 2004*), several faunistic publications and taxonomic studies incorporating DNA-based techniques have appeared. Hereafter, we have added comments to the species whose status rank has changed due to new genetic research, as well as species whose occurrence in Georgia is doubtful due to the availability of conflicting or ambiguous information/data, misidentifications, or a long-term lack of new records (Table 3).

## Distribution, habitat, phenology of Georgian butterflies

Most of recorded species are considered as widely distributed and common in both Georgia and the Caucasus. Twelve species: *Muschampia cribrellum*, *Pyrgus cacaliae*, *P. carthami*, *P. serratulae*, *Euphydryas iduna*, *Limenitis populi*, *Coenonympha tullia*, *Aricia teberdina*, *Cupido alcetas*, *Kretania sephirus*, *K. stekolnikovi* and *Polyommatus eros* are confined only to the Great Caucasus, while sixteen: *Colias caucasica*, *Pieris bowdeni*, *Melitaea ornata*, *Coenonympha symphita*, *Hipparchia fatua*, *Pseudochazara beroe*, *P. geyeri*, *P. mniszechii*, *Lycaena asabinus*, *Kretania modica*, *Polyommatus admetus*, *P. cyaneus*, *P. ninae*, *P. aserbeidschana*, *P. turcicus*, *P. damon* are restricted to the Lesser Caucasus. Seven species: *Eogenes alcides*, *Muschampia poggei*, *Papilio alexanor*, *Afarsia morgiana*, *Polyommatus altivagans*, *Tomares romanovi*, *Hyponephele comara* may be found only in the Djavakheti-Armenian plateau. The highest number of butterfly species are found in July, and almost half of the species are strictly univoltine (Table 4).

We created and provided the first list of Georgian vernacular names for butterfly species, which is as complete as possible (see Table 4). Such a list has not yet been

**Table 2 Updated list of butterflies of Georgia.**

| Abbr | Region name–English | Region name–Georgian |
|---|---|---|
| AB | Abkhazia | აფხაზეთი |
| AJ | Adjara | აჭარა |
| GU | Guria | გურია |
| IM | Imereti | იმერეთი |
| KA | K'akheti | კახეთი |
| KK | Kvemo Kartli | ქვემო ქართლი |
| MM | Mtskheta-Mtianeti | მცხეთა-მთიანეთი |
| RL | Rach'a-Lechkhumi-Kvemo Svaneti | რაჭა-ლეჩხუმი და ქვემო სვანეთი |
| SJ | Samtskhe-Javakheti | სამცხე-ჯავახეთი |
| SK | Shida Kartli | შიდა ქართლი |
| SZ | Samegrelo-Zemo Svaneti | სამეგრელო-ზემო სვანეთი |
| TB | Tbilisi | თბილისი |

Administrative divisions of Georgia

Table symbols legend:
○ Historical records (collected up to the 1945)
● Old records (collected between 1946 and 1991)
■ Recent records (collected after 1991)

| № | Species name | AB | AJ | GU | IM | KA | KK | MM | RL | SJ | SK | SZ | TB | [c] |
|---|---|---|---|---|---|---|---|---|---|---|---|---|---|---|
| | **Hesperiidae** Latreille, 1802 | | | | | | | | | | | | | |
| | **Pyrginae** Burmeister, 1878 | | | | | | | | | | | | | |
| 1 | *Erynnis tages* (Linnaeus, 1758) | ● | ■ | ● | ■ | ■ | ■ | ■ | ● | ■ | ● | ■ | ● | |
| 2 | *Erynnis marloyi* (Boisduval, 1834) | ● | | | | | ■ | ■ | | | | | ■ | |
| 3 | *Carcharodus alceae* (Esper, 1780) | ● | ■ | | ● | ■ | ■ | | | ■ | | | ■ | |
| 4 | *Carcharodus lavatherae* (Esper, 1783) | | | | | | ■ | ● | | ○ | | | ○ | |
| 5 | *Carcharodus floccifera* (Zeller, 1847) | ● | ● | | | ■ | ● | | | ● | | | ○ | |
| 6 | *Carcharodus orientalis* Reverdin, 1913 | ● | ● | | ● | | ■ | | | | | ● | | |
| 7 | *Spialia orbifer* (Hübner, 1823) | ● | | | ○ | ■ | ■ | | | ● | | | ■ | 1 |
| 8 | *Favria cribrellum* (Eversmann, 1841) | | | | | | ● | | | | | ○ | | 2 |
| 9 | *Muschampia tessellum* (Hübner, 1803) | | ● | | | | | | | | | ○ | | |
| 10* | *Muschampia baeticus* (Rambur, 1839) | | ● | ● | ● | | | | | ● | | | | 3 |
| 11 | *Muschampia poggei* (Lederer, 1858) | | | | | | ● | ● | | | | | | 4 |
| 12 | *Pyrgus melotis* (Duponchel, 1834) | ● | ○ | ● | | ■ | ■ | ■ | ■ | ■ | | ● | ● | |
| 13 | *Pyrgus serratulae* (Rambur, 1839) | ● | | ○ | | ○ | ● | ■ | | ● | | | | |
| 14 | *Pyrgus armoricanus* (Oberthür, 1910) | ■ | | | | ■ | ● | ■ | | ● | | ■ | ■ | |
| 15 | *Pyrgus alveus* Hübner, 1803 | ● | | | | ● | ■ | ■ | | ■ | | ● | ○ | |
| 16* | *Pyrgus jupei* (Alberti, 1967) | | ● | | ■ | | ● | | | ● | | | | |
| 17 | *Pyrgus sidae* (Esper, 1784) | | | | | ■ | ■ | ● | | ■ | | | ● | |
| 18 | *Pyrgus cacaliae* (Rambur, 1839) | ● | | | | | | | | | | | | |
| 19 | *Pyrgus cinarae* (Rambur, 1839) | | | | | | ● | ● | | | | | | |
| 20 | *Pyrgus carthami* (Hübner, 1813) | | | | | ● | ○ | | | ● | | | | 5 |
| | **Heteropterinae** Aurivillius, 1925 | | | | | | | | | | | | | |
| 21 | *Carterocephalus palaemon* (Pallas, 1771) | | | | | | | | | ○ | | | | 6 |
| 22 | *Heteropterus morpheus* (Pallas, 1771) | ● | ● | | ● | | | | | | | ● | | |
| | **Hesperiinae** Latreille, 1809 | | | | | | | | | | | | | |
| 23 | *Eogenes alcides* (Herrich-Schäffer, 1852) | | | | | ■ | | | | | | | ● | |
| 24 | *Gegenes nostrodamus* (Fabricius, 1793) | | | | | ● | ■ | | | | | | ● | |

| № | Species name | AB | AJ | GU | IM | KA | KK | MM | RL | SJ | SK | SZ | TB | [c] |
|---|---|---|---|---|---|---|---|---|---|---|---|---|---|---|
| 25 | *Thymelicus hyrax* (Lederer, 1861) | | | | | | ● | | | | | | | |
| 26 | *Thymelicus sylvestris* (Poda, 1761) | ● | | | ■ | ■ | | ■ | | ■ | ● | | ● | |
| 27 | *Thymelicus lineola* (Ochsenheimer, 1808) | ● | ■ | | ■ | ■ | ■ | ■ | | ■ | | | ■ | |
| 28 | *Ochlodes sylvanus* (Esper, 1777) | ● | ● | | ■ | ■ | ■ | | | ■ | ● | ■ | ■ | |
| 29 | *Hesperia comma* (Linnaeus, 1758) | ■ | ■ | | ■ | ■ | | ■ | | ○ | ■ | ■ | | |
| | **Papilionidae** Latreille, 1802 | | | | | | | | | | | | | |
| | **Parnassiinae** Duponchel, 1835 | | | | | | | | | | | | | |
| 30 | *Parnassius mnemosyne* (Linnaeus, 1758) | ● | ■ | | ■ | ○ | ● | | | ■ | | ■ | ○ | |
| 31* | *Parnassius nordmanni* Ménétriés, 1850 | ● | ● | ● | | | ● | ■ | ■ | | | ■ | | |
| 32 | *Parnassius phoebus* (Fabricius, 1793) | | ○ | | | | | | | | | | | |
| 33 | *Parnassius apollo* (Linnaeus, 1758) | ● | ■ | | ● | ■ | | ■ | | ■ | | ■ | ○ | |
| 34 | *Zerynthia caucasica* (Lederer, 1864) | ● | ■ | | ■ | ● | | | ● | ○ | | ● | | |
| | **Papilioninae** Latreille, 1802 | | | | | | | | | | | | | |
| 35 | *Iphiclides podalirius* (Linnaeus, 1758) | ■ | ■ | | ■ | ■ | | ■ | | ■ | ● | | ■ | |
| 36 | *Papilio alexanor* Esper, 1800 | | | | | ■ | | | | | | | | 7 |
| 37 | *Papilio machaon* Linnaeus, 1758 | ● | ■ | | ■ | ■ | ■ | ■ | | ■ | | ■ | ■ | |
| | **Pieridae** Swainson, 1820 | | | | | | | | | | | | | |
| | **Dismorphiinae** Schatz, 1887 | | | | | | | | | | | | | |
| 38 | *Leptidea sinapis* (Linnaeus, 1758) | ● | ● | | ○ | ■ | ■ | ● | | ■ | ● | ○ | ■ | 8 |
| 39 | *Leptidea juvernica* Williams, 1946 | ? | ? | | ■ | ■ | ? | ■ | | ? | ? | ? | ? | 8 |
| 40 | *Leptidea duponcheli* (Staudinger, 1871) | | | | | ■ | ■ | | | ■ | | | ■ | |
| | **Pierinae** Duponchel, 1835 | | | | | | | | | | | | | |
| 41 | *Anthocharis cardamines* (Linnaeus, 1758) | ● | ■ | | | ■ | ■ | ■ | | ■ | ● | ● | ■ | |
| 42 | *Anthocharis damone* Boisduval, 1836 | | | | | | ● | | | ■ | | | | |
| 43 | *Anthocharis gruneri* Herrich-Schäffer, 1851 | | | | | | | | | ○ | | | ■ | |
| 44 | *Zegris eupheme* (Esper, 1804) | | | | | ● | ● | | | ● | | | ● | |
| 45 | *Euchloe ausonia* (Hübner, 1804) | | ● | | ● | ■ | ■ | ■ | | ■ | | | ■ | |
| 46 | *Aporia crataegi* (Linnaeus, 1758) | ■ | ● | | | ■ | ■ | ■ | ■ | ■ | ● | ■ | ■ | |
| 47 | *Pieris brassicae* (Linnaeus, 1758) | ● | ■ | ● | ■ | ■ | ■ | ■ | ● | ■ | ● | ○ | ■ | |
| 48 | *Pieris mannii* (Mayer, 1851) | | | | | ● | | ● | ○ | | | | | 9 |
| 49 | *Pieris rapae* (Linnaeus, 1758) | ■ | ■ | | ○ | ■ | ■ | ■ | | ■ | | ○ | ■ | |
| 50 | *Pieris ergane* (Geyer, 1828) | | ● | ● | | ■ | | | | ● | | | | |
| 51 | *Pieris napi* (Linnaeus, 1758) | ● | ■ | | ■ | | ■ | ■ | | ■ | ● | | ● | |
| 52 | *Pieris bryoniae* (Hübner, 1806) | ● | | | | | | ● | ■ | | | ■ | | |
| 53* | *Pieris bowdeni* Eitschberger, 1984 | | | | | | | | | ○ | | | | 10 |
| 54 | *Pontia callidice* (Hübner, 1800) | | | | | ■ | ○ | ○ | ● | ● | ● | | | |
| 55 | *Pontia edusa* (Fabricius, 1777) | ● | ■ | ● | ● | ■ | ■ | ■ | ● | ■ | ■ | ● | ■ | 11 |
| 56 | *Pontia chloridice* (Hübner, 1813) | | | | | | | | | ■ | | | | |
| | **Coliadinae** Swainson, 1827 | | | | | | | | | | | | | |
| 57 | *Colias erate* (Esper, 1805) | ● | ■ | | ● | | ● | ● | | ○ | ● | ● | ■ | |
| 58 | *Colias croceus* (Geoffroy, 1785) | ● | ■ | ● | ■ | ■ | ■ | ■ | ● | ■ | ■ | ■ | ■ | |

(Continued)

| № | Species name | AB | AJ | GU | IM | KA | KK | MM | RL | SJ | SK | SZ | TB | [c] |
|---|---|---|---|---|---|---|---|---|---|---|---|---|---|---|
| 59 | *Colias chrysotheme* (Esper, 1781) | ● | ○ |  | ● | ● | ● |  | ● | ● |  | ● |  | 12 |
| 60 | *Colias aurorina* Herrich-Schäffer, 1850 | ○ |  |  | ○ | ■ | ● |  |  | ● |  |  | ○ |  |
| 61 | *Colias caucasica* Staudinger, 1871 |  |  |  | ○ | ● | ● |  |  | ■ |  |  |  |  |
| 62 | *Colias thisoa* Ménétriés, 1832 |  |  |  | ● | ● |  |  |  | ● |  |  |  |  |
| 63 | *Colias alfacariensis* Ribbe, 1905 | ● | ● |  | ● |  | ■ | ■ |  | ■ | ■ | ■ | ■ | 13 |
| 64 | *Gonepteryx rhamni* (Linnaeus, 1758) | ● | ■ | ● | ■ | ■ | ■ | ■ | ■ | ■ | ● | ■ | ■ |  |
| 65 | *Gonepteryx farinosa* (Zeller, 1847) |  | ● |  |  |  | ■ | ■ |  | ● |  |  | ■ |  |
| | **Lycaenidae** Leach, 1815 | | | | | | | | | | | | | |
| | **Lycaeninae** Leach, 1815 | | | | | | | | | | | | | |
| 66 | *Lycaena phlaeas* (Linnaeus, 1760) | ● | ■ |  | ■ | ■ | ■ | ■ |  | ■ | ● | ■ | ■ |  |
| 67 | *Lycaena helle* Denis & Schiffermüller, 1775 | ● |  |  |  |  |  |  |  | ○ |  | ■ |  |  |
| 68 | *Lycaena dispar* (Haworth, 1802) | ● |  | ● | ■ | ■ | ■ | ● |  | ● | ● | ■ |  |  |
| 69 | *Lycaena virgaureae* (Linnaeus, 1758) | ● | ● | ● | ● | ● | ● | ■ | ■ | ● | ● | ■ |  |  |
| 70 | *Lycaena tityrus* Poda, 1761 | ■ | ● |  | ● | ● | ■ | ■ |  | ■ |  | ■ | ■ |  |
| 71 | *Lycaena alciphron* Rottemburg, 1775 |  |  |  |  | ● | ■ | ■ |  | ■ |  |  | ■ |  |
| 72 | *Lycaena candens* (Herrich-Schäffer, 1844) | ■ |  |  | ○ | ■ | ● | ■ |  | ■ |  | ● |  | 14 |
| 73 | *Lycaena thersamon* (Esper, 1784) | ● |  |  | ○ | ● | ● | ● |  | ■ |  | ● | ■ |  |
| 74 | *Lycaena thetis* Klug, 1834 |  |  |  |  | ● |  |  |  |  |  |  |  | 15 |
| 75* | *Lycaena japhetica* (Nekrutenko & Effendi, 1983) |  |  |  |  |  |  |  |  |  | ■ |  |  | 16 |
| 76 | *Lycaena asabinus* (Herrich-Schäffer, 1851) |  |  |  |  | ● | ● |  |  | ● |  |  |  |  |
| 77 | *Lycaena ochimus* (Herrich-Schäffer, 1851) | ● |  |  |  |  | ● |  |  | ● |  |  |  |  |
| | **Polyommatinae** Swainson, 1827 | | | | | | | | | | | | | |
| 78 | *Lampides boeticus* (Linnaeus, 1767) | ● | ■ |  | ● | ● |  |  |  | ■ |  | ■ |  |  |
| 79 | *Leptotes pirithous* (Linnaeus, 1767) | ■ | ● | ● | ● |  |  |  |  |  |  | ● |  |  |
| 80* | *Tarucus theophrastus* (Fabricius, 1793) |  |  |  |  | ● | ○ |  |  |  |  |  |  | 17 |
| 81 | *Tarucus balkanica* (Freyer, 1844) | ● |  |  |  | ■ | ■ | ■ |  |  |  |  | ■ |  |
| 82 | *Cupido minimus* (Fuessly, 1775) | ● | ■ |  |  | ● | ● | ■ |  | ■ |  | ● | ● |  |
| 83 | *Cupido osiris* (Meigen, 1829) |  |  |  | ● | ● | ● |  |  | ● | ● | ○ |  |  |
| 84 | *Cupido argiades* (Pallas, 1771) | ● | ■ |  | ■ | ● | ● |  |  | ■ | ● | ■ | ● |  |
| 85 | *Cupido alcetas* (Hoffmannsegg, 1804) | ● |  |  |  |  |  |  |  |  |  |  |  |  |
| 86 | *Celastrina argiolus* (Linnaeus, 1758) | ● |  | ● | ● |  |  |  |  | ● |  | ■ |  |  |
| 87 | *Pseudophilotes vicrama* (Moore, 1865) |  | ● | ● |  | ■ | ● | ■ |  | ■ |  |  |  | 18 |
| 88 | *Scolitantides orion* (Pallas, 1771) |  | ● |  |  |  |  |  |  |  |  |  |  | 19 |
| 89 | *Glaucopsyche alexis* (Poda, 1761) | ● | ● |  | ● |  | ● | ● |  |  |  |  | ■ |  |
| 90 | *Phengaris arion* (Linnaeus, 1758) | ■ |  |  | ○ | ■ | ■ | ● |  | ■ | ○ |  |  |  |
| 91 | *Phengaris teleius* (Bergsträsser, 1779) |  |  |  |  |  |  |  |  | ● |  | ● |  |  |
| 92 | *Phengaris nausithous* (Bergsträsser, 1779) |  |  |  |  | ● | ■ |  |  |  |  |  |  |  |
| 93 | *Phengaris alcon* (Denis & Schiffermüller, 1775) |  |  |  | ● |  | ■ |  |  | ■ | ● |  |  |  |
| 94 | *Luthrodes galba* (Lederer, 1855) |  |  |  |  | ● |  |  |  |  |  |  |  |  |
| 95 | *Plebejus argus* (Linnaeus, 1758) | ● | ■ | ● | ● | ● | ■ | ■ | ■ | ■ | ■ | ■ | ● |  |
| 96 | *Plebejus idas* (Linnaeus, 1761) | ● |  |  |  |  | ■ |  |  | ■ |  |  |  |  |

| № | Species name | AB | AJ | GU | IM | KA | KK | MM | RL | SJ | SK | SZ | TB | [c] |
|---|---|---|---|---|---|---|---|---|---|---|---|---|---|---|
| | **Table 2 (continued)** | | | | | | | | | | | | | |
| 97 | *Plebejus argyrognomon* (Bergsträsser, 1779) | ● | ■ | | ■ | | | ■ | | ■ | ● | ■ | ● | |
| 98 | *Plebejus christophi* (Staudinger, 1874) | | | | | | | | | ● | | | | 20 |
| 99 | *Kretania sephirus* (Frivaldszky, 1835) | ● | | | | | ■ | | | | | | | 21 |
| 100 | *Kretania modica* (Vérity, 1935) | | | | | | | | | ■ | | | | 21 |
| 101 | *Kretania stekolnikovi* Stradomsky & Tikhonov, 2015 | | | | ■ | | | | | | | | | 21 |
| 102 | *Kretania eurypilus* (Freyer, 1851) | | | | | ● | | | | ● | | | | |
| 103 | *Plebejidea loewii* (Zeller, 1847) | | | | | | | | | ■ | | | | |
| 104 | *Aricia agestis* (Denis & Schiffermüller, 1775) | ● | | | ● | ■ | ■ | ■ | | ■ | ■ | ■ | ■ | |
| 105 | *Aricia artaxerxes* (Fabricius, 1793) | ● | | | | | ○ | ■ | | ■ | | ■ | | |
| 106 | *Aricia anteros* (Freyer, 1838) | | | | ● | ■ | | ■ | | ■ | ● | ■ | | |
| 107 | *Eumedonia eumedon* (Esper, 1780) | ■ | | | ○ | ■ | ● | ■ | ■ | ■ | ■ | | | |
| 108 | *Cyaniris semiargus* (Rottemburg, 1775) | ● | ● | | | ■ | ■ | ■ | | ■ | ■ | ■ | ■ | |
| 109* | *Agriades optilete* (Knoch, 1781) | | | | | | ● | | | ● | | ● | | 22 |
| 110 | *Agriades dardanus* (Freyer, 1843) | ● | | | ● | ■ | | ■ | | ■ | | ■ | | 23 |
| 111 | *Lysandra bellargus* (Rottemburg, 1775) | ● | ● | | | ■ | ■ | ■ | | ■ | ■ | ○ | ● | |
| 112 | *Lysandra corydonius* (Herrich-Schäffer, 1852) | ● | | | ● | ■ | ■ | ■ | ■ | ○ | ● | ■ | | 24 |
| 113 | *Neolysandra coelestina* (Eversmann, 1843) | | | | | ■ | | ● | | ■ | | | | |
| 114* | *Polyommatus escheri* (Hübner, 1823) | ● | | | | | | | | ● | | | | 25 |
| 115 | *Polyommatus dorylas* (Denis & Schiffermüller, 1775) | | | | ○ | | ○ | ■ | | ■ | | | | |
| 116 | *Polyommatus amandus* (Schneider, 1792) | ● | | | | ● | ■ | ■ | | ■ | | ■ | ○ | |
| 117 | *Polyommatus thersites* (Cantener, 1835) | | ● | | | ■ | ■ | ■ | | ■ | ● | | ■ | |
| 118 | *Polyommatus icarus* (Rottemburg, 1775) | ● | ■ | ● | ■ | ■ | ■ | ■ | ● | ■ | ● | ● | ■ | 26 |
| 119 | *Polyommatus eros* (Ochsenheimer, 1808) | ● | | | | ● | | ■ | | | | ■ | | |
| 120 | *Polyommatus daphnis* (Denis & Schiffermüller, 1775) | ● | ● | | ○ | ● | ■ | ■ | | ■ | ● | ■ | ■ | |
| 121 | *Polyommatus admetus* (Esper, 1783) | | | | | | | | | ○ | | | ○ | 27 |
| 122 | *Polyommatus ripartii* (Freyer, 1830) | | | | | | ■ | ■ | | | ● | | | |
| 123* | *Polyommatus eriwanensis* (Forster, 1960) | ● | | | | | | | | | | | ● | |
| 124 | *Polyommatus damon* (Denis & Schiffermüller, 1775) | | | | | | ● | | | ■ | | | | |
| 125* | *Polyommatus wagneri* (Forster, 1956) | | | | | | | | | ● | | | | 29 |
| 126 | *Polyommatus altivagans* (Forster, 1956) | | | | | ● | | | | | | | | 29 |
| 127 | *Polyommatus cyanea* (Staudinger, 1899) | | | | | ● | | ● | | ■ | | | | 30 |
| 128 | *Polyommatus poseidon* (Herrich-Schäffer, 1851) | | | | | | | | | ■ | | | | 31 |
| 129* | *Polyommatus teberdina* (Sheljuzhko, 1934) | ● | | | | ● | | ● | | | ● | | | |
| 130 | *Polyommatus turcicus* (Koçak, 1977) | | | | | | | | | ■ | | | | |
| 131 | *Polyommatus aserbeidschana* (Forster, 1956) | | | | | | | | | ■ | | | | 32 |
| 132 | *Polyommatus ninae* (Forster, 1956) | | | | | | | | | | | | ○ | 33 |
| 133 | *Polyommatus demavendi* (Pfeiffer, 1938) | | | | | | | | | ● | | | | |
| 134 | *Afarsia morgiana* (Kirby, 1871) | | | | | | ● | | | | | | ● | 34 |
| | **Theclinae** Swainson, 1831 | | | | | | | | | | | | | |
| 135 | *Satyrium w-album* (Knoch, 1782) | ● | ● | | | ■ | | | ○ | ● | | ● | | |
| 136 | *Satyrium spini* (Denis & Schiffermüller, 1775) | | | | | ■ | ● | | | ■ | | | ● | |

(Continued)

| № | Species name | AB | AJ | GU | IM | KA | KK | MM | RL | SJ | SK | SZ | TB | [c] |
|---|---|---|---|---|---|---|---|---|---|---|---|---|---|---|
| | **Table 2** (continued) | | | | | | | | | | | | | |
| 137 | *Satyrium abdominalis* (Gerhard, 1850) | | | | | ■ | | ● | | ● | | | | 35 |
| 138 | *Satyrium ilicis* (Esper, 1779) | ● | | | ● | ● | ● | ■ | | ■ | ● | | ● | |
| 139 | *Satyrium acaciae* (Fabricius, 1787) | | | | | ● | ● | | | ● | | | ○ | |
| 140 | *Satyrium ledereri* (Boisduval, 1848) | | | | | ■ | ● | | | ■ | | | ■ | |
| 141 | *Satyrium hyrcanica* (Riley, 1939) | | | | | ● | | | | | | | | |
| 142* | *Satyrium lunulata* (Erschoff, 1874) | | | | | ● | | | | | | | | 36 |
| 143* | *Callophrys mystaphia* Miller, 1913 | | | | | ● | | | | | | | | 37 |
| 144 | *Callophrys chalybeitincta* Sovinsky, 1905 | ● | ■ | | | ■ | ● | ● | | ● | ● | ■ | ■ | 38 |
| 145 | *Thecla betulae* (Linnaeus, 1758) | ● | | | ● | ● | | | | ■ | | | | |
| 146 | *Favonius quercus* (Linnaeus, 1758) | ● | | | ■ | ● | ■ | ● | | ■ | | | ■ | |
| 147 | *Tomares callimachus* (Eversmann, 1848) | | | | | ■ | ● | | | ● | | | ● | |
| 148* | *Tomares romanovi* (Christoph, 1882) | | | | | ■ | | | | | | | | |
| | **Nymphalidae** Rafinesque, 1815 | | | | | | | | | | | | | |
| | **Limenitidinae** Behr, 1864 | | | | | | | | | | | | | |
| 149 | *Limenitis populi* (Linnaeus, 1758) | ● | | | | | | | | | | | | |
| 150 | *Limenitis camilla* (Linnaeus, 1764) | ● | ● | | ● | | ■ | ■ | | ■ | ● | | ■ | |
| 151 | *Limenitis reducta* Staudinger, 1901 | ● | ● | | ○ | ● | ■ | ■ | | ■ | ● | ○ | ■ | |
| 152 | *Neptis rivularis* (Scopoli, 1763) | ● | | | | ■ | ■ | ■ | | ■ | | | ■ | |
| | **Heliconiinae** Swainson, 1822 | | | | | | | | | | | | | |
| 153* | *Boloria caucasica* (Lederer, 1852) | ● | | ○ | ● | | ■ | | ■ | ■ | | ■ | | 39 |
| 154 | *Boloria eunomia* (Esper, 1800) | | | | | | | | ○ | | | | | 40 |
| 155 | *Boloria euphrosyne* (Linnaeus, 1758) | ● | ■ | ● | ● | | ● | ■ | | | ■ | ● | | |
| 156 | *Boloria selene* (Denis & Schiffermüller, 1775) | | | ● | ● | | ● | ● | | | | | | 41 |
| 157 | *Boloria dia* (Linnaeus, 1767) | ● | ● | ● | ● | ■ | ■ | ■ | | ■ | ● | ● | ○ | |
| 158 | *Issoria lathonia* (Linnaeus, 1758) | ● | ● | ● | ● | ■ | ■ | ■ | | ■ | ■ | ■ | ■ | |
| 159 | *Argynnis paphia* (Linnaeus, 1758) | ● | ■ | ■ | ■ | ■ | ■ | ■ | ■ | ■ | ■ | ■ | ■ | |
| 160 | *Argynnis pandora* (Denis & Schiffermüller, 1775) | ● | | | ● | ■ | ■ | ■ | | ■ | ■ | ■ | ■ | |
| 161 | *Speyeria aglaja* (Linnaeus, 1758) | ● | ● | | | ■ | ■ | ■ | ○ | ■ | ■ | ■ | ■ | |
| 162 | *Fabriciana adippe* (Denis & Schiffermüller, 1775) | ● | ● | | | ■ | ● | ■ | ○ | ■ | ■ | ■ | ● | |
| 163 | *Fabriciana niobe* (Linnaeus, 1758) | ● | | | | ■ | ■ | ■ | | ■ | ● | | ● | |
| 164 | *Brenthis ino* (Rottemburg, 1775) | | | | | ■ | ○ | | | ■ | | | | |
| 165 | *Brenthis daphne* (Bergsträsser, 1780) | ● | | | | ■ | | | | ■ | | | | |
| 166 | *Brenthis hecate* (Denis & Schiffermüller, 1775) | | | | | ■ | ■ | ● | | ■ | | ■ | ■ | |
| | **Apaturinae** Boisduval, 1840 | | | | | | | | | | | | | |
| 167 | *Apatura ilia* (Denis & Schiffermüller, 1775) | ● | | | ■ | | ● | ■ | ■ | ● | | | ● | |
| 168 | *Thaleropis ionia* Eversmann, 1851 | | ■ | | | | | | | | | | | 42 |
| | **Nymphalinae** Swainson, 1827 | | | | | | | | | | | | | |
| 169 | *Melitaea cinxia* (Linnaeus, 1758) | ● | ■ | | ■ | ■ | ■ | ■ | | ■ | | ■ | ● | |
| 170 | *Melitaea arduinna* (Esper, 1783) | | | | | ● | | | | ■ | ● | | ● | |
| 171 | *Melitaea diamina* (Lang, 1789) | ● | | | ● | | | ■ | | ■ | ■ | | | |
| 172 | *Melitaea phoebe* (Denis & Schiffermüller, 1775) | ● | ● | | ○ | ■ | ■ | | | ● | ● | | ● | |

| № | Species name | AB | AJ | GU | IM | KA | KK | MM | RL | SJ | SK | SZ | TB | [c] |
|---|---|---|---|---|---|---|---|---|---|---|---|---|---|---|
| | **Table 2 (continued)** | | | | | | | | | | | | | |
| 173 | *Melitaea ornata* Christoph, 1893 | | | | | | ● | | | ● | | | | |
| 174 | *Melitaea interrupta* Kolenati, 1846 | ● | | | ● | ■ | ■ | ■ | ■ | ■ | | ■ | ● | 43 |
| 175 | *Melitaea persea* Kollar, 1850 | | | | | ■ | ● | | | | | | | |
| 176* | *Melitaea casta* (Kollar, 1848) | | | | | | | ● | | | | | | 44 |
| 177 | *Melitaea trivia* (Denis & Schiffermüller, 1775) | | | | | ● | | ■ | | ○ | | | | |
| 178 | *Melitaea athalia* (Rottemburg, 1775) | ● | ● | | ● | ■ | ■ | ● | ● | ■ | | | ● | |
| 179* | *Melitaea caucasogenita* Vérity, 1930 | | | | | | | ■ | | ■ | ■ | ■ | ■ | |
| 180 | *Melitaea aurelia* Nickerl, 1850 | | | | | | | | | ■ | | | | |
| 181* | *Melitaea alatauica* Staudinger, 1881 | ● | | | | | | | | ● | | | | 45 |
| 182 | *Euphydryas iduna* (Dalman, 1816) | | | | | ● | | ● | | | | | | |
| 183 | *Euphydryas aurinia* (Rottemburg, 1775) | | ● | | ● | ■ | | ● | | ● | | | ● | |
| 184 | *Euphydryas discordia* Bolshakov & Korb, 2013 | | | | | | | | | ■ | | | | 46 |
| 185 | *Euphydryas orientalis* (Herrich-Schäffer, 1851) | | | | ● | | | | | | | | | 47 |
| 186 | *Vanessa atalanta* (Linnaeus, 1758) | ■ | ■ | ■ | ■ | ■ | ■ | ■ | ■ | ■ | ● | ● | ● | |
| 187 | *Vanessa cardui* (Linnaeus, 1758) | ● | ■ | ● | ■ | ■ | ■ | ■ | ● | ■ | ● | ● | ■ | |
| 188 | *Araschnia levana* (Linnaeus, 1758) | ● | | | | | ● | | | | | | | |
| 189 | *Aglais urticae* (Linnaeus, 1758) | ● | ● | ● | ■ | ■ | ■ | ● | ■ | ■ | ● | ● | ● | |
| 190 | *Aglais io* (Linnaeus, 1758) | ■ | ■ | ● | ● | ■ | ■ | ■ | ■ | ■ | ● | ● | ■ | |
| 191 | *Nymphalis antiopa* (Linnaeus, 1758) | ■ | | | ■ | ■ | ■ | ● | ● | ● | ● | ● | ● | |
| 192 | *Nymphalis polychloros* (Linnaeus, 1758) | ● | ● | | ■ | | ● | ■ | | ● | ● | ● | ■ | |
| 193 | *Nymphalis xanthomelas* (Denis & Schiffermüller, 1775) | | | | | | | | ● | ● | | | | |
| 194 | *Nymphalis vaualbum* (Denis & Schiffermüller, 1775) | ● | | | | | | | | ○ | | | ■ | |
| 195 | *Polygonia egea* (Cramer, 1775) | | | | | ○ | ■ | ■ | | ■ | | | ■ | |
| 196 | *Polygonia c-album* (Linnaeus, 1758) | ● | ■ | ● | ● | ■ | ■ | ● | ■ | ■ | ● | ● | ● | |
| | **Satyrinae** Boisduval, 1833 | | | | | | | | | | | | | |
| 197 | *Coenonympha pamphilus* (Linnaeus, 1758) | ● | ● | ● | ■ | ■ | ■ | ■ | ● | ● | ● | ● | ■ | |
| 198 | *Coenonympha tullia* (Müller, 1764) | ● | | | | | | | | | | | | 48 |
| 199 | *Coenonympha glycerion* (Borkhausen, 1788) | | | | | ● | ● | ■ | | ■ | | | | |
| 200 | *Coenonympha arcania* (Linnaeus, 1760) | ● | ● | | ■ | ● | ■ | ■ | | ■ | ● | | ○ | |
| 201 | *Coenonympha leander* (Esper, 1784) | | | | | ● | | ■ | | ● | | ■ | | |
| 202* | *Coenonympha saadi* Kollar, 1849 | | | | | ● | | | | ● | | | | |
| 203* | *Coenonympha symphita* Lederer, 1870 | | ● | | ● | ● | ● | | | ■ | | | | |
| 204 | *Kirinia climene* (Esper, 1783) | ● | | | ○ | | ■ | ■ | | ■ | | | ■ | |
| 205 | *Pararge aegeria* (Linnaeus, 1758) | ■ | ■ | | ○ | ■ | ■ | ■ | ● | ■ | ● | | ■ | |
| 206 | *Lasiommata megera* (Linnaeus, 1767) | ● | ■ | | ■ | ■ | ■ | ■ | | ■ | ● | ● | ■ | |
| 207 | *Lasiommata petropolitana* (Fabricius, 1787) | | | | | ● | ■ | ○ | ● | ■ | | | | |
| 208 | *Lasiommata maera* (Linnaeus, 1758) | ● | | | ○ | ■ | ● | ■ | ● | ■ | ● | ● | ■ | |
| 209 | *Maniola jurtina* (Linnaeus, 1758) | ● | ■ | ● | ■ | ■ | ■ | ■ | ■ | ■ | ● | ● | ■ | |
| 210 | *Hyponephele lycaon* (Kühn, 1774) | ● | ■ | | ● | ■ | ● | ● | | ■ | ● | | ○ | |
| 211 | *Hyponephele lupinus* (Costa, 1836) | | | | | ■ | | ● | | | | | ■ | |
| 212* | *Hyponephele amardaea* (Lederer, 1869) | ● | | | | | | ● | | ● | ● | | | 49 |

(Continued)

| № | Species name | AB | AJ | GU | IM | KA | KK | MM | RL | SJ | SK | SZ | TB | [c] |
|---|---|---|---|---|---|---|---|---|---|---|---|---|---|---|
| 213* | *Hyponephele comara* (Lederer, [1870]) | | | | | | | | | ■ | | | | 50 |
| 214 | *Melanargia russiae* (Esper, 1783) | | ■ | | | ■ | ● | ■ | ● | ■ | ● | | ■ | |
| 215 | *Melanargia galathea* (Linnaeus, 1758) | ● | | | ■ | ■ | ■ | ■ | ● | ■ | ■ | ● | ■ | |
| 216* | *Melanargia teneates* (Ménétriés, 1832) | | | | | | ● | ● | | | | ○ | | 51 |
| 217 | *Melanargia larissa* (Geyer, 1828) | | | | | ■ | ● | ■ | | ■ | ● | | ■ | |
| 218* | *Melanargia hylata* (Ménétriés, 1832) | ○ | | | | | | | | | | | | 52 |
| 219 | *Brintesia circe* (Fabricius, 1775) | | | ● | ■ | ● | ■ | ■ | | ■ | | | ■ | |
| 220 | *Arethusana arethusa* (Denis & Schiffermüller, 1775) | | | | | ● | ○ | ○ | | ■ | ● | | ● | |
| 221 | *Chazara briseis* (Linnaeus, 1764) | ● | | | ○ | ■ | ■ | ■ | ■ | ■ | ● | ● | ■ | |
| 222 | *Chazara persephone* (Hübner, 1805) | | | | | ● | ○ | ○ | | ● | | | ○ | |
| 223 | *Pseudochazara alpina* (Staudinger, 1878) | | | ● | | ■ | ● | ○ | | ● | | | | |
| 224 | *Pseudochazara mniszechii* (Herrich-Schäffer, 1851) | | | | | | | ● | | ● | | ● | ● | |
| 225 | *Pseudochazara geyeri* (Herrich-Schaffer, 1846) | | | | | | ○ | | | ● | | | | |
| 226 | *Pseudochazara beroe* (Herrich-Schaffer, 1844) | | | | | | | | | ● | | | | |
| 227 | *Pseudochazara pelopea* (Klug, 1832) | ● | | ○ | | ● | ■ | ■ | | ● | | | ● | |
| 228 | *Satyrus iranica* Schwingenschuss, 1939 | | ● | | ○ | ○ | ● | ● | | ■ | ○ | | ○ | |
| 229 | *Satyrus amasina* Staudinger, 1861 | | | | | ● | | | | ● | | | | |
| 230 | *Minois dryas* (Scopoli, 1763) | ● | ■ | | ■ | ● | ■ | ■ | | ■ | | ■ | ■ | |
| 231 | *Hipparchia statilinus* (Hufnagel, 1766) | | | | ○ | ■ | ● | ○ | | ■ | | | ● | |
| 232 | *Hipparchia fatua* Freyer, 1843 | | | | ● | | | | | ■ | | | | |
| 233 | *Hipparchia pellucida* (Stauder, 1924) | ● | | | | ■ | ■ | ■ | | ■ | | ○ | ■ | |
| 234 | *Hipparchia syriaca* (Staudinger, 1871) | ● | ● | | ● | ■ | ● | ■ | | ■ | ● | | ■ | |
| 235 | *Hipparchia autonoe* (Esper, 1783) | ● | | | | | | | | | | | | |
| 236 | *Hipparchia parisatis* (Kollar, 1849) | | | | | ■ | | | | | | | | |
| 237 | *Erebia aethiops* (Esper, 1777) | ● | ■ | | ■ | ■ | ■ | ■ | ■ | ■ | ● | ■ | ○ | |
| 238 | *Erebia medusa* (Denis & Schiffermüller, 1775) | ● | ● | | ■ | ○ | ■ | ■ | | ■ | ○ | ● | | |
| 239 | *Erebia iranica* Grum-Grshimailo, 1895 | ● | ● | | ● | ● | ● | ● | | ■ | | ● | ○ | |
| 240* | *Erebia graucasica* Jachontov, 1909 | ● | ● | | ■ | ■ | ● | ● | | ■ | ○ | ● | | |
| 241* | *Erebia melancholica* Herrich-Schäffer, 1846 | ■ | ■ | ● | ○ | ■ | ■ | ■ | | ■ | ■ | | ■ | |
| 242* | *Erebia hewitsonii* Lederer, 1864 | | ■ | | ■ | | | | | ■ | | | | |
| 243 | *Proterebia afra* (Fabricius, 1787) | | | | | ■ | ● | ● | | | | ● | | |
| | **Libytheinae** Boisduval, 1833 | | | | | | | | | | | | | |
| 244 | *Libythea celtis* (Laicharting, 1782) | ● | | | | ■ | | | | ○ | | | | |

**Note:**
The presence of all species in each region is provided. Explanations of the abbreviations and symbols used are presented at the top of the table. The № column contains the species index. Optionally, it may include a Caucasus endemic symbol (*) or an asterisk (*) indicating that the occurrence of this species in Georgia is questionable. The numbers in column [c] refer to the comments presented in Table 3. If there are records from more than one period, the symbol for the most recent period is used in each region's column.

published in any article. Creating this list, in our opinion, was very important out of respect for the traditional native names. We extended this list by supplementing it with Russian names, because the Russian language was used in Georgia during the Soviet times, serving as a lingua franca and a language of inter-ethnic communication. Furthermore, since many young people use English, we also added English names. We hope that a list of
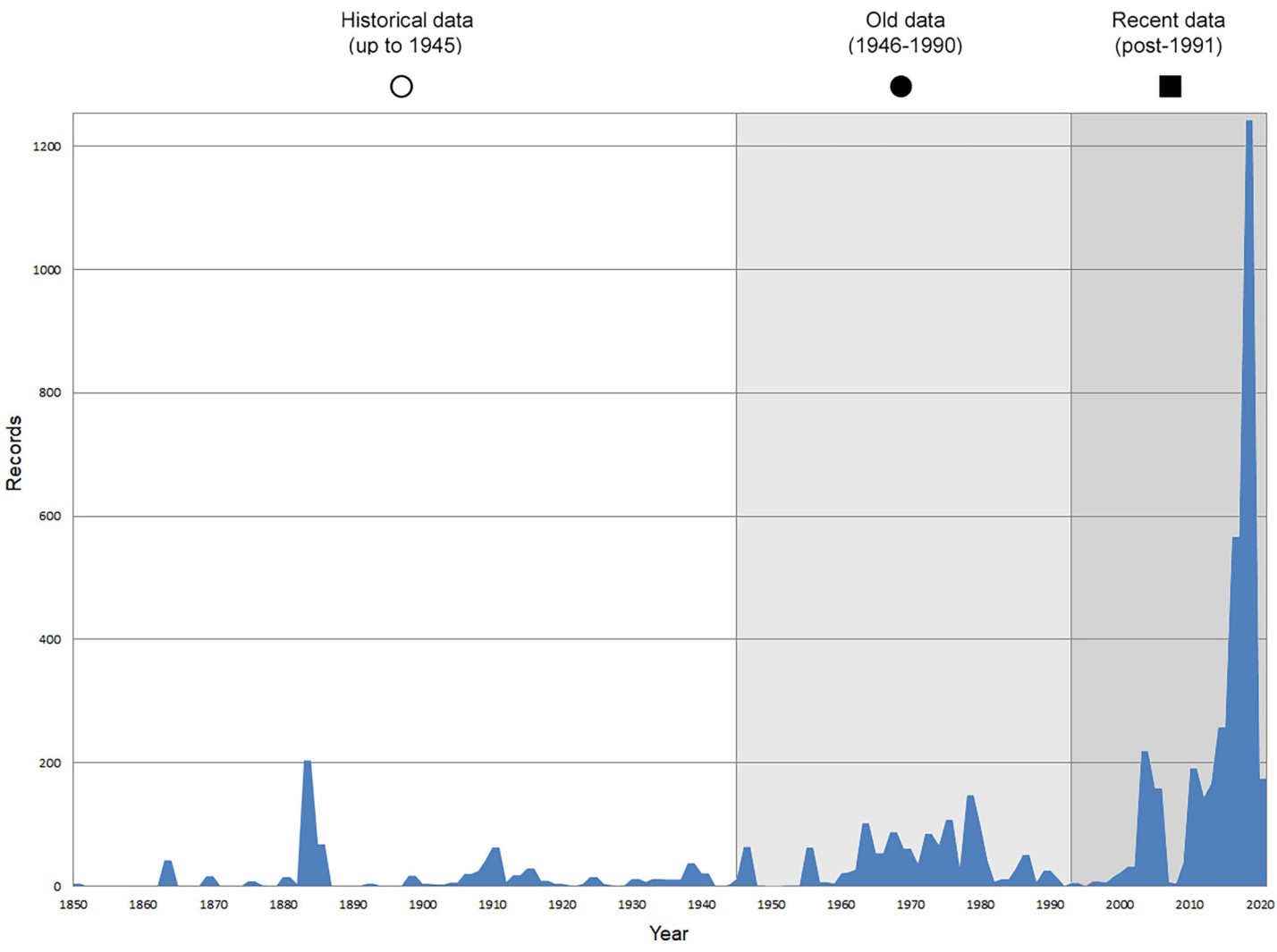

**Figure 2 Number of records collected in every year in three defined periods (see Materials and Methods).**

vernacular names provides useful guides to identify butterflies for the wider entomological community.

The distribution of numerous species is still poorly known because many regions of Georgia have not been explored extensively. Undoubtedly the best studied regions are: Samtskhe-Javakheti (1,713 records, 191 species) and Mtskheta-Mtianeti (1,245 records, 145 species). Some regions, such as K'akheti, Kvemo Kartli and Abkhazia are also well investigated (more than 460 records), with more than 130 species found in them. It is noteworthy that in Samegrelo-Zemo Svaneti and Ajaria, despite having more than 790 records, less than 100 species were noted. The smallest number of records and species were collected in Guria (only 44 records and 30 species) which is really surprising (Table 5, Fig. 3). A relatively large number of records collected from neighbouring regions and direct access to the Black Sea coast suggests that the real number of species in this region is probably much higher than presented.

**Table 3 Comments on new and doubtful butterfly species of Georgia.**

| # | № | Species name | Comments |
|---|---|---|---|
| 1 | 7. | *Spialia orbifer* (Hübner, 1823) | *Didmanidze (2004)* reported this species as *Spialia sertorius orbifer* (Hübner, 1823). According to the latest revisions of the *Spialia* genus (*Hernández-Roldán et al., 2016*), the *sertorius* species complex includes five species: *S. orbifer* (Hübner, 1823) distributed in Eastern Europe and Asia, the parapatric *S. sertorius* (Hoffmannsegg, 1804) occuring in Western Europe and north-western Africa, *S. ali* (Oberthür, 1881), *S. therapne* (Rambur, 1832) and *S. rosae* Hernández-Roldán, Dapporto, Dincă, Vicente & Vila, 2016. We consider that old literature records of *S. sertorius* for the Caucasus refer to *S. orbifer*. |
| 2 | 8. | *Favria cribrellum* (Eversmann, 1841) | Species reported by *Didmanidze (2004)*, based on only two records. *Tshikolovets & Nekrutenko (2012)* suggest that this was probably a misidentification, although the species is present in the North Caucasus and may also be found in the northern part of Georgia. Consequently, those records need verification. |
| 3 | 10*. | *Muschampia baeticus* (Rambur, 1839) | Species reported only by *Didmanidze (2004)*, not confirmed by other authors and not mentioned by *Tshikolovets & Nekrutenko (2012)*. The closest known localities for this species are in Turkey and Iran (*Kudrna et al., 2011*). Probably a misidentification. |
| 4 | 11. | *Muschampia poggei* (Lederer, 1858) | Species known so far only from areas south of the Lesser Caucasus such as Syria, Armenia, Azerbaijan or Iran (*Korb & Bolshakov, 2016*). Similar to *Tshikolovets & Nekrutenko (2012)*, we believe that records provided by *Didmanidze (1976, 2004)* from Georgia require confirmation. |
| 5 | 20. | *Pyrgus carthami* (Hübner, 1813) | Recorded by *Romanoff (1884)*, but it is probably a misidentification (*Tshikolovets & Nekrutenko, 2012*). *Didmanidze (2004)* reported this species as *Pyrgus fritillarius* Poda, 1761 considered as a synonym of *P. carthami*. The closest confirmed localities for this species are in the extreme north western part of the Great Caucasus (*Tshikolovets & Nekrutenko, 2012*). Consequently, all records from Georgia require verification. |
| 6 | 21. | *Carterocephalus palaemon* (Pallas, 1771) | Species known from very old data and only one location: Borjomi (*Romanoff, 1884*). It has never been recorded again in Georgia. It is impossible to confuse this species with any other, so it is likely to be extinct. |
| 7 | 36. | *Papilio alexanor* Esper, 1800 | Species not previously reported in Georgia. A single specimen has been found close to David Gareja monastery in K'akheti (*Viter, 2022*). |
| 8 | 38. 39. | *Leptidea juvernica* Williams, 1946 *Leptidea sinapis* (Linnaeus, 1758) | After the molecular studies on wood white *Leptidea* butterflies, a new species has been described (*Dincă et al., 2011*). The oldest known name assigned to the new species is *juvernica* (Williams, 1946), described as a subspecies of Irish populations of *L. reali* (*Mazel, 2001*). In the territory of Georgia, both species may occur: previously described *L. sinapis* (Linnaeus, 1758) and new species, *L. juvernica* Williams, 1946. It is impossible to distinguish which old records of *L. sinapis* refer to *L. sinapis* or *L. juvernica*. |
| 9 | 48. | *Pieris mannii* (Mayer, 1851) | Species reported only by *Didmanidze (2004)* and not included by *Tshikolovets & Nekrutenko (2012)* so it requires confirmation. The closest known localities for this species are in the Republic of Dagestan (Russia, North Caucasus) (*Mamedova, 2013*). |
| 10 | 53. | *Pieris bowdeni* Eitschberger, 1984 | Known only from a single specimen from the Zoological Museum in Kyiv; requires confirmation. The closest known localities for this species are in Armenia (*Tshikolovets & Nekrutenko, 2012*). |
| 11 | 55. | *Pontia edusa* (Fabricius, 1777) | *P. edusa* (Fabricius, 1777) has been separated from its sibling species *P. daplidice* (Linnaeus, 1758) based on enzyme electrophoretic (*Geiger & Scholl, 1982*; *Geiger, Descimon & Scholl, 1988*; *Wagener, 1988*) and mtDNA studies (*John et al., 2013*). It is generally known that *P daplidice* is recorded mainly in Western Europe, North Africa and Cyprus, while *P. edusa* occurs in the Italian Peninsula, Sicily and Eastern Europe (*Dennis et al., 2008*; *John et al., 2013*; *Vodă et al., 2016*). There were no genetic studies on these species in the Caucasus, but *Anikin, Sachkov & Zolotuhin (1993)* suggested that *P. edusa* inhabits the Volga River valley, Crimea, Middle Asia and the Caucasus, while *P. daplidice* occurs to the north and east of the Rovno-Kiev-Kaluga-N. Novgorod-Chelyabinsk line till Primorye. Based on the distribution maps given in those publications, we assume that all old literature records of *P. daplidice* from Georgia should be assigned to *P. edusa*, but this requires corroboration. |

| # | № | Species name | Comments |
|---|---|---|---|
| 12 | 59. | *Colias chrysotheme* (Esper, 1781) | A relatively large number of records from Georgia collected by Didmanidze are probably incorrect or doubtful and require confirmation due to the possibility of confusion of this species with other species of the genus *Colias*. Moreover, *Didmanidze (2004)* reported this species near Batumi, referring to the publication by *Filipjev (1916)*. In the indicated article, only *Colias edusa* F. [sic!] is listed in this area, and there are no data about the species *C. chrysotheme*. *C. edusa* is a synonym of *C. croceus* (Geoffroy in A.F. Fourcroy, 1785). The closest confirmed localities for *C. chrysotheme* are on the northern slopes of the Great Caucasus (*Tshikolovets & Nekrutenko, 2012*). Due to these ambiguities, all records from Georgia except Abkhazia require verification, because the material is probably misidentified. |
| 13 | 63. | *Colias alfacariensis* Ribbe, 1905 | Currently, *C. alfacariensis* and *C. hyale* (Linnaeus, 1758) are treated as sister species (see *Brunton, 1998*; *Cleary, Descimon & Menken, 2002*). It is almost impossible to make a proper identification based on collected adults due to the large variation in wing patterns and other morphological features. *C. hyale* was reported from the Caucasus region by several different authors, including by *Didmanidze (2004)*, but following the publications by *Brunton (1998)* and *Cleary, Descimon & Menken (2002)*, all those records should be regarded as *C. alfacariensis*. |
| 14 | 72. | *Lycaena candens* (Herrich-Schäffer, 1844) | *Beuret (1952)* discovered that *Lycaena hippothoe* (Linnaeus, 1761) could be divided into two different species: *L. hippothoe* occurring from East Asia to Spain and *L. candens* inhabiting a small region from Iran to the Balkans. Based on the distribution maps, all records originally provided by *Didmanidze (2004)* as *L. hippothoe* in our work have been assigned to *L. candens* (*Beuret, 1954*). |
| 15 | 74. | *Lycaena thetis* Klug, 1834 | *Didmanidze (2004)* reported this species in Vashlovani National Park (K'akheti). The closest known localities for this species are in Southern Armenia and Southern Azerbaijan (*Snegovaya & Petrov, 2019*). Similar to *Tshikolovets & Nekrutenko (2012)*, we believe that those records are misidentified. |
| 16 | 75. | *Lycaena japhetica* (Nekrutenko & Effendi, 1983) | For decades, *Lycaena (Athamanthia) japhetica* was treated as a species from the Caspian area of Eastern Azerbaijan (*Tuzov et al., 2000*). In 2017, the species was found more than 400 km from known populations, in the vicinity of Kaspi town in Georgia (*Morgun, 2019*). |
| 17 | 80*. | *Tarucus theophrastus* (Fabricius, 1793) | Distributed in the Afrotropical Realm (*Deghiche-Diab & Deghiche, 2021*). All records for this species reported by *Didmanidze (2004)* are doubtful and require verification. |
| 18 | 87. | *Pseudophilotes vicrama* (Moore, 1865) | *Didmanidze (2004)* reported two species from the genus *Pseudophilotes*: *P. baton* (Bergsträsser, 1779) and *P. vicrama*. Unfortunately, *P. baton* is not present in the Caucasus and Transcaucasia. Individuals collected by Didmanidze probably belong to *P. vicrama* or *P. bavius* (Eversmann, 1832). The closest known localities for *P. bavius* are in the Northern Caucasus (Russia) and in Armenia (*Tshikolovets & Nekrutenko, 2012*), but the species has not been reported in Georgia so far. In agreement with *Tshikolovets & Nekrutenko (2012)*, we believe that the records from Georgia require verification. |
| 19 | 88. | *Scolitanides orion* (Pallas, 1771) | Reported by *Didmanidze (2004)* from only one locality in Georgia (Khelvachauri, close to Batumi) but not confirmed by any other authors. A very distinctive species, impossible to confuse with any other. Species reported from Artvin Province in Turkey, close to Batumi (*Karaçetin & Welch, 2011*). Record requires confirmation. |
| 20 | 98. | *Plebejus christophi* (Staudinger, 1874) | Known only from two localities in Georgia: Aspindza and Khando (*Didmanidze, 1979*). The closest confirmed localities for this species are in Southern Armenia and Azerbaijan. Records from Georgia require confirmation (*Tshikolovets & Nekrutenko, 2012*). |
| 21 | 99. 100. 101. | *Kretania modica* (Verity, 1935) *Kretania sephirus* (Frivaldszky, 1835) *Kretania stekolnikovi* (Stradomsky & Tikhonov, 2015) | After a molecular study on the *Kretania pylaon* species group, three distinct species have been separated: *K. sephirus* (Frivaldszky, 1835), *K. modica* Vérity, 1935 and *K. stekolnikovi* Stradomsky & Tikhonov, 2015 (*Stradomsky & Tikhonov, 2015*). All records originally assigned by *Didmanidze (2004)* to *K. sephirus* have been divided in our work into those three species based on the distribution maps presented in the above-mentioned publication. |
| 22 | 110*. | *Agriades optilete* (Knoch, 1781) | Records provided by *Didmanidze (2004)* are doubtful and require verification. Based on available literature, this species has not been observed in the Caucasus. The closest known localities for that species are in Eastern Ukraine, the Volga region in Russia and Kazakhstan (*Korb & Bolshakov, 2011b*, *2016*). |

(Continued)
| # | № | Species name | Comments |
|---|---|---|---|
| | | **Table 3** (continued) | |
| 23 | 111. | *Agriades dardanus* (Freyer, 1843) | Following *Higgins & Riley (1970)*, *Didmanidze (2004)* reported this species as *Agriades glandon dardanus* Freyer, 1944. According to *Nekrutenko (1974)* *A. glandon dardanus* should be considered as a subspecies of *A. pyrenaicus*. Unfortunately, this publication was omitted by Didmanidze. Taking the phylogenetic study by *Talavera et al. (2013)* into account, all recorded specimens should belong to *A. pyrenaicus dardanus*. Moreover, the aforementioned subspecies was elevated to species level (*Wiemers et al., 2018*). |
| 24 | 113. | *Lysandra corydonius* (Herrich-Schäffer, 1852) | *Didmanidze (2004)* reported this species as *Lysandra corydon* (Poda, 1761) following the publication by *Jachontov (1914)*, where it was reported as *L. corydon ciscaucasica*. The subspecies has been synonymised with *Polyommatus corydonius ciscaucasicus* by *Schurian (1988)*. *P. corydon* has not been reported in Georgia (*Tshikolovets & Nekrutenko, 2012*). The closest known locality for *Polyommatus (Lysandra) corydon* is on the extreme northern slopes of the Great Caucasus (*Lvovsky & Morgun, 2007*). |
| 25 | 115*. | *Polyommatus escheri* (Hübner, 1823) | The species distribution extends from Southern Europe to Northern Africa. The closest known locality for *P. escheri* is the European part of Turkey (*Karaçetin & Welch, 2011*). Two records given by *Didmanidze (2004)* are doubtful and require verification. |
| 26 | 119. | *Polyommatus icarus* (Rottemburg, 1775) | *Didmanidze (2004)* reported *P. icarus* and *P. persica* Bienert, 1870 as two separate species. Curently *P. persica* is a subspecies of *P. icarus* (*Vodolazhsky & Stradomsky, 2008*) and all old records of *P. persica* have been assigned to *P. icarus*. |
| 27 | 122. | *Polyommatus admetus* (Esper, 1783) | Only two known historical records from Georgia, which were not included by *Tshikolovets & Nekrutenko (2012)*. The closest known localities for this species are in Southern Armenia and Southern Azerbaijan (*Tshikolovets & Nekrutenko, 2012*). It is highly likely that either that the specimens were misidentified or this species no longer occurs in Georgia. |
| 28 | 126. | *Polyommatus wagneri* (Forster, 1956) | *Didmanidze (2004)* reported this species as *Polyommatus damone wagneri* (Forster, 1956). Meanwhile, the subspecies was elevated to species level (*Lukhtanov et al., 2023*). |
| 29 | 127. | *Polyommatus altivagans* (Forster, 1956) | A few records from K'akheti in Georgia (*Didmanidze, 2004*), which were not included by *Tshikolovets & Nekrutenko (2012)*. A relatively large number of records from Armenia, Azerbaijan and Dagestan (*Tshikolovets & Nekrutenko, 2012*) may suggest that this species could also be found in other regions of Georgia. |
| 30 | 128. | *Polyommatus cyanea* (Staudinger, 1899) | The taxonomic position of this species (and subspecies) has changed several times since the studies on the group which were conducted mainly by *Forster (1956, 1960, 1961)*, which described new subspecies within the species *Agrodiaetus xerxes*. Based on the publication by *Forster (1956)*, *Didmanidze (2004)* reported this species as *Polyommatus xerxes pseudocyanea*. Decades later, the species "*xerxes*" has been considered as subspecies of *cyanea* within damon-group (*Eckweiler & Häuser, 1997*). |
| 31 | 129. | *Polyommatus poseidon* (Herrich-Schäffer, 1851) | The presence of this species in Georgia has been proven based on a new method using chromosomal and molecular markers (see *Lukhtanov & Tikhonov, 2005*). |
| 32 | 132. | *Polyommatus aserbeidschana* (Forster, 1956) | Initially, it was suspected that this species only occurs in Azerbaijan but, in recent times, it has been confirmed in Iran (*Eckweiler & Häuser, 1997*), Turkey (*Karaçetin & Welch, 2011*) and Armenia (*Lukhtanov & Dantchenko, 2017*). Recently, the species has also been found in Georgia in the Trialetsky Ridge (situated within the Samtskhe-Javakheti region) and the Meskhetsky Ridge (located in the Lesser Caucasus) (*Dantchenko, 2000*). |
| 33 | 133. | *Polyommatus ninae* (Forster, 1956) | The closest known localities for this species are in Southern Armenia and Southern Azerbaijan (*Tshikolovets & Nekrutenko, 2012*). One record from Tzarskije Kolodtzy (currently known as Dedoplistskaro) in the vicinity of Tbilisi (*Forster, 1956*) requires verification, but it is probably a misidentification. |
| 34 | 135. | *Afarsia morgiana* (Kirby, 1871) | *Didmanidze (2004)* reported this species as *Vaccinia hyrcana* Lederer, 1869 following *Lewis (1974)* nomenclature. Two records from Tbilisi and Shulaveri require confirmation. The closest known localities for this species are in Southern Armenia and Southern Azerbaijan (*Snegovaya & Petrov, 2019*). |
| 35 | 138. | *Satyrium abdominalis* (Gerhard, 1850) | *Didmanidze (2004)* reported this species as *Satyrium acaciae abdominalis* (Gerhard, [1850]). The status of this taxon is ambiguous. Some lepidopterologists consider it as a bona species, *S. abdominalis*, while others regard it as a subspecies of *S. acaciae*. Similar to *Rajaei et al. (2023)* in this article, we support the position of this taxon as a valid species. |
| 36 | 143*. | *Satyrium lunulata* (Erschoff, 1874) | Single record from Kakheti (*Didmanidze, 1979*) which is doubtful and requires verification. The closest known localities for this species are in Iran, Tajikistan and Uzbekistan (*Tuzov et al., 2000*). |

| # | № | Species name | Comments |
|---|---|---|---|
| 37 | 144. | *Callophrys mystaphia* Miller, 1913 | Single record from Vashlovani National Park (K'akheti) (*Didmanidze, 2004*) which requires verification. The closest known localities for this species are in Azerbaijan, Iran and Turkey (*Krupitsky & Kolesnichenko, 2013*). |
| 38 | 145. | *Callophrys chalybeitincta* Sovinsky, 1905 | *Didmanidze (2004)* reported this species as *Callophrys rubi chalybeitincta* Sovinsky, 1905. The subspecies was elevated to species level (*Stradomsky & Vodolazhsky, 2011*). *C. rubi* (Linnaeus, 1758) is not present in Georgia. |
| 39 | 154. | *Boloria caucasica* (Lederer, 1852) | *Didmanidze (2004)* reported this species as *Boloria pales caucasica* Staudinger, 1861. The subspecies was elevated to species level (*Simonsen et al., 2010*). |
| 40 | 155. | *Boloria eunomia* (Esper, 1800) | Single historical record from Akhaltsikhe (Samtskhe-Javakheti) (*Bohatsch, 1886*) which requires confirmation. |
| 41 | 158. | *Boloria selene* (Denis & Schiffermüller, 1775) | Species known from a large number of records listed only by *Didmanidze (2004)*, described as a common species but not confirmed by any other authors. All records and specimens of this species require verification. The closest known localities for *Boloria selene* are in Armenia (*Tshikolovets & Nekrutenko, 2012*). |
| 42 | 169. | *Thaleropis ionia* (Fischer de Waldheim & Eversmann, 1851) | Species not previously reported from Georgia. The single specimen was found close to Kvariati village in Ajaria (*Global Biodiversity Information Facility, 2022*). |
| 43 | 175. | *Melitaea interrupta* Kolenati, 1846 | *Didmanidze (2004)* reported this species as *Melitaea transcaucasica* Turati, 1919. In a few other publications based on old data recorded by Didmanidze from the Caucasus, this species is still presented as *M. transcaucasica*, but currently it is known as a synonym of *M. interrupta* (*Kemal & Koçak, 2013*). |
| 44 | 177*. | *Melitaea casta* (Kollar, 1848) | Endemic species of Iran (*Rajaei et al., 2023*). The only one record listed by *Didmanidze (2004)* is doubtful and requires confirmation. |
| 45 | 182*. | *Melitaea alatauica* Staudinger, 1881 | Endemic species of the Dzungarian Alatau in south-eastern Kazakhstan (*Zhdanko, 2005*; *Lukhtanov, Gagarina & Pazhenkova, 2021*). Records listed by *Didmanidze (2004)* are not confirmed in any other available resources; probably a misidentification. |
| 46 | 185. | *Euphydryas discordia* Bolshakov & Korb, 2013 | During the study on females from the nominative subspecies of *Euphydryas aurinia* (Rottemburg, 1775) from many regions of the Caucasus, a new species to science, *E. discordia*, was described. It is a cryptic species of *E. aurinia* group (*Bolshakov & Korb, 2013*). |
| 47 | 186. | *Euphydryas orientalis* (Herrich-Schäffer, 1851) | Only one record from Zekari Pass (Imereti) reported by *Didmanidze (2004)*; it requires verification. The closest known localities for this species are in Central Turkey (*Korb & Bolshakov, 2011a*). |
| 48 | 199. | *Coenonympha tullia* Müller, 1764 | Records from Bakuriani and Tskhra-Tskaro reported by *Didmanidze (2004)* are misidentified (*Tshikolovets & Nekrutenko, 2012*). The mentioned authors provided a relatively large number of records from Karachay-Cherkessia in Russia. We suspect that this species may occur in Abkhazia. |
| 49 | 213*. | *Hyponephele amardaea* (Lederer, 1869) | The closest known localities for this species are in Iran and Turkmenistan (*Tuzov et al., 1997*). Records listed by *Didmanidze (2004)* are not confirmed in any other available resources. Probably a misidentification. |
| 50 | 214*. | *Hyponephele comara* (Lederer, [1870]) | The closest known localities for this species are in Iran and Southern Azerbaijan (*Tshikolovets & Nekrutenko, 2012*). There is only one record from Khando (*Didmanidze, 2004*); it requires confirmation. Similar to *Tshikolovets & Nekrutenko (2012)*, we believe this is a misidentification. |
| 51 | 217*. | *Melanargia teneates* (Ménétriés, 1832) | *Didmanidze (2004)* reported this species from Svaneti referring to the publication by *Romanoff (1884)*. In fact, Romanoff gives locality name as "Souant" (arid part of the Talysh Mts., Azerbaijan). It can be assumed that Didmanidze considered that "Souant" means "Svaneti" (in Georgia) or the name was mistyped. Unfortunately, Romanoff made a typographical error in the locality name while referring to a publication by *Ménétriés (1832)*. Ménétriés has provided the correct name for the locality as "Zouvant" (region in the Lesser Caucasus at the Iranian border close to the district town of Lerik). Moreover, Didmanidze provided two own records from Georgia (Svaneti) that have been deemed misidentifications (*Tshikolovets & Nekrutenko, 2012*). |
| 52 | 219*. | *Melanargia hylata* (Ménétriés, 1832) | Single erroneous record from Sukhumi in Abkhazia (*Didmanidze, 2004*). The author listed this species citing the publication by *Romanoff (1884)*. In fact, Romanoff gives locality name as "Schakouh" in the Talysh Mountains (Azerbaijan/Iran). Didmanidze's indication of a locality for this species as "Sukhumi" (in Abkhazia) is an error. A locality for this species in the Talysh Mountains was subsequently confirmed by *Christoph (1886)*. |

**Note:**
(#) Comment number used in column [c] in Table 2. (№) Species index used in Table 2.

**Table 4 List of Georgian butterflies with the presence status in each region and vernacular names index.**

| Hesperiidae, Latreille, 1809 | | Pyrginae, Burmeister, 1878 |
|---|---|---|

**1. *Erynnis tages* (Linnaeus, 1758)**

| | |
|---|---|
| **English name:** | Dingy Skipper |
| **Georgian name:** | მოშავო თავმსხვილა |
| **Russian name:** | Бурокрылка Тагес |
| **Distr. Caucasus:** | Widespread. |
| **Distr. Georgia:** | Widespread and common. |
| **Flight:** | Bivoltine in the lowlands, April to August. Univoltine in the mountains, late May to July. |
| **Habitat:** | Various open, grassy habitats. |

**2. *Erynnis marloyi* (Boisduval, 1834)**

| | |
|---|---|
| **English name:** | Inky Skipper |
| **Georgian name:** | |
| **Russian name:** | Бурокрылка Марлоя |
| **Distr. Caucasus:** | Central and eastern part of the Great Caucasus, Lesser Caucasus. |
| **Distr. Georgia:** | Local and rare. |
| **Flight:** | Uni- or bivoltine, April to October depending on the altitude. |
| **Habitat:** | Dry, stony and clayey places, rocky gorges and slopes. |

**3. *Carcharodus alceae* (Esper, 1780)**

| | |
|---|---|
| **English name:** | Mallow Skipper |
| **Georgian name:** | ბალბის თავმსხვილა |
| **Russian name:** | Зубчатокрылка альцея |
| **Distr. Caucasus:** | Widespread. |
| **Distr. Georgia:** | Local and rare. |
| **Flight:** | Multivoltine in the lowlands, April to October. Univoltine in the mountains. |
| **Habitat:** | Meadows, steppes, grassy slopes, woodland clearings. |

**4. *Carcharodus lavatherae* (Esper, 1783)**

| | |
|---|---|
| **English name:** | Marbled Skipper |
| **Georgian name:** | |
| **Russian name:** | Толстоголовка чистецовая |
| **Distr. Caucasus:** | Widespread. |
| **Distr. Georgia:** | Local, extremely rare. |
| **Flight:** | Univoltine, June to August. |
| **Habitat:** | Dry grassland with rocky calcareous places, dry slopes. |

**5. *Carcharodus floccifera* (Zeller, 1847)**

| | |
|---|---|
| **English name:** | Tufted Marbled Skipper |
| **Georgian name:** | |
| **Russian name:** | Зубчатокрылка шандровая |
| **Distr. Caucasus:** | Widespread. |
| **Distr. Georgia:** | Widespread but not frequent. |
| **Flight:** | Bivoltine in the lowlands, May to August. Univoltine in the high mountains, June to August. |
| **Habitat:** | Meadows, grassy slopes, woodland clearings. |

| | | |
|---|---|---|
| 6. *Carcharodus orientalis* Reverdin, 1913 | **English name:** | Oriental Marbled Skipper |
| | **Georgian name:** | |
| | **Russian name:** | Толстоголовка восточная |
| | **Distr. Caucasus:** | Widespread, except Colchis and Kura lowlands. |
| | **Distr. Georgia:** | Local and not common. |
| | **Flight:** | Uni- or bivoltine, late April to late September depending on the altitude. |
| | **Habitat:** | Steppes, meadows, stony and calcareous hills. |
| 7. *Spialia orbifer* (Hübner, 1823) | **English name:** | Hungarian skipper |
| | **Georgian name:** | ასკილის თავმსხვილა |
| | **Russian name:** | Спиалия круглопятнистая |
| | **Distr. Caucasus:** | Widespread. |
| | **Distr. Georgia:** | Widespread and common. |
| | **Flight:** | Uni- or bivoltine, late May to August. |
| | **Habitat:** | Grassy slopes, dry meadows, steppes, woodland clearings. |
| 8. *Favria cribrellum* (Eversmann, 1841) | **English name:** | Spinose Skipper |
| | **Georgian name:** | |
| | **Russian name:** | Мушампия решетчатая |
| | **Distr. Caucasus:** | Northern slopes of the Great Caucasus. |
| | **Distr. Georgia:** | Single records from central Georgia require verification. |
| | **Flight:** | Univoltine, May to July. |
| | **Habitat:** | Dry meadows, steppes, grassy places in river valleys, rocky slopes and screes. |
| 9. *Muschampia tessellum* (Hübner, 1803) | **English name:** | Tessellated Skipper |
| | **Georgian name:** | |
| | **Russian name:** | Мушампия большая |
| | **Distr. Caucasus:** | Widespread. |
| | **Distr. Georgia:** | Single records only, extremely rare. |
| | **Flight:** | Univoltine, June to August depending on the altitude. |
| | **Habitat:** | Steppe slopes, dry meadows, woodland clearings and glades. |
| 10*. *Muschaampia baeticus* (Rambur, 1839) | **English name:** | Southern Marbled Skipper |
| | **Georgian name:** | |
| | **Russian name:** | |
| | **Distr. Caucasus:** | Unknown. |
| | **Distr. Georgia:** | |
| | **Flight:** | Multivoltine, May to October. |
| | **Habitat:** | Rocky slopes, dry grasslands. |

| 11. *Muschampia poggei* (Lederer, 1858) | | |
|---|---|---|
| **English name:** | Pogges Skipper | |
| **Georgian name:** | | |
| **Russian name:** | Толстоголовка Погге | |
| **Distr. Caucasus:** | Djavakheti-Armenian plateau. | |
| **Distr. Georgia:** | Single records from central Georgia require verification. | |
| **Flight:** | Univoltine, June to July. | |
| **Habitat:** | Steppes, woodland clearings, xerophytic slopes. | |

| 12. *Pyrgus melotis* (Duponchel, 1834) | | |
|---|---|---|
| **English name:** | Agean Skipper | |
| **Georgian name:** | | |
| **Russian name:** | Темнокрылка мелотис | |
| **Distr. Caucasus:** | Widespread in mountains. | |
| **Distr. Georgia:** | Widespread but not common. | |
| **Flight:** | Uni- or multivoltine, April to September depending on the altitude. | |
| **Habitat:** | Mountain meadows and steppes. | |

| 13. *Pyrgus serratulae* (Rambur, 1839) | | |
|---|---|---|
| **English name:** | Olive Skipper | |
| **Georgian name:** | | |
| **Russian name:** | Темнокрылка серпуховая | |
| **Distr. Caucasus:** | Great Caucasus. | |
| **Distr. Georgia:** | Local and rare. | |
| **Flight:** | Univoltine, May to August. | |
| **Habitat:** | Open, grassy places in woodland, dry meadows, steppes. | |

| 14. *Pyrgus armoricanus* (Oberthür, 1910) | | |
|---|---|---|
| **English name:** | Oberthür's Grizzled Skipper | |
| **Georgian name:** | | |
| **Russian name:** | Толстоголовка арморикская | |
| **Distr. Caucasus:** | Great and Lesser Caucasus. | |
| **Distr. Georgia:** | Local and not frequent. | |
| **Flight:** | Bivoltine in the lowlands, May to August. Univoltine in the mountains, June to August. | |
| **Habitat:** | Dry stony and calcareous hills, rocky gullies. | |

| 15. *Pyrgus alveus* Hübner, 1803 | | |
|---|---|---|
| **English name:** | Large Grizzled Skipper | |
| **Georgian name:** | | |
| **Russian name:** | Темнокрылка белопятнистая | |
| **Distr. Caucasus:** | Widespread except deserts and semi-deserts. | |
| **Distr. Georgia:** | Widespread and frequent. | |
| **Flight:** | Univoltine, May to August. | |
| **Habitat:** | Open, grassy hills, steppes, woodland clearings. | |

| | | |
|---|---|---|
| **16\*. *Pyrgus jupei* (Alberti, 1967)** | **English name:** | Caucasian Skipper |
| | **Georgian name:** | |
| | **Russian name:** | Темнокрылка Юпе |
| | **Distr. Caucasus:** | Widespread in the Great Caucasus, local in the Lesser Caucasus. |
| | **Distr. Georgia:** | Local, not common. |
| | **Flight:** | Univoltine, late June to August. |
| | **Habitat:** | Subalpine meadows, damp steppes, forest clearings. |
| **17. *Pyrgus sidae* (Esper, 1784)** | **English name:** | Yellow-banded Skipper |
| | **Georgian name:** | |
| | **Russian name:** | Темнокрылка Сида |
| | **Distr. Caucasus:** | Widespread. |
| | **Distr. Georgia:** | South-eastern regions only, not frequent. |
| | **Flight:** | Univoltine, mid May to August. |
| | **Habitat:** | Dry scrub, slopes with steppe vegetation, subalpine meadows. |
| **18. *Pyrgus cacaliae* (Rambur, 1839)** | **English name:** | Dusky Grizzled Skipper |
| | **Georgian name:** | |
| | **Russian name:** | |
| | **Distr. Caucasus:** | Western part of the Great Caucasus. |
| | **Distr. Georgia:** | Abkhazia only. |
| | **Flight:** | Local and not frequent. |
| | **Habitat:** | Slopes with short alpine vegetation, meadows near streams and small rivers, low scrub. |
| **19. *Pyrgus cinarae* (Rambur, 1839)** | **English name:** | Sandy Grizzled Skipper |
| | **Georgian name:** | |
| | **Russian name:** | Темнокрылка артишоковая |
| | **Distr. Caucasus:** | Central and eastern part of the Great Caucasus, Lesser Caucasus. |
| | **Distr. Georgia:** | Local and very rare. |
| | **Flight:** | Univoltine, mid-June to August. |
| | **Habitat:** | Open grassy hills and clearings in woodland, virgin steppes with rocky and calcareous places. |
| **20. *Pyrgus carthami* (Hübner, 1813)** | **English name:** | Safflower Skipper |
| | **Georgian name:** | |
| | **Russian name:** | Темнокрылка сероватая |
| | **Distr. Caucasus:** | Extreme north-western part of the Great Caucasus. |
| | **Distr. Georgia:** | Records from the Lesser Caucasus requires verification. |
| | **Flight:** | Univoltine, mid-May to September depending on the altitude. |
| | **Habitat:** | Open grassy places, rocky and calcareous gullies and hills. |

*(Continued)*

| Hesperiidae, Latreille, 1809 | Heteropterinae, Aurivillius, 1925 | |
|---|---|---|
| **21. *Carterocephalus palaemon*** (Pallas, 1771) | **English name:** | Chequered Skipper |
| | **Georgian name:** | |
| | **Russian name:** | Крепкоголовка Палемон |
| | **Distr. Caucasus:** | Great Caucasus, western part of the Lesser Caucasus. |
| | **Distr. Georgia:** | Single historical record from Borjomi. |
| | **Flight:** | Univoltine, May to August depending on the altitude. |
| | **Habitat:** | Damp meadows, river valleys, woodland clearings. |
| **22. *Heteropterus morpheus*** (Pallas, 1771) | **English name:** | Large Chequered Skipper |
| | **Georgian name:** | |
| | **Russian name:** | Разнокрылка Морфей |
| | **Distr. Caucasus:** | Great Caucasus and western part of the Lesser Caucasus. |
| | **Distr. Georgia:** | Western regions, locally common. |
| | **Flight:** | Univoltine, May to August. |
| | **Habitat:** | Damp meadows, woodland clearings with swamps. |
| Hesperiidae, Latreille, 1809 | Hesperiinae, Latreille, 1809 | |
| **23. *Eogenes alcides*** (Herrich-Schäffer, 1852) | **English name:** | Alcides Skipper |
| | **Georgian name:** | |
| | **Russian name:** | Толстоголовка алкид |
| | **Distr. Caucasus:** | Djavakheti-Armenian plateau. |
| | **Distr. Georgia:** | Vashlovani National Park (K'akheti). |
| | **Flight:** | Uni- or bivoltine, late May to September. |
| | **Habitat:** | Arid foothills, semi-deserts, gorges with xerophytic vegetation, sometimes dry steppes. |
| **24. *Gegenes nostrodamus*** (Fabricius, 1793) | **English name:** | Mediterranean Skipper |
| | **Georgian name:** | |
| | **Russian name:** | Толстоголовка Нострадам |
| | **Distr. Caucasus:** | Kura, Samur and Arax river valleys with adjoining foothills. |
| | **Distr. Georgia:** | K'akheti and Mtskheta-Mtianeti, extremely rare. |
| | **Flight:** | Multivoltine, May to October. |
| | **Habitat:** | Places with xerophytic vegetation, rocky gorges, semi-deserts. |
| **25. *Thymelicus hyrax*** (Lederer, 1861) | **English name:** | Levantine Skipper |
| | **Georgian name:** | |
| | **Russian name:** | Бронзовокрылка иракская |
| | **Distr. Caucasus:** | Extreme north-western part of the Great Caucasus, Lesser Caucasus. |
| | **Distr. Georgia:** | Kvemo Kartli. |
| | **Flight:** | Univoltine, May to July. |
| | **Habitat:** | Grassy slopes. |

| 26. *Thymelicus sylvestris* (Poda, 1761) | | |
|---|---|---|
| | **English name:** | Small Skipper |
| | **Georgian name:** | ტყის მსხვილთავა |
| | **Russian name:** | Бронзовокрылка лесная |
| | **Distr. Caucasus:** | Great and Lesser Caucasus. |
| | **Distr. Georgia:** | Widespread and common. |
| | **Flight:** | Univoltine, May to September. |
| | **Habitat:** | Woodland clearings and glades, steppes, meadows, sometimes cultivated areas. |

| 27. *Thymelicus lineola* (Ochsenheimer, 1808) | | |
|---|---|---|
| | **English name:** | Essex Skipper |
| | **Georgian name:** | მურა პატარა მსხვლთავა |
| | **Russian name:** | Бронзовокрылка тире |
| | **Distr. Caucasus:** | Widespread. |
| | **Distr. Georgia:** | Widespread and common. |
| | **Flight:** | Univoltine, early May to late August. |
| | **Habitat:** | Various grassy habitats. |

| 28. *Ochlodes sylvanus* (Esper, 1777) | | |
|---|---|---|
| | **English name:** | Large Skipper |
| | **Georgian name:** | პეპელა |
| | **Russian name:** | Толстоголовка лесовик |
| | **Distr. Caucasus:** | Widespread. |
| | **Distr. Georgia:** | Widespread and common. |
| | **Flight:** | Uni- or bivoltine in the lowlands, May to August. |
| | **Habitat:** | Dry meadows, slopes, woodland clearings. |

| 29. *Hesperia comma* (Linnaeus, 1758) | | |
|---|---|---|
| | **English name:** | Silver-spotted Skipper |
| | **Georgian name:** | თავმსხვილა კომა |
| | **Russian name:** | Толстоголовка запятая |
| | **Distr. Caucasus:** | Great and Lesser Caucasus. |
| | **Distr. Georgia:** | Widespread in the mountains, common. |
| | **Flight:** | Univoltine, late June to August depending on the altitude. |
| | **Habitat:** | Steppes, hills with xerophytic vegetation, sometimes woodland clearings. |

| **Papionidae**, Latreille, 1802 | | **Parnassiinae**, Duponchel, 1835 |
|---|---|---|

| 30. *Parnassius mnemosyne* (Linnaeus, 1758) | | |
|---|---|---|
| | **English name:** | Clouded Apollo |
| | **Georgian name:** | შავი აპოლონი |
| | **Russian name:** | Парусник Мнемозина |
| | **Distr. Caucasus:** | Great and Lesser Caucasus. |
| | **Distr. Georgia:** | Widespread in the mountain forests between 800 and 2,400 m. |
| | **Flight:** | Univoltine at lower elevations, April to June. In the high mountains – June to August. |
| | **Habitat:** | Forest glades, grassy slopes, meadows. |

(Continued)

| 31*. *Parnassius nordmanni* Ménétries, 1850 | English name: | Caucasian Apollo | |
| --- | --- | --- | --- |
| | Georgian name: | ნორდმანის აპოლონი | |
| | Russian name: | Парусник Нордманна | |
| | Distr. Caucasus: | Great Caucasus and Lesser Caucasus. | |
| | Distr. Georgia: | High mountains 2,000–4,000 m, sometimes frequent. | |
| | Flight: | Univoltine, July to August. | |
| | Habitat: | Subalpine meadows, screes, rocky slopes. | |

| 32. *Parnassius phoebus* (Fabricius, 1793) | English name: | Phoebus Apollo | |
| --- | --- | --- | --- |
| | Georgian name: | ალპური აპოლონი | |
| | Russian name: | Парусник Феб | |
| | Distr. Caucasus: | Great Caucasus and western part of the Lesser Caucasus. | |
| | Distr. Georgia: | Lesser Caucasus, Adjara mountains. | |
| | Flight: | Univoltine, August to early September. | |
| | Habitat: | Alpine meadows. | |

| 33. *Parnassius apollo* (Linnaeus, 1758) | English name: | Apollo | |
| --- | --- | --- | --- |
| | Georgian name: | აპოლონი | |
| | Russian name: | Парусник Аполлон | |
| | Distr. Caucasus: | Great Caucasus and western part of the Lesser Caucasus. | |
| | Distr. Georgia: | Widespread in the mountains. Locally common in the subalpine zone. | |
| | Flight: | Univoltine, June to August depending on the altitude. | |
| | Habitat: | Stony meadows, rocks, screes. | |

| 34. *Zerynthia caucasica* (Lederer, 1864) | English name: | Caucasian Festoon | |
| --- | --- | --- | --- |
| | Georgian name: | კავკასიური თაისი | |
| | Russian name: | Зеринтия кавказская | |
| | Distr. Caucasus: | Western part of the Great Caucasus and Lesser Caucasus. | |
| | Distr. Georgia: | Great Caucasus and Adjaria. Fairly common. | |
| | Flight: | Univoltine, early April to June depending on the altitude. | |
| | Habitat: | Deciduous forests edges, glades, river valleys. | |

| **Papionidae**, Latreille, 1802 | **Papilioninae**, Latreille, 1802 | | |
| --- | --- | --- | --- |

| 35. *Iphiclides podalirius* (Linnaeus, 1758) | English name: | Scarce Swallowtail | |
| --- | --- | --- | --- |
| | Georgian name: | პოდალირი | |
| | Russian name: | Ификлид Подалирий | |
| | Distr. Caucasus: | Widespread. | |
| | Distr. Georgia: | Common in lower altitudes, up to 2,500 m. | |
| | Flight: | Two or three broods from April to October. | |
| | Habitat: | Gardens, grassy slopes, scrubs. | |

| 36. *Papilio alexanor* Esper, 1800 | English name: | Southern Swallowtail |
| | Georgian name: | |
| | Russian name: | Парусник Алексанор |
| | Distr. Caucasus: | Djavakheti-Armenian plateau. |
| | Distr. Georgia: | Single record from David Gareja monastery (Kakheti). |
| | Flight: | Univoltine, from May to July depending on the altitude. |
| | Habitat: | Rocky slopers and gorges, stony areas. |
| 37. *Papilio machaon* Linnaeus, 1758 | English name: | Swallowtail |
| | Georgian name: | მაქაონი |
| | Russian name: | Хвостоносец Махаон |
| | Distr. Caucasus: | Widespread. |
| | Distr. Georgia: | Common in the mountain ranges. |
| | Flight: | Multivoltine in the lowlands, April to October. Univoltine in the high mountains, May to August. |
| | Habitat: | All types of habitats. |

**Pieridae**, Swainson, 1820 — **Dismorphiinae**, Schatz, 1887

| 38. *Leptidea sinapis* (Linnaeus, 1758) | English name: | Wood White |
| | Georgian name: | |
| | Russian name: | Беляночка горошковая |
| | Distr. Caucasus: | Widespread except Kura lowland. |
| | Distr. Georgia: | Widespread and frequent. |
| | Flight: | Multivoltine, April to October. |
| | Habitat: | Meadows, steppes, glades, woodland clearings. |
| 39. *Leptidea juvernica* Williams, 1946 | English name: | Cryptic Wood White |
| | Georgian name: | |
| | Russian name: | Беляночка ирландская |
| | Distr. Caucasus: | Unknown. |
| | Distr. Georgia: | Unknown. |
| | Flight: | Multivoltine, April to October. |
| | Habitat: | Meadows, steppes, glades, woodland clearings. |
| 40. *Leptidea duponcheli* (Staudinger, 1871) | English name: | Eastern Wood White |
| | Georgian name: | დიუპონშელის თეთრულა |
| | Russian name: | Беляночка Дюпоншеля |
| | Distr. Caucasus: | Lesser Caucasus, sporadic Great Caucasus. |
| | Distr. Georgia: | Lesser Caucasus. Mostly in arid regions but not common. |
| | Flight: | Multivoltine, April to October. |
| | Habitat: | Stony steppes, rocky slopes, gorges. |

*(Continued)*

| Pieridae, Swainson, 1820 | | Pierinae, Duponchel, 1835 | |
|---|---|---|---|
| 41. *Anthocharis cardamines* (Linnaeus, 1758) | English name: | Orange Tip | |
| | Georgian name: | გაზაფხულის თეთრულა | |
| | Russian name: | Зорька обыкновенная | |
| | Distr. Caucasus: | Widespread. | |
| | Distr. Georgia: | Widespread and common. | |
| | Flight: | Univoltine, April to July depending on the altitude. | |
| | Habitat: | Meadows, glades. | |
| 42. *Anthocharis damone* Boisduval, 1836 | English name: | Eastern Orange Tip | |
| | Georgian name: | ამიერკავკასიური აისი | |
| | Russian name: | Зорька Дамона | |
| | Distr. Caucasus: | Eastern part of the Great Caucasus, Lesser Caucasus. | |
| | Distr. Georgia: | Samtskhe-Javakheti, Kvemo Kartli. Very rare. | |
| | Flight: | Univoltine, April to June. | |
| | Habitat: | Open grassy places, river valleys, glades. | |
| 43. *Anthocharis gruneri* Herrich-Schäffer, 1851 | English name: | Grüner's Orange Tip | |
| | Georgian name: | გრუნერის აისი | |
| | Russian name: | Зорька Грюнера | |
| | Distr. Caucasus: | Lesser Caucasus, sporadic in the Great Caucasus. | |
| | Distr. Georgia: | Lesser Caucasus, extremely rare. | |
| | Flight: | Univoltine, April to July depending on the altitude. | |
| | Habitat: | Dry woodlands and grassy areas. | |
| 44. *Zegris eupheme* (Esper, 1804) | English name: | Sooty Orange Tip | |
| | Georgian name: | | |
| | Russian name: | Зегрис Эвфема | |
| | Distr. Caucasus: | North-western part of the Great Caucasus, Lesser Caucasus. | |
| | Distr. Georgia: | Lesser Caucasus. | |
| | Flight: | Univoltine: April to June. | |
| | Habitat: | Dry rocky slopes, steppes, gorges. | |
| 45. *Euchloe ausonia* (Hübner, 1804) | English name: | Eastern Dappled White | |
| | Georgian name: | განთიადის თეთრულაა | |
| | Russian name: | Зорька белая | |
| | Distr. Caucasus: | Southern slopes of the Great Caucasus, Lesser Caucasus. | |
| | Distr. Georgia: | Lesser Caucasus. | |
| | Flight: | Uni- or bivoltine, April to July depending on the locality and altitude. | |
| | Habitat: | Deserts, semi-deserts, rocky slopes, woodland clearings. | |

| 46. *Aporia crataegi* (Linnaeus, 1758) | | |
|---|---|---|
| | **English name:** | Black-veined White |
| | **Georgian name:** | უნელის თეთრულა |
| | **Russian name:** | Боярышница обыкновенная |
| | **Distr. Caucasus:** | Widespread. |
| | **Distr. Georgia:** | Widespread and locally common. |
| | **Flight:** | Multivoltine, April to October. |
| | **Habitat:** | Meadows, gardens, steppes, cultivated areas. |

| 47. *Pieris brassicae* (Linnaeus, 1758) | | |
|---|---|---|
| | **English name:** | Large White |
| | **Georgian name:** | კომბოსტოს თეთრულა |
| | **Russian name:** | Белянка капустная |
| | **Distr. Caucasus:** | Widespread. |
| | **Distr. Georgia:** | Widespread and common. |
| | **Flight:** | Multivoltine, April to September. Two, three, sometimes four broods depending on the locality and altitude. |
| | **Habitat:** | Gardens, cultivated areas, meadows, steppes, woodlands. |

| 48. *Pieris mannii* (Mayer, 1851) | | |
|---|---|---|
| | **English name:** | Southern Small White |
| | **Georgian name:** | |
| | **Russian name:** | Белянка Манна |
| | **Distr. Caucasus:** | Not reported. |
| | **Distr. Georgia:** | Central and western part of the Great Caucasus. |
| | **Flight:** | Multivoltine, April to September. |
| | **Habitat:** | Sunny slopes. |

| 49. *Pieris rapae* (Linnaeus, 1758) | | |
|---|---|---|
| | **English name:** | Small White |
| | **Georgian name:** | თალგამის თეთრულა |
| | **Russian name:** | Белянка репная |
| | **Distr. Caucasus:** | Widespread. |
| | **Distr. Georgia:** | Widespread and common. |
| | **Flight:** | Multivoltine, March to October. |
| | **Habitat:** | Various types, mostly cultivated areas. |

| 50. *Pieris ergane* (Geyer, 1828) | | |
|---|---|---|
| | **English name:** | Mountain Small White |
| | **Georgian name:** | მდოგვის თეთრულა |
| | **Russian name:** | Белянка Эргана |
| | **Distr. Caucasus:** | Central part of northern slopes of the Great Caucasus, Lesser Caucasus. |
| | **Distr. Georgia:** | Lesser Caucasus: Adjara, Guria, Kakheti. |
| | **Flight:** | Multivoltine, April to September, two or three broods. |
| | **Habitat:** | Dry slopes, gorges, rocky steppes. |

(Continued)

| 51. *Pieris napi* (Linnaeus, 1758) | English name: | Green-veined White |
|---|---|---|
|  | Georgian name: | თალგამურას თეთრულა |
| | Russian name: | Белянка брюквенная |
| | Distr. Caucasus: | Widespread. |
| | Distr. Georgia: | Widespread but not common. |
| | Flight: | Multivoltine, March to October. |
| | Habitat: | Various types, mostly cultivated areas. |

| 52. *Pieris bryoniae* (Hübner, 1806) | English name: | Mountain Green-veined White |
|---|---|---|
|  | Georgian name: | |
| | Russian name: | Белянка бриония |
| | Distr. Caucasus: | Great and Lesser Caucasus. |
| | Distr. Georgia: | Abkhasia, Samegrelo-Zemo Svaneti and Samtskhe-Javakheti. |
| | Flight: | Univoltine, May to August. |
| | Habitat: | Grassy slopes, mountain meadows, woodland clearings. |

| 53*. *Pieris bowdeni* Eitschberger, 1984 | English name: | Bowden's White |
|---|---|---|
|  | Georgian name: | |
| | Russian name: | Белянка Боудена |
| | Distr. Caucasus: | Western part of the Lesser Caucasus. |
| | Distr. Georgia: | Single record, requires verification. |
| | Flight: | Univoltine, June to August. |
| | Habitat: | Various grassy habitats. |

| 54. *Pontia callidice* (Hübner, 1800) | English name: | Peak White |
|---|---|---|
|  | Georgian name: | |
| | Russian name: | Белянка альпийская |
| | Distr. Caucasus: | Great and Lesser Caucasus. |
| | Distr. Georgia: | Widespread but not common. |
| | Flight: | Uni- or bivoltine, late May to August. |
| | Habitat: | Mountain meadows, steppes, screes. |

| 55. *Pontia edusa* (Fabricius, 1777) | English name: | Eastern Bath White |
|---|---|---|
|  | Georgian name: | მომწვანო თეთრულა |
| | Russian name: | Белянка рапсовая |
| | Distr. Caucasus: | Widespread. |
| | Distr. Georgia: | Widespread and common. |
| | Flight: | Multivoltine, late April to October. |
| | Habitat: | Various open habitats including cultivated areas. |

| | | |
|---|---|---|
| 56. *Pontia chloridice* (Hübner, 1813) | **English name:** | Small Bath White |
| | **Georgian name:** | |
| | **Russian name:** | Белянка степная |
| | **Distr. Caucasus:** | Central and Eastern part of the Great Caucasus, Lesser Caucasus. |
| | **Distr. Georgia:** | Samtskhe-Javakheti. |
| | **Flight:** | Bivoltine, April-May to July, sometimes August. |
| | **Habitat:** | Dry habitats: steppes, semi-deserts, slopes, gorges. |

**Pieridae**, Swainson, 1820 — **Coliadinae**, Swainson, 1827

| | | |
|---|---|---|
| 57. *Colias erate* (Esper, 1805) | **English name:** | Eastern Pale Clouded Yellow |
| | **Georgian name:** | |
| | **Russian name:** | Желтушка степная |
| | **Distr. Caucasus:** | Widespread. |
| | **Distr. Georgia:** | Widespread and common. |
| | **Flight:** | Multivoltine, May to October. |
| | **Habitat:** | Steppes, hill sides, semi-deserts. |

| | | |
|---|---|---|
| 58. *Colias croceus* (Geoffroy, 1785) | **English name:** | Clouded Yellow |
| | **Georgian name:** | ჩვეულებრივი ყვითელა |
| | **Russian name:** | Желтушка шафрановая |
| | **Distr. Caucasus:** | Widespread. |
| | **Distr. Georgia:** | Widespread and common. |
| | **Flight:** | Multivoltine, May to October. |
| | **Habitat:** | Meadows, steppes, hills, cultivated areas. |

| | | |
|---|---|---|
| 59. *Colias chrysotheme* (Esper, 1781) | **English name:** | Lesser Clouded Yellow |
| | **Georgian name:** | მომწვანო ყვითელა |
| | **Russian name:** | Желтушка золотистая |
| | **Distr. Caucasus:** | Great and Lesser Caucasus. |
| | **Distr. Georgia:** | Great and Lesser Caucasus. Widespread but not common. |
| | **Flight:** | April to October, two or three broods depending on the locality and altitude. |
| | **Habitat:** | River valleys, hills, steppes, scrub. |

| | | |
|---|---|---|
| 60. *Colias aurorina* Herrich-Schäffer, 1850 | **English name:** | Greek Clouded Yellow |
| | **Georgian name:** | |
| | **Russian name:** | Желтушка Аврорина |
| | **Distr. Caucasus:** | Widespread in the mountains. |
| | **Distr. Georgia:** | Lesser Caucasus. Fairly common in arid habitats. |
| | **Flight:** | Univoltine, late June to July depending on the altitude. |
| | **Habitat:** | Dry, rocky slopes, stony steppes. |

(Continued)

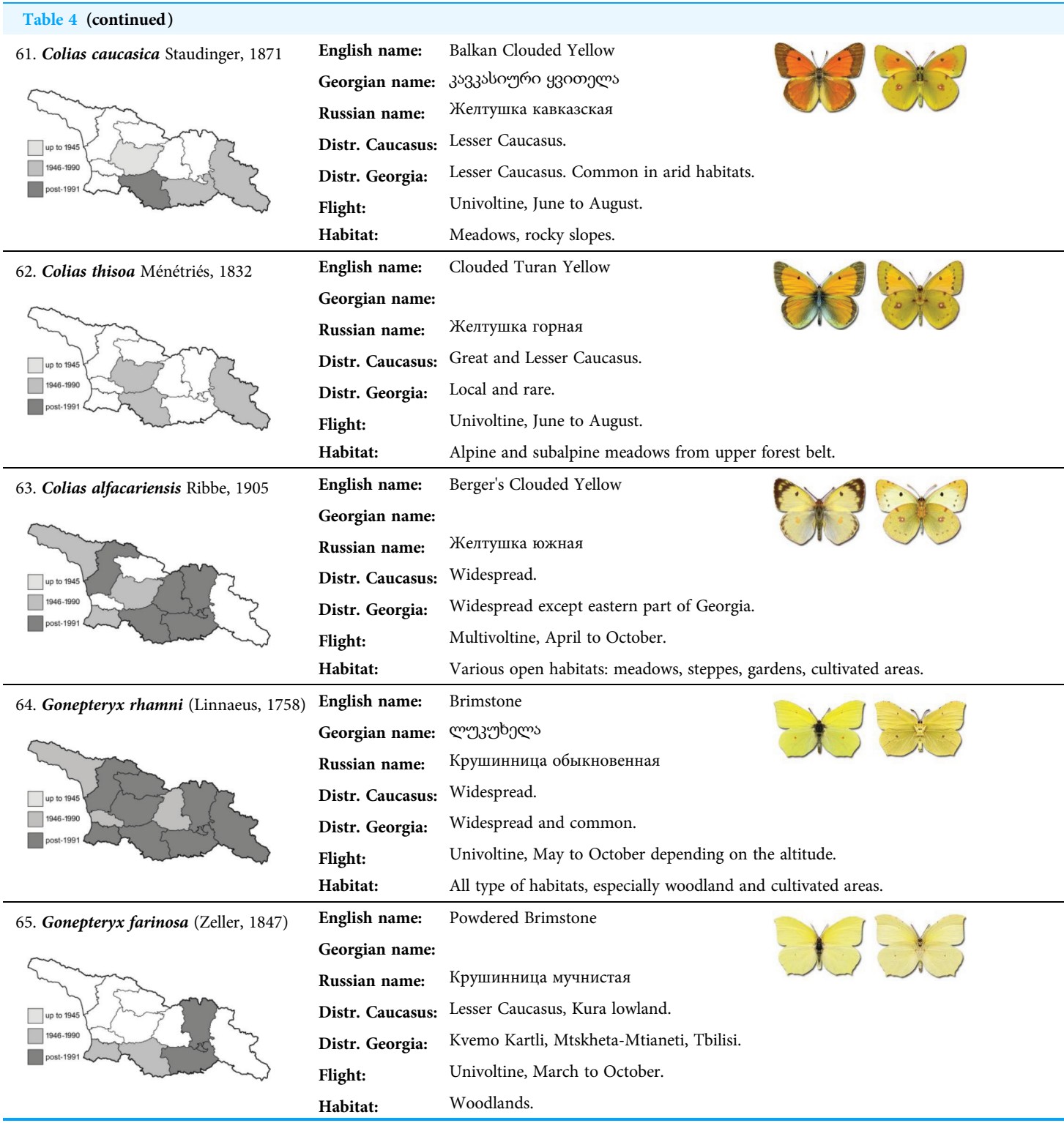

| 61. *Colias caucasica* Staudinger, 1871 | | |
|---|---|---|
| **English name:** | Balkan Clouded Yellow | |
| **Georgian name:** | კავკასიური ყვითელა | |
| **Russian name:** | Желтушка кавказская | |
| **Distr. Caucasus:** | Lesser Caucasus. | |
| **Distr. Georgia:** | Lesser Caucasus. Common in arid habitats. | |
| **Flight:** | Univoltine, June to August. | |
| **Habitat:** | Meadows, rocky slopes. | |

| 62. *Colias thisoa* Ménétriés, 1832 | | |
|---|---|---|
| **English name:** | Clouded Turan Yellow | |
| **Georgian name:** | | |
| **Russian name:** | Желтушка горная | |
| **Distr. Caucasus:** | Great and Lesser Caucasus. | |
| **Distr. Georgia:** | Local and rare. | |
| **Flight:** | Univoltine, June to August. | |
| **Habitat:** | Alpine and subalpine meadows from upper forest belt. | |

| 63. *Colias alfacariensis* Ribbe, 1905 | | |
|---|---|---|
| **English name:** | Berger's Clouded Yellow | |
| **Georgian name:** | | |
| **Russian name:** | Желтушка южная | |
| **Distr. Caucasus:** | Widespread. | |
| **Distr. Georgia:** | Widespread except eastern part of Georgia. | |
| **Flight:** | Multivoltine, April to October. | |
| **Habitat:** | Various open habitats: meadows, steppes, gardens, cultivated areas. | |

| 64. *Gonepteryx rhamni* (Linnaeus, 1758) | | |
|---|---|---|
| **English name:** | Brimstone | |
| **Georgian name:** | ლეკუხელა | |
| **Russian name:** | Крушинница обыкновенная | |
| **Distr. Caucasus:** | Widespread. | |
| **Distr. Georgia:** | Widespread and common. | |
| **Flight:** | Univoltine, May to October depending on the altitude. | |
| **Habitat:** | All type of habitats, especially woodland and cultivated areas. | |

| 65. *Gonepteryx farinosa* (Zeller, 1847) | | |
|---|---|---|
| **English name:** | Powdered Brimstone | |
| **Georgian name:** | | |
| **Russian name:** | Крушинница мучнистая | |
| **Distr. Caucasus:** | Lesser Caucasus, Kura lowland. | |
| **Distr. Georgia:** | Kvemo Kartli, Mtskheta-Mtianeti, Tbilisi. | |
| **Flight:** | Univoltine, March to October. | |
| **Habitat:** | Woodlands. | |

| Lycaenidae, Leach, 1815 | | Lycaeninae, Leach, 1815 |
|---|---|---|

**66. *Lycaena phlaeas* (Linnaeus, 1760)**

| | |
|---|---|
| **English name:** | Small Copper |
| **Georgian name:** | მუქ-წითელა მრავალთვალა |
| **Russian name:** | Червонец пятнистый |
| **Distr. Caucasus:** | Widespread. |
| **Distr. Georgia:** | Widespread and common. |
| **Flight:** | Multivoltine, May to October depending on the altitude. |
| **Habitat:** | Various grassy habitats. |

**67. *Lycaena helle* (Denis & Schiffermüller, 1775)**

| | |
|---|---|
| **English name:** | Violet Copper |
| **Georgian name:** | |
| **Russian name:** | Червонец Гелла |
| **Distr. Caucasus:** | Western part of the Great and Lesser Caucasus. |
| **Distr. Georgia:** | Western part of the Great and Lesser Caucasus. Extremely rare. |
| **Flight:** | Univoltine, sometimes bivoltine, May to August. |
| **Habitat:** | Damp, flowery meadows, river banks. |

**68. *Lycaena dispar* (Haworth, 1802)**

| | |
|---|---|
| **English name:** | Large Copper |
| **Georgian name:** | მჭაუნას მრავალთვალა |
| **Russian name:** | Червонец непарный |
| **Distr. Caucasus:** | Widespread except arid areas. |
| **Distr. Georgia:** | Widespread, sometimes frequent. |
| **Flight:** | Univoltine, sometimes bivoltine, late April to October. |
| **Habitat:** | Meadows, farmland, damp steppes. |

**69. *Lycaena virgaureae* (Linnaeus, 1758)**

| | |
|---|---|
| **English name:** | Scarce Copper |
| **Georgian name:** | ალისფერი მრავალთვალა |
| **Russian name:** | Червонец огненный |
| **Distr. Caucasus:** | Great and Lesser Caucasus. |
| **Distr. Georgia:** | Widespread but not common. |
| **Flight:** | Univoltine, late June to August. |
| **Habitat:** | Meadows, damp steppes, river valleys. |

**70. *Lycaena tityrus* (Poda, 1761)**

| | |
|---|---|
| **English name:** | Sooty Copper |
| **Georgian name:** | |
| **Russian name:** | Червонец чёрнопятнистый |
| **Distr. Caucasus:** | Widespread. |
| **Distr. Georgia:** | Widespread. Fairly frequent except in high mountain areas. |
| **Flight:** | Bivoltine, May to September. |
| **Habitat:** | Dry, grassy meadows, steppes, slopes, roadsides. |

(Continued)

| 71. *Lycaena alciphron* (Rottemburg, 1775) | English name: | Purple-shot Copper |
|---|---|---|
| | Georgian name: | ალისფერი მრავალთვალა |
| | Russian name: | Червонец фиолетовый |
| | Distr. Caucasus: | Widespread except desert and semi-desert areas. |
| | Distr. Georgia: | South-eastern part of Georgia, local. |
| | Flight: | Univoltine, May to July depending on the altitude. |
| | Habitat: | River valleys, meadows, woodland clearings, grassy steppes. |

| 72. *Lycaena candens* (Herrich-Schäffer, 1844) | English name: | Balkan Copper |
|---|---|---|
| | Georgian name: | |
| | Russian name: | Червонец чистый |
| | Distr. Caucasus: | Great and Lesser Caucasus. |
| | Distr. Georgia: | Great and Lesser Caucasus. |
| | Flight: | Univoltine, mid-June to late August. |
| | Habitat: | Subalpine and alpine meadows, forest clearings and glades. |

| 73. *Lycaena thersamon* (Esper, 1784) | English name: | Lesser Fiery Copper |
|---|---|---|
| | Georgian name: | თერზამონი |
| | Russian name: | Червонец блестящий |
| | Distr. Caucasus: | Widespread. |
| | Distr. Georgia: | Widespread but not frequent. |
| | Flight: | Multivoltine, mid-April to October. One-three broods depending on the altitude. |
| | Habitat: | Dry, grassy slopes, meadows, river valleys. |

| 74. *Lycaena thetis* Klug, 1834 | English name: | Golden Copper |
|---|---|---|
| | Georgian name: | |
| | Russian name: | Червонец тетис |
| | Distr. Caucasus: | Djavakheti-Armenian Plateau and Talysh Mountains. |
| | Distr. Georgia: | Vashlovani National Park (K'akheti). |
| | Flight: | Univoltine, mid-July to late August. |
| | Habitat: | Gorges, rocky slopes with xerophytic vegetation. |

| 75*. *Lycaena japhetica* (Nekrutenko & Effendi, 1983) | English name: | |
|---|---|---|
| | Georgian name: | |
| | Russian name: | Атамантия Яфетида |
| | Distr. Caucasus: | Arid areas of western part of Caspian Sea. |
| | Distr. Georgia: | Single location in the vicinity of Kaspi (Shida Kartli). |
| | Flight: | Univoltine, May to late June. |
| | Habitat: | Semi-deserts, slopes with xerophytic vegetation. |

| 76. *Lycaena asabinus* (Herrich-Schäffer, 1851) | English name: | Anatolian Fiery Copper |
|---|---|---|
| | Georgian name: | |
| | Russian name: | Червонец асабинус |
| | Distr. Caucasus: | Central part of the Lesser Caucasus. |
| | Distr. Georgia: | K'akheti and Kvemo Kartli, Extremely rare. |
| | Flight: | Bivoltine, mid-May to late June and mid-July to late August. |
| | Habitat: | Rocky slopes, dry mountain meadows. |

| 77. *Lycaena ochimus* (Herrich-Schäffer, 1851) | English name: | Turkish Fiery Copper |
|---|---|---|
| | Georgian name: | |
| | Russian name: | Червонец охим |
| | Distr. Caucasus: | North-western slopes of the Great Caucasus and Lesser Caucasus. |
| | Distr. Georgia: | Kvemo Kartli, Samtskhe-Javakheti and Abkhazia. |
| | Flight: | Bivoltine, May to mid of June and mid of July to September. |
| | Habitat: | Slopes with xerophytic vegetation, dry steppes. |

| **Lycaenidae**, Leach, 1815 | **Polyommatinae**, Swainson, 1827 |
|---|---|

| 78. *Lampides boeticus* (Linnaeus, 1767) | English name: | Long-tailed Blue |
|---|---|---|
| | Georgian name: | სამყურას მრავალთვალა |
| | Russian name: | Голубянка гороховая |
| | Distr. Caucasus: | Widespread except high mountains. |
| | Distr. Georgia: | Local and rare. Migrant. |
| | Flight: | Multivoltine, April to September. |
| | Habitat: | Grassy areas, meadows, river valleys. |

| 79. *Leptotes pirithous* (Linnaeus, 1767) | English name: | Lang's Short-tailed Blue |
|---|---|---|
| | Georgian name: | |
| | Russian name: | Голубянка Пиритой |
| | Distr. Caucasus: | Western part of the Great and Lesser Caucasus. |
| | Distr. Georgia: | Western part only. |
| | Flight: | Multivoltine, April to November. |
| | Habitat: | Rocky hills and slopes, sometimes fields and gardens. |

| 80*. *Tarucus theophrastus* (Fabricius, 1793) | English name: | Common Tiger Blue |
|---|---|---|
| | Georgian name: | |
| | Russian name: | |
| | Distr. Caucasus: | Unknown. |
| | Distr. Georgia: | Unknown. |
| | Flight: | |
| | Habitat: | Dry sandy and stony areas. |

| 81. *Tarucus balkanica* (Freyer, 1844) | English name: | Little Tiger Blue |
|---|---|---|
| | Georgian name: | ბალკანური ცისფერა |
| | Russian name: | Голубянка балканская |
| | Distr. Caucasus: | Central and eastern part of the Great Caucasus, Lesser Caucasus. |
| | Distr. Georgia: | Abkhazia, K'akheti, Kvemo Kartli. |
| | Flight: | Multivoltine, April to September depending on the altitude. |
| | Habitat: | Semi-deserts, arid foothills, bushy slopes. |
| 82. *Cupido minimus* (Fuessly, 1775) | English name: | Little Blue |
| | Georgian name: | ჯუჯა ცისფერა |
| | Russian name: | Голубянка малая |
| | Distr. Caucasus: | Widespread. |
| | Distr. Georgia: | Widespread and common. |
| | Flight: | Bivoltine, May-September, one brood in high mountains. |
| | Habitat: | Meadows and steppes. |
| 83. *Cupido osiris* (Meigen, 1829) | English name: | Osiris Blue |
| | Georgian name: | პატარა ცისფერა |
| | Russian name: | Голубянка Озирис |
| | Distr. Caucasus: | Widespread. |
| | Distr. Georgia: | Central and eastern part, very rare. |
| | Flight: | Bivoltine, April-May and June-September, high in mountains single brood. |
| | Habitat: | Meadows, woodland glades and clearings. |
| 84. *Cupido argiades* (Pallas, 1771) | English name: | Short-tailed Blue |
| | Georgian name: | მოკლეკუდა ცისფერა |
| | Russian name: | Голубянка аргиад |
| | Distr. Caucasus: | Widespread. |
| | Distr. Georgia: | Widespread and common. |
| | Flight: | Bivoltine, April-June and late June-August. |
| | Habitat: | Forests, meadows, steppes. |
| 85. *Cupido alcetas* (Hoffmannsegg, 1804) | English name: | Provencal Short-Tailed Blue |
| | Georgian name: | |
| | Russian name: | Голубянка альцетас |
| | Distr. Caucasus: | Northern slopes of Great Caucasus |
| | Distr. Georgia: | North Abkhazia. |
| | Flight: | Bi- or three voltine depending on altitude. Rare. |
| | Habitat: | Woodland meadows, mountain pastures. |

| 86. *Celastrina argiolus* (Linnaeus, 1758) | English name: | Holly Blue |
| | Georgian name: | გაზაფხულის ცისფერა |
| | Russian name: | Голубянка весенняя |
| | Distr. Caucasus: | Widespread. |
| | Distr. Georgia: | Widespread and common. |
| | Flight: | Bivoltine, April-May and June-August. |
| | Habitat: | Woodland glades and clearings, sometimes gardens. |

| 87. *Pseudophilotes vicrama* (Moore, 1865) | English name: | Eastern Baton Blue |
| | Georgian name: | |
| | Russian name: | Голубянка викрама |
| | Distr. Caucasus: | Widespread. |
| | Distr. Georgia: | Mainly Lesser Caucasus, not rare. |
| | Flight: | Bivoltine, from April to September. Univoltine in the high mountains. |
| | Habitat: | Steppes with rocky and calcareous places. |

| 88. *Scolitantides orion* (Pallas, 1771) | English name: | Chequered Blue |
| | Georgian name: | |
| | Russian name: | Голубянка Орион |
| | Distr. Caucasus: | Only historical records from Armenia. |
| | Distr. Georgia: | Single record from Ajaria. |
| | Flight: | Bivoltine, April to August. |
| | Habitat: | Woodland clearings, river valleys, steppes. |

| 89. *Glaucopsyche alexis* (Poda, 1761) | English name: | Green-underside Blue |
| | Georgian name: | პეპელა |
| | Russian name: | Голубянка Алексис |
| | Distr. Caucasus: | Widespread. |
| | Distr. Georgia: | Widespread, mainly in the Lesser Caucasus, not common. |
| | Flight: | Univoltine, late May to mid-July. |
| | Habitat: | Various grassy habitats. |

| 90. *Phengaris arion* (Linnaeus, 1758) | English name: | Large Blue |
| | Georgian name: | |
| | Russian name: | Голубянка Арион |
| | Distr. Caucasus: | Widespread. |
| | Distr. Georgia: | Local and rare. |
| | Flight: | Univoltine, mid-May to August. |
| | Habitat: | Various habitats: steppes, slopes, bushy moorland areas, arid mountains. |

Maps legend: up to 1945, 1946-1990, post-1991

### 91. *Phengaris teleius* (Bergsträsser, 1779)

| | |
|---|---|
| **English name:** | Scarce Large Blue |
| **Georgian name:** | |
| **Russian name:** | Голубянка точечная |
| **Distr. Caucasus:** | Western part of the Great and Lesser Caucasus. |
| **Distr. Georgia:** | Single locations in the western part only. |
| **Flight:** | Univoltine, mid-June to mid-July. |
| **Habitat:** | Mountain meadows, damp forest glades. |

### 92. *Phengaris nausithous* (Bergsträsser, 1779)

| | |
|---|---|
| **English name:** | Dusky Large Blue |
| **Georgian name:** | |
| **Russian name:** | Голубянка черноватая |
| **Distr. Caucasus:** | Great and Lesser Caucasus. |
| **Distr. Georgia:** | Lesser Caucasus, extremely rare. |
| **Flight:** | Univoltine, late June to mid-August. |
| **Habitat:** | Damp meadows and woodland clearings, swamps, forest glades. |

### 93. *Phengaris alcon* (Denis & Schiffermüller, 1775)

| | |
|---|---|
| **English name:** | Alcon Blue |
| **Georgian name:** | |
| **Russian name:** | Голубянка Алькон |
| **Distr. Caucasus:** | Great and Lesser Caucasus. |
| **Distr. Georgia:** | Central part, mainly in the Lesser Caucasus, rare. |
| **Flight:** | Univoltine, June-August. |
| **Habitat:** | Damp meadows and woodland clearings, swamps, grassy slopes. |

### 94. *Luthrodes galba* (Lederer, 1855)

| | |
|---|---|
| **English name:** | Small Desert Blue |
| **Georgian name:** | |
| **Russian name:** | Голубянка гальба |
| **Distr. Caucasus:** | Talysh, Kura and Arax valleys. |
| **Distr. Georgia:** | K'akheti only. |
| **Flight:** | Multivoltine, May to October. |
| **Habitat:** | Dry, rocky places, deserts and semi-deserts. |

### 95. *Plebejus argus* (Linnaeus, 1758)

| | |
|---|---|
| **English name:** | Silver-studded Blue |
| **Georgian name:** | არგუსი |
| **Russian name:** | Голубянка Аргус |
| **Distr. Caucasus:** | Widespread except semi-deserts. |
| **Distr. Georgia:** | Widespread, common but not abundant. |
| **Flight:** | Uni- or bivoltine, April to August depending on the altitude. |
| **Habitat:** | Various grassy places. |

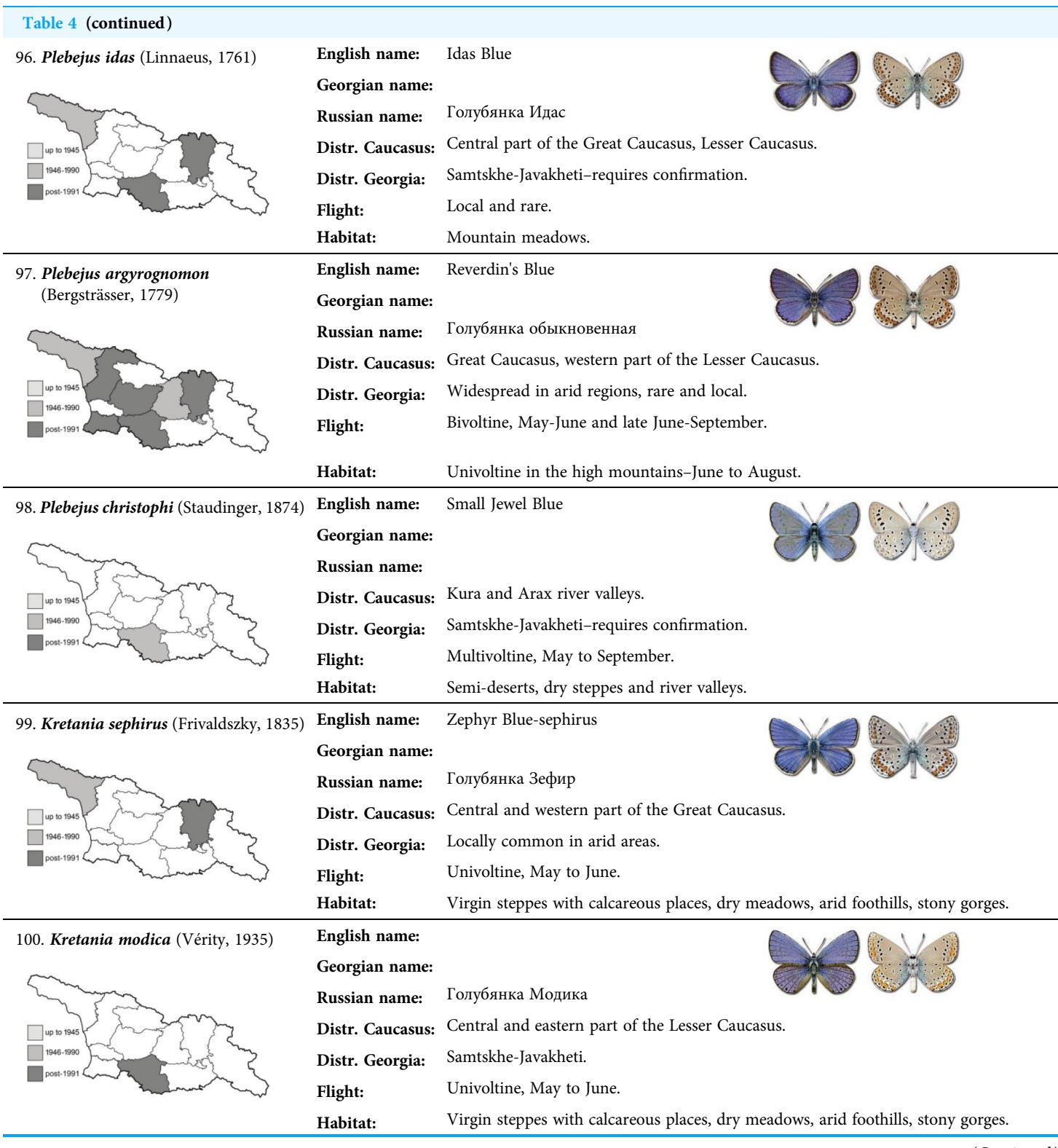

| | | |
|---|---|---|
| 96. *Plebejus idas* (Linnaeus, 1761) | **English name:** | Idas Blue |
| | **Georgian name:** | |
| | **Russian name:** | Голубянка Идас |
| | **Distr. Caucasus:** | Central part of the Great Caucasus, Lesser Caucasus. |
| | **Distr. Georgia:** | Samtskhe-Javakheti–requires confirmation. |
| | **Flight:** | Local and rare. |
| | **Habitat:** | Mountain meadows. |
| 97. *Plebejus argyrognomon* (Bergsträsser, 1779) | **English name:** | Reverdin's Blue |
| | **Georgian name:** | |
| | **Russian name:** | Голубянка обыкновенная |
| | **Distr. Caucasus:** | Great Caucasus, western part of the Lesser Caucasus. |
| | **Distr. Georgia:** | Widespread in arid regions, rare and local. |
| | **Flight:** | Bivoltine, May-June and late June-September. |
| | **Habitat:** | Univoltine in the high mountains–June to August. |
| 98. *Plebejus christophi* (Staudinger, 1874) | **English name:** | Small Jewel Blue |
| | **Georgian name:** | |
| | **Russian name:** | |
| | **Distr. Caucasus:** | Kura and Arax river valleys. |
| | **Distr. Georgia:** | Samtskhe-Javakheti–requires confirmation. |
| | **Flight:** | Multivoltine, May to September. |
| | **Habitat:** | Semi-deserts, dry steppes and river valleys. |
| 99. *Kretania sephirus* (Frivaldszky, 1835) | **English name:** | Zephyr Blue-sephirus |
| | **Georgian name:** | |
| | **Russian name:** | Голубянка Зефир |
| | **Distr. Caucasus:** | Central and western part of the Great Caucasus. |
| | **Distr. Georgia:** | Locally common in arid areas. |
| | **Flight:** | Univoltine, May to June. |
| | **Habitat:** | Virgin steppes with calcareous places, dry meadows, arid foothills, stony gorges. |
| 100. *Kretania modica* (Vérity, 1935) | **English name:** | |
| | **Georgian name:** | |
| | **Russian name:** | Голубянка Модика |
| | **Distr. Caucasus:** | Central and eastern part of the Lesser Caucasus. |
| | **Distr. Georgia:** | Samtskhe-Javakheti. |
| | **Flight:** | Univoltine, May to June. |
| | **Habitat:** | Virgin steppes with calcareous places, dry meadows, arid foothills, stony gorges. |

(Continued)

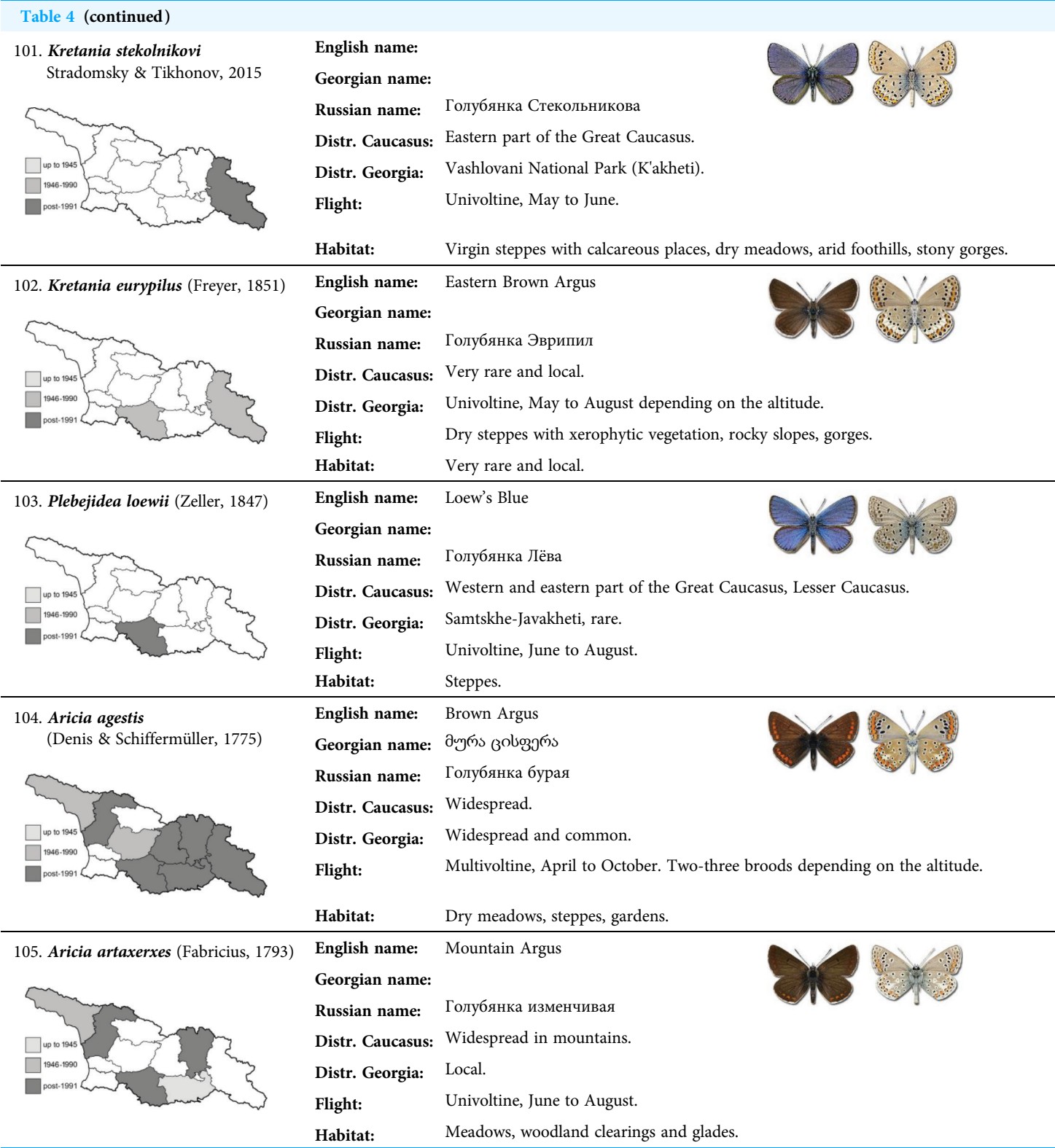

| | | |
|---|---|---|
| 101. *Kretania stekolnikovi* Stradomsky & Tikhonov, 2015 | **English name:** | |
| | **Georgian name:** | |
| | **Russian name:** | Голубянка Стекольникова |
| | **Distr. Caucasus:** | Eastern part of the Great Caucasus. |
| | **Distr. Georgia:** | Vashlovani National Park (K'akheti). |
| | **Flight:** | Univoltine, May to June. |
| | **Habitat:** | Virgin steppes with calcareous places, dry meadows, arid foothills, stony gorges. |
| 102. *Kretania eurypilus* (Freyer, 1851) | **English name:** | Eastern Brown Argus |
| | **Georgian name:** | |
| | **Russian name:** | Голубянка Эврипил |
| | **Distr. Caucasus:** | Very rare and local. |
| | **Distr. Georgia:** | Univoltine, May to August depending on the altitude. |
| | **Flight:** | Dry steppes with xerophytic vegetation, rocky slopes, gorges. |
| | **Habitat:** | Very rare and local. |
| 103. *Plebejidea loewii* (Zeller, 1847) | **English name:** | Loew's Blue |
| | **Georgian name:** | |
| | **Russian name:** | Голубянка Лёва |
| | **Distr. Caucasus:** | Western and eastern part of the Great Caucasus, Lesser Caucasus. |
| | **Distr. Georgia:** | Samtskhe-Javakheti, rare. |
| | **Flight:** | Univoltine, June to August. |
| | **Habitat:** | Steppes. |
| 104. *Aricia agestis* (Denis & Schiffermüller, 1775) | **English name:** | Brown Argus |
| | **Georgian name:** | მურა ცისფერა |
| | **Russian name:** | Голубянка бурая |
| | **Distr. Caucasus:** | Widespread. |
| | **Distr. Georgia:** | Widespread and common. |
| | **Flight:** | Multivoltine, April to October. Two-three broods depending on the altitude. |
| | **Habitat:** | Dry meadows, steppes, gardens. |
| 105. *Aricia artaxerxes* (Fabricius, 1793) | **English name:** | Mountain Argus |
| | **Georgian name:** | |
| | **Russian name:** | Голубянка изменчивая |
| | **Distr. Caucasus:** | Widespread in mountains. |
| | **Distr. Georgia:** | Local. |
| | **Flight:** | Univoltine, June to August. |
| | **Habitat:** | Meadows, woodland clearings and glades. |

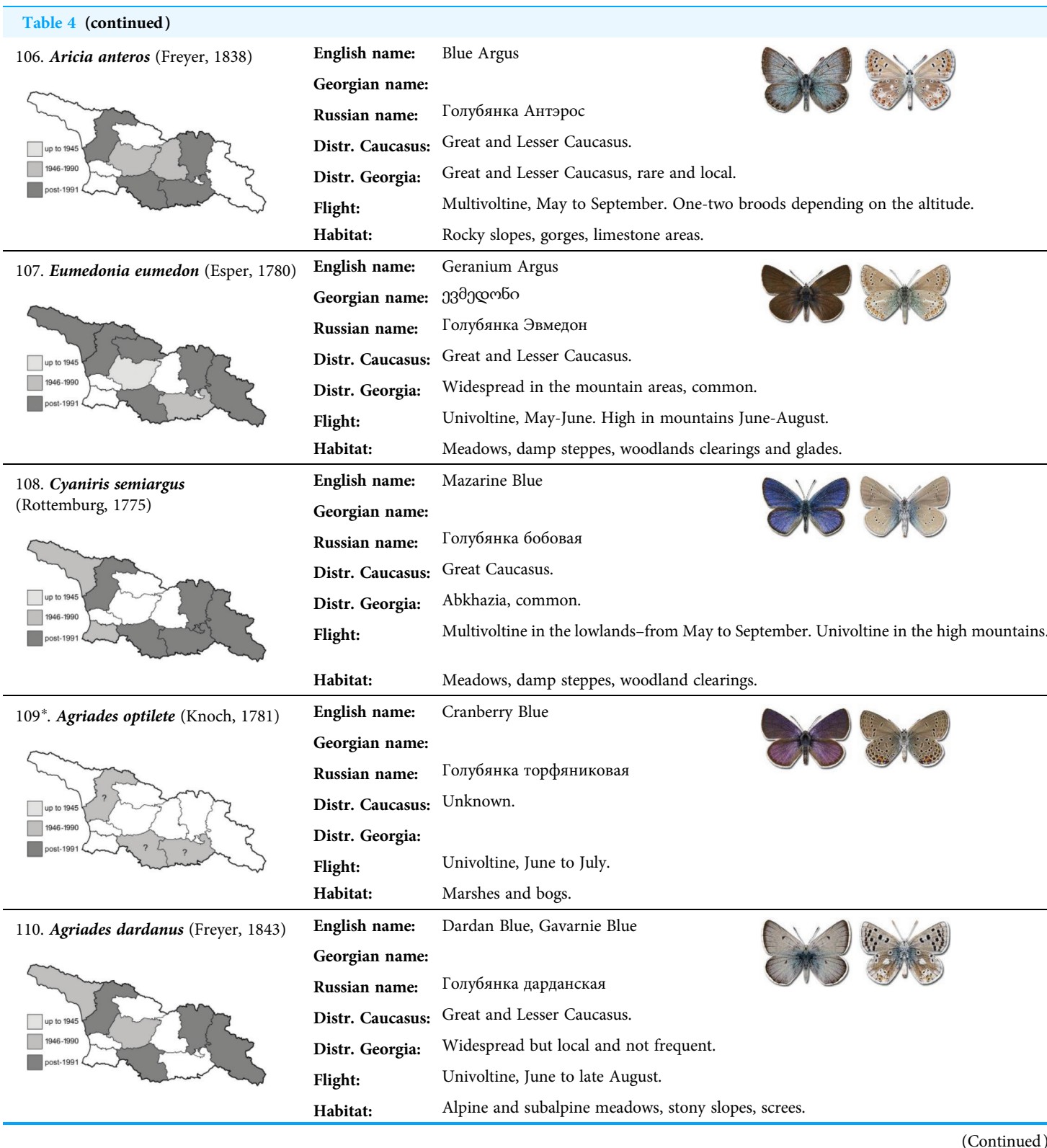

| | | |
|---|---|---|
| **106. *Aricia anteros*** (Freyer, 1838) | **English name:** | Blue Argus |
| | **Georgian name:** | |
| | **Russian name:** | Голубянка Антэрос |
| | **Distr. Caucasus:** | Great and Lesser Caucasus. |
| | **Distr. Georgia:** | Great and Lesser Caucasus, rare and local. |
| | **Flight:** | Multivoltine, May to September. One-two broods depending on the altitude. |
| | **Habitat:** | Rocky slopes, gorges, limestone areas. |
| **107. *Eumedonia eumedon*** (Esper, 1780) | **English name:** | Geranium Argus |
| | **Georgian name:** | ევმედონი |
| | **Russian name:** | Голубянка Эвмедон |
| | **Distr. Caucasus:** | Great and Lesser Caucasus. |
| | **Distr. Georgia:** | Widespread in the mountain areas, common. |
| | **Flight:** | Univoltine, May-June. High in mountains June-August. |
| | **Habitat:** | Meadows, damp steppes, woodlands clearings and glades. |
| **108. *Cyaniris semiargus*** (Rottemburg, 1775) | **English name:** | Mazarine Blue |
| | **Georgian name:** | |
| | **Russian name:** | Голубянка бобовая |
| | **Distr. Caucasus:** | Great Caucasus. |
| | **Distr. Georgia:** | Abkhazia, common. |
| | **Flight:** | Multivoltine in the lowlands–from May to September. Univoltine in the high mountains. |
| | **Habitat:** | Meadows, damp steppes, woodland clearings. |
| **109*. *Agriades optilete*** (Knoch, 1781) | **English name:** | Cranberry Blue |
| | **Georgian name:** | |
| | **Russian name:** | Голубянка торфяниковая |
| | **Distr. Caucasus:** | Unknown. |
| | **Distr. Georgia:** | |
| | **Flight:** | Univoltine, June to July. |
| | **Habitat:** | Marshes and bogs. |
| **110. *Agriades dardanus*** (Freyer, 1843) | **English name:** | Dardan Blue, Gavarnie Blue |
| | **Georgian name:** | |
| | **Russian name:** | Голубянка дарданская |
| | **Distr. Caucasus:** | Great and Lesser Caucasus. |
| | **Distr. Georgia:** | Widespread but local and not frequent. |
| | **Flight:** | Univoltine, June to late August. |
| | **Habitat:** | Alpine and subalpine meadows, stony slopes, screes. |

*(Continued)*

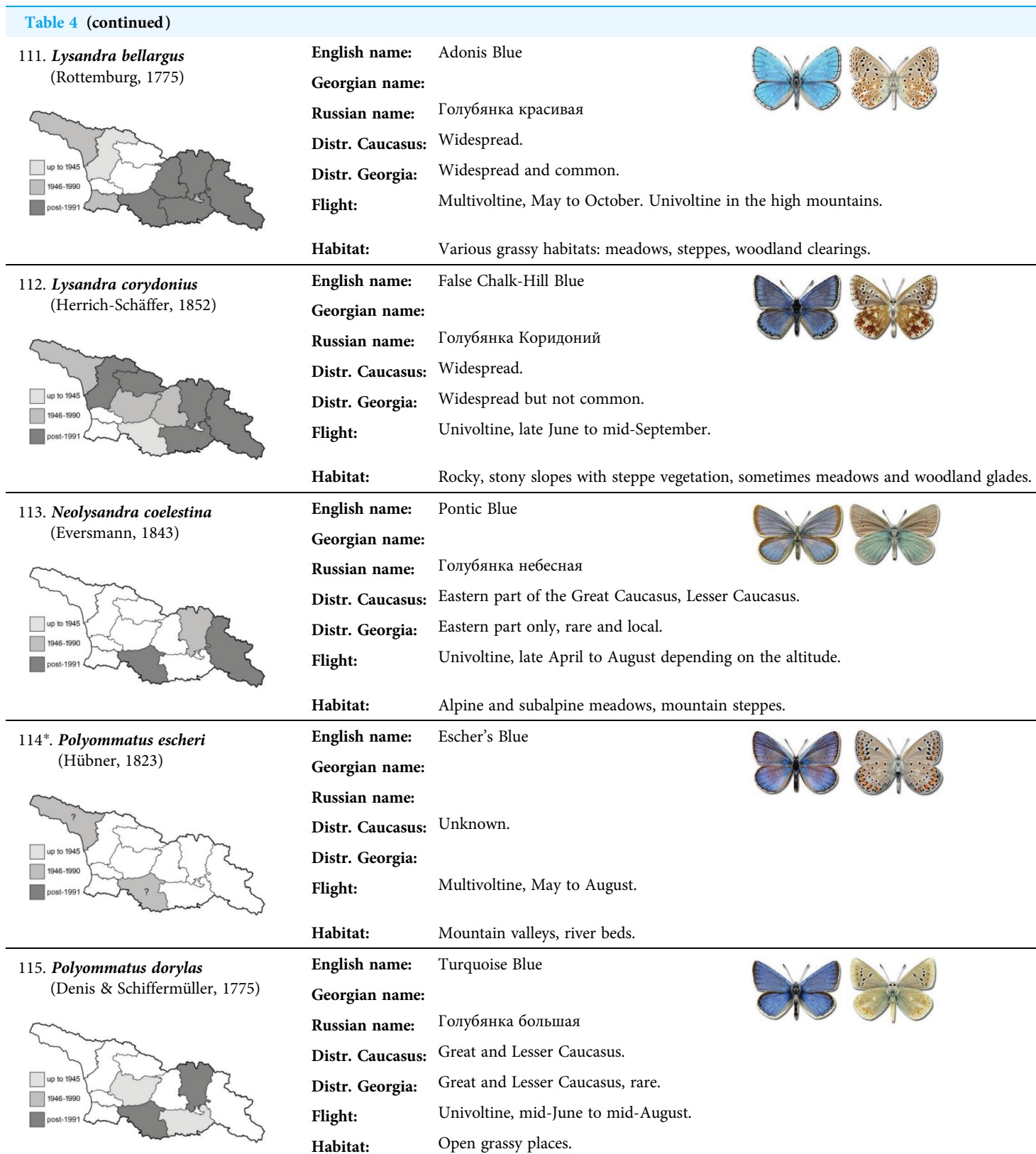

| | | |
|---|---|---|
| 111. *Lysandra bellargus* (Rottemburg, 1775) | **English name:** | Adonis Blue |
| | **Georgian name:** | |
| | **Russian name:** | Голубянка красивая |
| | **Distr. Caucasus:** | Widespread. |
| | **Distr. Georgia:** | Widespread and common. |
| | **Flight:** | Multivoltine, May to October. Univoltine in the high mountains. |
| | **Habitat:** | Various grassy habitats: meadows, steppes, woodland clearings. |
| 112. *Lysandra corydonius* (Herrich-Schäffer, 1852) | **English name:** | False Chalk-Hill Blue |
| | **Georgian name:** | |
| | **Russian name:** | Голубянка Коридоний |
| | **Distr. Caucasus:** | Widespread. |
| | **Distr. Georgia:** | Widespread but not common. |
| | **Flight:** | Univoltine, late June to mid-September. |
| | **Habitat:** | Rocky, stony slopes with steppe vegetation, sometimes meadows and woodland glades. |
| 113. *Neolysandra coelestina* (Eversmann, 1843) | **English name:** | Pontic Blue |
| | **Georgian name:** | |
| | **Russian name:** | Голубянка небесная |
| | **Distr. Caucasus:** | Eastern part of the Great Caucasus, Lesser Caucasus. |
| | **Distr. Georgia:** | Eastern part only, rare and local. |
| | **Flight:** | Univoltine, late April to August depending on the altitude. |
| | **Habitat:** | Alpine and subalpine meadows, mountain steppes. |
| 114*. *Polyommatus escheri* (Hübner, 1823) | **English name:** | Escher's Blue |
| | **Georgian name:** | |
| | **Russian name:** | |
| | **Distr. Caucasus:** | Unknown. |
| | **Distr. Georgia:** | |
| | **Flight:** | Multivoltine, May to August. |
| | **Habitat:** | Mountain valleys, river beds. |
| 115. *Polyommatus dorylas* (Denis & Schiffermüller, 1775) | **English name:** | Turquoise Blue |
| | **Georgian name:** | |
| | **Russian name:** | Голубянка большая |
| | **Distr. Caucasus:** | Great and Lesser Caucasus. |
| | **Distr. Georgia:** | Great and Lesser Caucasus, rare. |
| | **Flight:** | Univoltine, mid-June to mid-August. |
| | **Habitat:** | Open grassy places. |

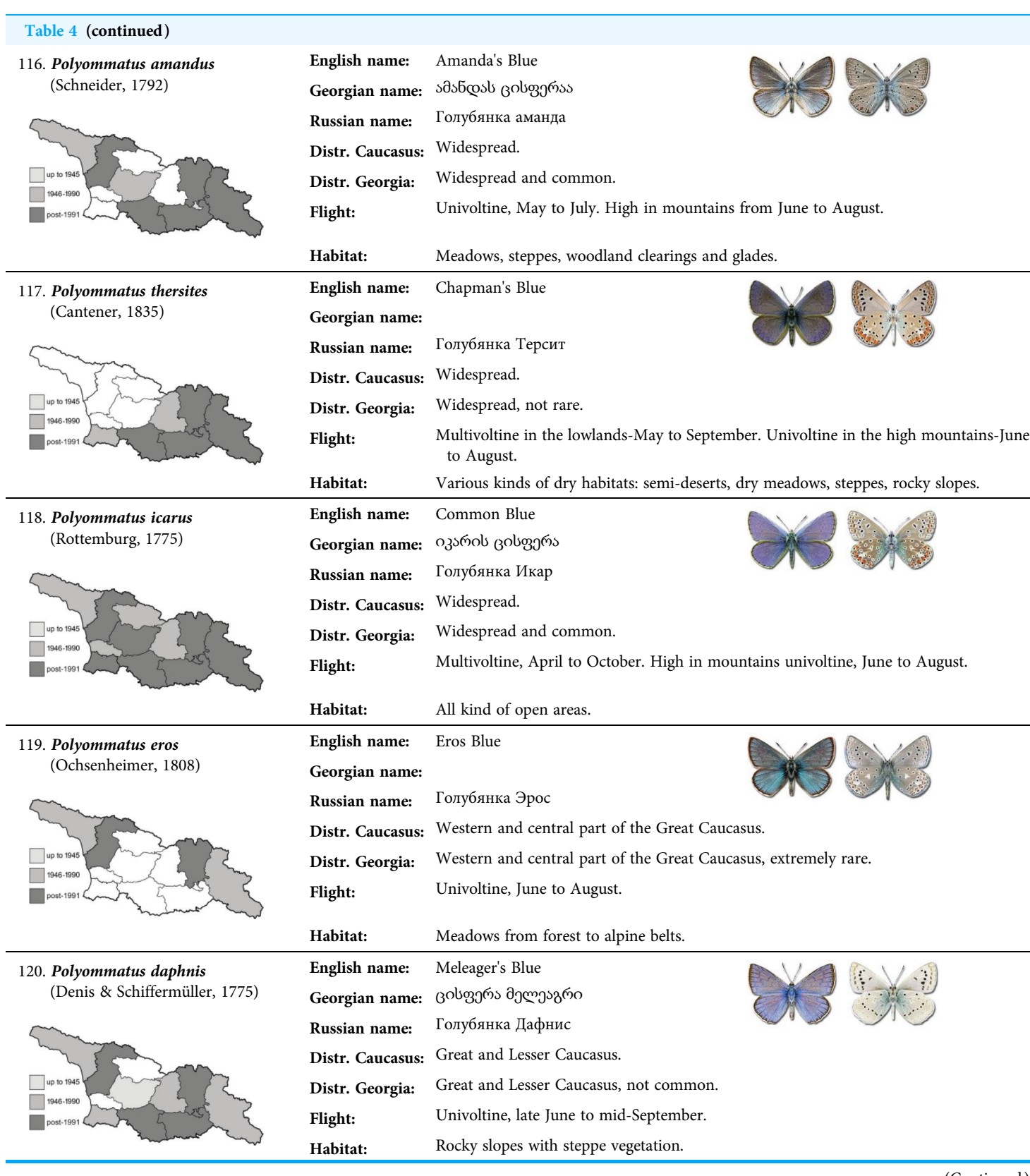

| | | |
|---|---|---|
| 116. *Polyommatus amandus* (Schneider, 1792) | **English name:** | Amanda's Blue |
| | **Georgian name:** | ამანდას ცისფერაა |
| | **Russian name:** | Голубянка аманда |
| | **Distr. Caucasus:** | Widespread. |
| | **Distr. Georgia:** | Widespread and common. |
| | **Flight:** | Univoltine, May to July. High in mountains from June to August. |
| | **Habitat:** | Meadows, steppes, woodland clearings and glades. |
| 117. *Polyommatus thersites* (Cantener, 1835) | **English name:** | Chapman's Blue |
| | **Georgian name:** | |
| | **Russian name:** | Голубянка Терсит |
| | **Distr. Caucasus:** | Widespread. |
| | **Distr. Georgia:** | Widespread, not rare. |
| | **Flight:** | Multivoltine in the lowlands-May to September. Univoltine in the high mountains-June to August. |
| | **Habitat:** | Various kinds of dry habitats: semi-deserts, dry meadows, steppes, rocky slopes. |
| 118. *Polyommatus icarus* (Rottemburg, 1775) | **English name:** | Common Blue |
| | **Georgian name:** | იკარის ცისფერა |
| | **Russian name:** | Голубянка Икар |
| | **Distr. Caucasus:** | Widespread. |
| | **Distr. Georgia:** | Widespread and common. |
| | **Flight:** | Multivoltine, April to October. High in mountains univoltine, June to August. |
| | **Habitat:** | All kind of open areas. |
| 119. *Polyommatus eros* (Ochsenheimer, 1808) | **English name:** | Eros Blue |
| | **Georgian name:** | |
| | **Russian name:** | Голубянка Эрос |
| | **Distr. Caucasus:** | Western and central part of the Great Caucasus. |
| | **Distr. Georgia:** | Western and central part of the Great Caucasus, extremely rare. |
| | **Flight:** | Univoltine, June to August. |
| | **Habitat:** | Meadows from forest to alpine belts. |
| 120. *Polyommatus daphnis* (Denis & Schiffermüller, 1775) | **English name:** | Meleager's Blue |
| | **Georgian name:** | ცისფერა მელეაგრი |
| | **Russian name:** | Голубянка Дафнис |
| | **Distr. Caucasus:** | Great and Lesser Caucasus. |
| | **Distr. Georgia:** | Great and Lesser Caucasus, not common. |
| | **Flight:** | Univoltine, late June to mid-September. |
| | **Habitat:** | Rocky slopes with steppe vegetation. |

*(Continued)*

| 121. *Polyommatus admetus* (Esper, 1783) | | |
|---|---|---|
| | English name: | Anomalous Blue |
| | Georgian name: | |
| | Russian name: | Голубянка адмет |
| | Distr. Caucasus: | Central part of the Lesser Caucasus. |
| | Distr. Georgia: | Two historical records only. Probably extinct in Georgia now. |
| | Flight: | Univoltine, mid-June to August. |
| | Habitat: | Rocky slopes, dry meadows, mountain steppes. |

| 122. *Polyommatus ripartii* (Freyer, 1830) | | |
|---|---|---|
| | English name: | Ripart's Anomalous Blue |
| | Georgian name: | |
| | Russian name: | Голубянка Рипперта |
| | Distr. Caucasus: | Great and Lesser Caucasus. |
| | Distr. Georgia: | Kvemo Kartli, Mtskheta-Mtianeti and Shida Kartli. |
| | Flight: | Univoltine, mid-June to late August depending on the altitude. |
| | Habitat: | Various types of dry, grassy areas. |

| 123*. *Polyommatus eriwanensis* (Forster, 1960) | | |
|---|---|---|
| | English name: | Eriwan Anamolous Blue |
| | Georgian name: | |
| | Russian name: | Голубянка ереванская |
| | Distr. Caucasus: | Great Caucasus and Lesser Caucasus. |
| | Distr. Georgia: | Two locations only, rare. |
| | Flight: | Univoltine, mid-June to early August. |
| | Habitat: | Various types of dry grasslands. |

| 124. *Polyommatus damon* (Denis & Schiffermüller, 1775) | | |
|---|---|---|
| | English name: | Damon Blue |
| | Georgian name: | |
| | Russian name: | Голубянка Дамон |
| | Distr. Caucasus: | Lesser Caucasus, Djavakheti-Armenian plateau, Talysh. |
| | Distr. Georgia: | Samtskhe-Javakheti, single record from Kvemo Kartli. |
| | Flight: | Univoltine, late June to August. |
| | Habitat: | Dry meadows, steppes. |

| 125*. *Polyommatus wagneri* (Forster, 1960) | | |
|---|---|---|
| | English name: | Wagner's Blue |
| | Georgian name: | |
| | Russian name: | |
| | Distr. Caucasus: | Not reported before. |
| | Distr. Georgia: | Samtskhe-Javakheti, extremely rare. |
| | Flight: | Univoltine, June to August. |
| | Habitat: | Flowered steppes, rocky slopes. |

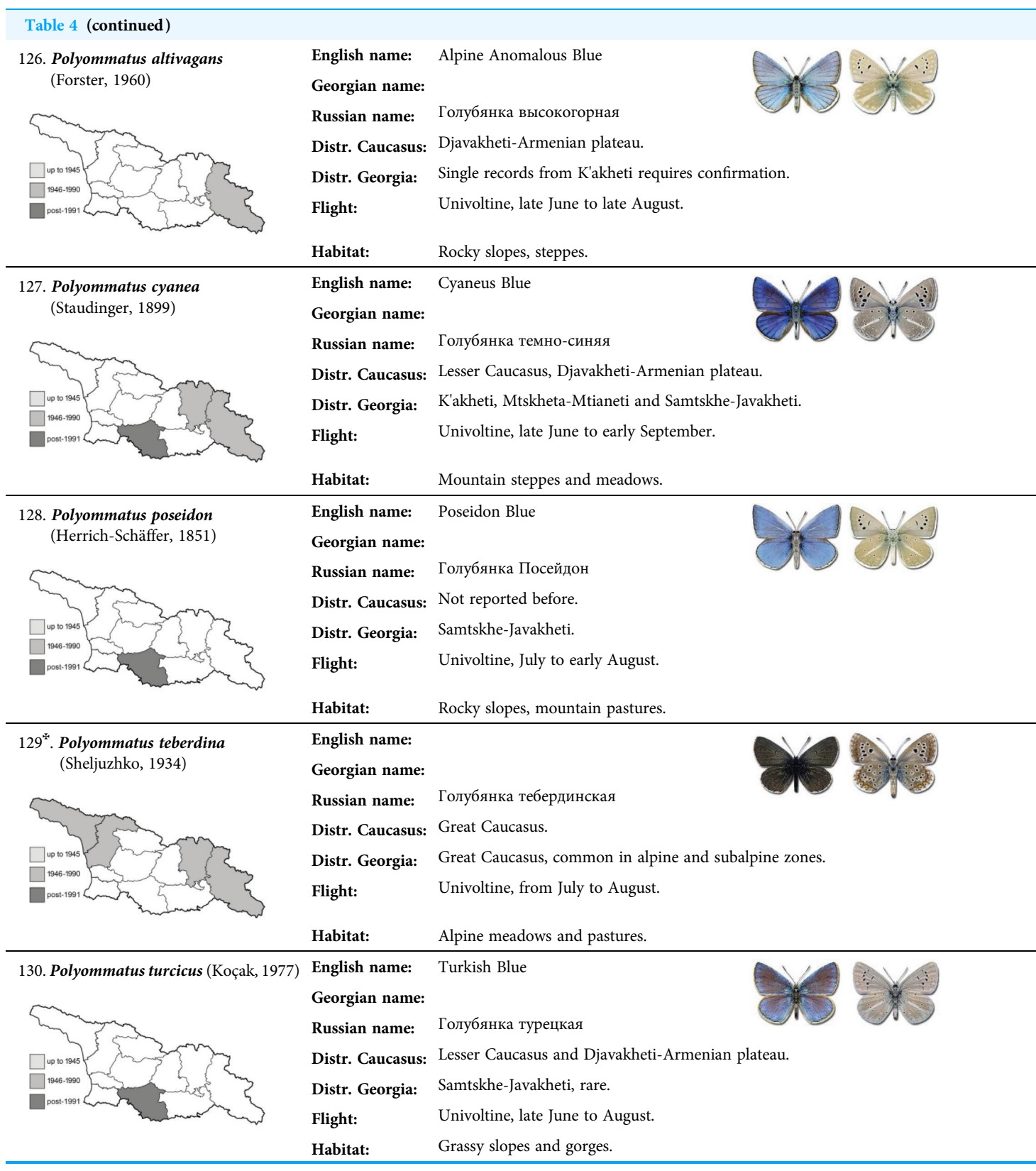

**126. *Polyommatus altivagans***
(Forster, 1960)

| | |
|---|---|
| **English name:** | Alpine Anomalous Blue |
| **Georgian name:** | |
| **Russian name:** | Голубянка высокогорная |
| **Distr. Caucasus:** | Djavakheti-Armenian plateau. |
| **Distr. Georgia:** | Single records from K'akheti requires confirmation. |
| **Flight:** | Univoltine, late June to late August. |
| **Habitat:** | Rocky slopes, steppes. |

*up to 1945 / 1946-1990 / post-1991*

**127. *Polyommatus cyanea***
(Staudinger, 1899)

| | |
|---|---|
| **English name:** | Cyaneus Blue |
| **Georgian name:** | |
| **Russian name:** | Голубянка темно-синяя |
| **Distr. Caucasus:** | Lesser Caucasus, Djavakheti-Armenian plateau. |
| **Distr. Georgia:** | K'akheti, Mtskheta-Mtianeti and Samtskhe-Javakheti. |
| **Flight:** | Univoltine, late June to early September. |
| **Habitat:** | Mountain steppes and meadows. |

**128. *Polyommatus poseidon***
(Herrich-Schäffer, 1851)

| | |
|---|---|
| **English name:** | Poseidon Blue |
| **Georgian name:** | |
| **Russian name:** | Голубянка Посейдон |
| **Distr. Caucasus:** | Not reported before. |
| **Distr. Georgia:** | Samtskhe-Javakheti. |
| **Flight:** | Univoltine, July to early August. |
| **Habitat:** | Rocky slopes, mountain pastures. |

**129*. *Polyommatus teberdina***
(Sheljuzhko, 1934)

| | |
|---|---|
| **English name:** | |
| **Georgian name:** | |
| **Russian name:** | Голубянка тебердинская |
| **Distr. Caucasus:** | Great Caucasus. |
| **Distr. Georgia:** | Great Caucasus, common in alpine and subalpine zones. |
| **Flight:** | Univoltine, from July to August. |
| **Habitat:** | Alpine meadows and pastures. |

**130. *Polyommatus turcicus*** (Koçak, 1977)

| | |
|---|---|
| **English name:** | Turkish Blue |
| **Georgian name:** | |
| **Russian name:** | Голубянка турецкая |
| **Distr. Caucasus:** | Lesser Caucasus and Djavakheti-Armenian plateau. |
| **Distr. Georgia:** | Samtskhe-Javakheti, rare. |
| **Flight:** | Univoltine, late June to August. |
| **Habitat:** | Grassy slopes and gorges. |

| Table 4 (continued) | | | |
|---|---|---|---|
| 131. *Polyommatus aserbeidschana* (Forster, 1960) | **English name:** | Azerbaijan Blue | |
| | **Georgian name:** | | |
| | **Russian name:** | Голубянка азербайджанская | |
| | **Distr. Caucasus:** | Lesser Caucasus, Djavakheti-Armenian plateau. | |
| | **Distr. Georgia:** | Samtskhe-Javakheti, rare. | |
| | **Flight:** | Univoltine, late June to late August. | |
| | **Habitat:** | Stony steppes, xerothermic slopes. | |
| 132. *Polyommatus ninae* (Forster, 1960) | **English name:** | Nina's Blue | |
| | **Georgian name:** | | |
| | **Russian name:** | Голубянка Нины | |
| | **Distr. Caucasus:** | Central part of the Lesser Caucasus, Djavakheti-Armenian plateau. | |
| | **Distr. Georgia:** | Single location in the vicinity of Tbilisi, requires verification. | |
| | **Flight:** | Univoltine, early July to late August. | |
| | **Habitat:** | Dry xerophytic slopes, dry rocky steppes. | |
| 133. *Polyommatus demavendi* (Pfeiffer, 1938) | **English name:** | Persian Anomalous Blue | |
| | **Georgian name:** | | |
| | **Russian name:** | Голубянка демавендская | |
| | **Distr. Caucasus:** | Eastern part of the Great Caucasus, Lesser Caucasus. | |
| | **Distr. Georgia:** | Samtskhe-Javakheti, rare. | |
| | **Flight:** | Univoltine, late June to August. | |
| | **Habitat:** | Steppes, rocky places, sometimes dry meadows and forest glades. | |
| 134. *Afarsia morgiana* (Kirby, 1871) | **English name:** | Persian Blue | |
| | **Georgian name:** | | |
| | **Russian name:** | Голубянка Моргиана | |
| | **Distr. Caucasus:** | Djavakheti-Armenian Plateau. | |
| | **Distr. Georgia:** | Tbilisi, Kvemo Kartli. | |
| | **Flight:** | Univoltine, early June to late July. | |
| | **Habitat:** | Rocky slopes, steppes, cliffs. | |
| **Lycaenidae**, Leach, 1815 | | **Theclinae**, Swainson, 1831 | |
| 135. *Satyrium w-album* (Knoch, 1782) | **English name:** | White-letter Hairstreak | |
| | **Georgian name:** | თელის კუდალა | |
| | **Russian name:** | Хвостатка вязовая | |
| | **Distr. Caucasus:** | Widespread. | |
| | **Distr. Georgia:** | Widespread and frequent. | |
| | **Flight:** | Univoltine, mid-June to early August. | |
| | **Habitat:** | Mixed forests, forest edges, gardens. | |

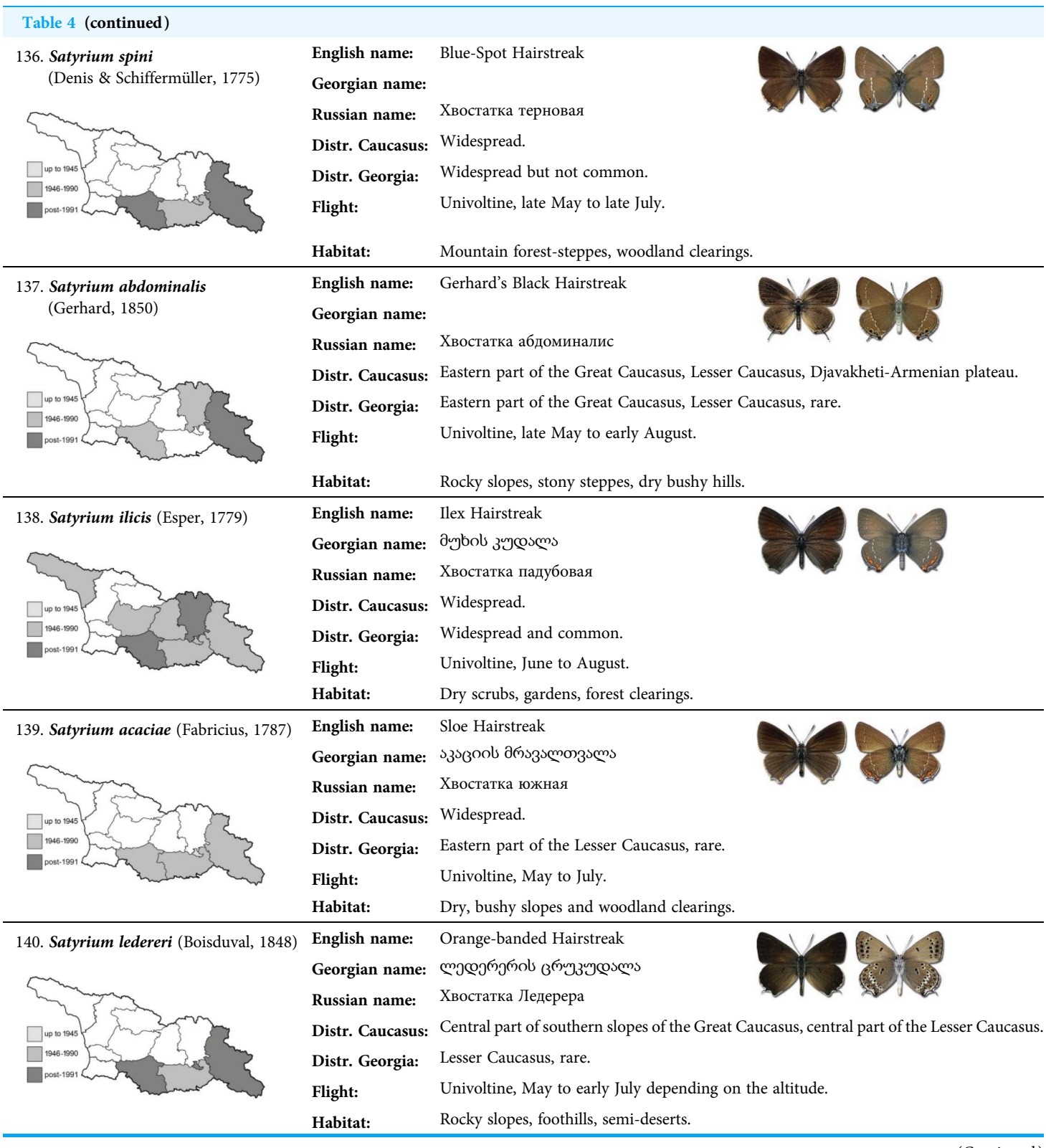

| | | |
|---|---|---|
| 136. *Satyrium spini* (Denis & Schiffermüller, 1775) | **English name:** | Blue-Spot Hairstreak |
| | **Georgian name:** | |
| | **Russian name:** | Хвостатка терновая |
| | **Distr. Caucasus:** | Widespread. |
| | **Distr. Georgia:** | Widespread but not common. |
| | **Flight:** | Univoltine, late May to late July. |
| | **Habitat:** | Mountain forest-steppes, woodland clearings. |
| 137. *Satyrium abdominalis* (Gerhard, 1850) | **English name:** | Gerhard's Black Hairstreak |
| | **Georgian name:** | |
| | **Russian name:** | Хвостатка абдоминалис |
| | **Distr. Caucasus:** | Eastern part of the Great Caucasus, Lesser Caucasus, Djavakheti-Armenian plateau. |
| | **Distr. Georgia:** | Eastern part of the Great Caucasus, Lesser Caucasus, rare. |
| | **Flight:** | Univoltine, late May to early August. |
| | **Habitat:** | Rocky slopes, stony steppes, dry bushy hills. |
| 138. *Satyrium ilicis* (Esper, 1779) | **English name:** | Ilex Hairstreak |
| | **Georgian name:** | მუხის კუდალა |
| | **Russian name:** | Хвостатка падубовая |
| | **Distr. Caucasus:** | Widespread. |
| | **Distr. Georgia:** | Widespread and common. |
| | **Flight:** | Univoltine, June to August. |
| | **Habitat:** | Dry scrubs, gardens, forest clearings. |
| 139. *Satyrium acaciae* (Fabricius, 1787) | **English name:** | Sloe Hairstreak |
| | **Georgian name:** | აკაციის მრავალთვალა |
| | **Russian name:** | Хвостатка южная |
| | **Distr. Caucasus:** | Widespread. |
| | **Distr. Georgia:** | Eastern part of the Lesser Caucasus, rare. |
| | **Flight:** | Univoltine, May to July. |
| | **Habitat:** | Dry, bushy slopes and woodland clearings. |
| 140. *Satyrium ledereri* (Boisduval, 1848) | **English name:** | Orange-banded Hairstreak |
| | **Georgian name:** | ლედერერის ცრუკუდალა |
| | **Russian name:** | Хвостатка Ледерера |
| | **Distr. Caucasus:** | Central part of southern slopes of the Great Caucasus, central part of the Lesser Caucasus. |
| | **Distr. Georgia:** | Lesser Caucasus, rare. |
| | **Flight:** | Univoltine, May to early July depending on the altitude. |
| | **Habitat:** | Rocky slopes, foothills, semi-deserts. |

(Continued)

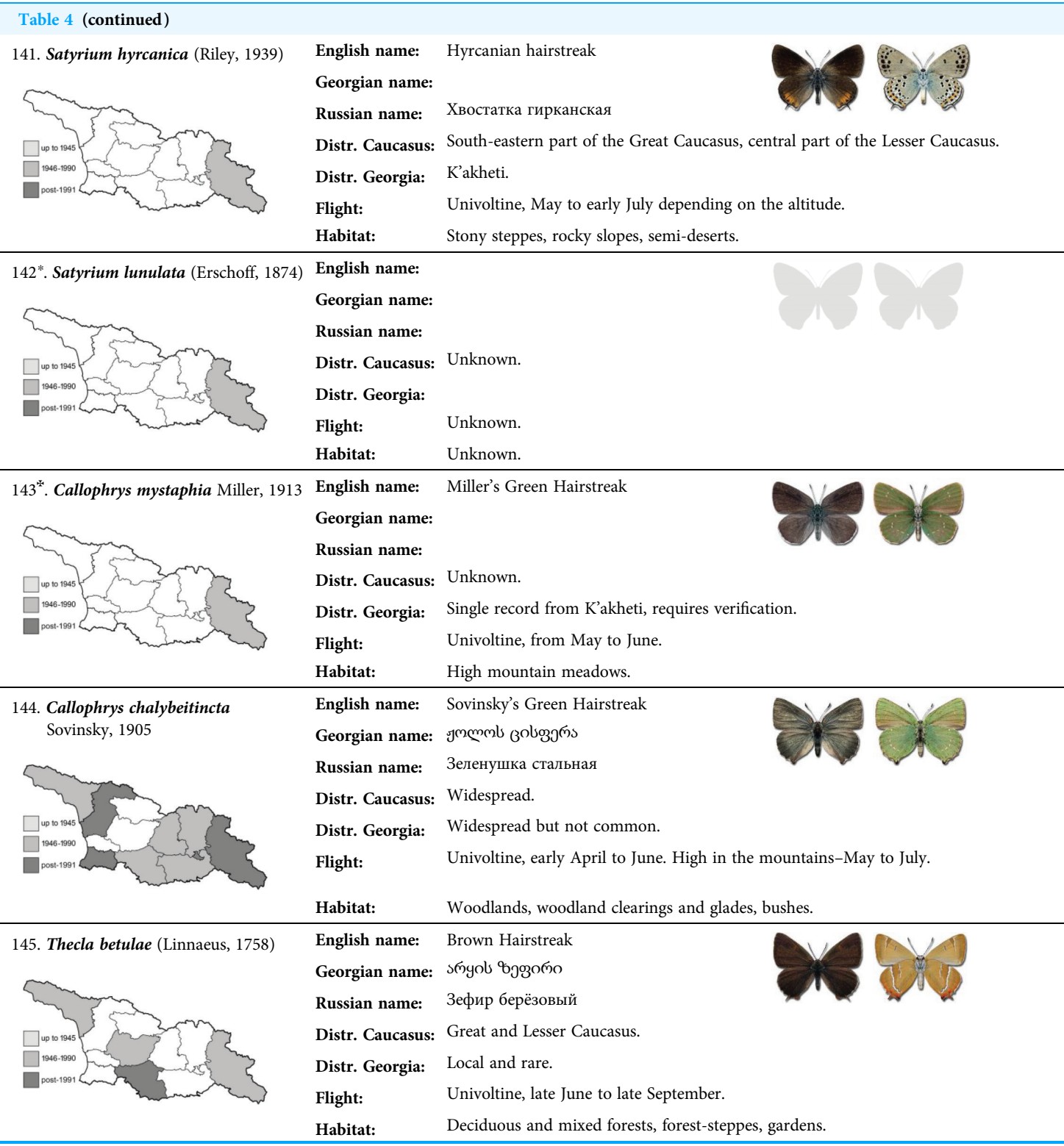

| | | |
|---|---|---|
| 141. *Satyrium hyrcanica* (Riley, 1939) | **English name:** | Hyrcanian hairstreak |
| | **Georgian name:** | |
| | **Russian name:** | Хвостатка гирканская |
| | **Distr. Caucasus:** | South-eastern part of the Great Caucasus, central part of the Lesser Caucasus. |
| | **Distr. Georgia:** | K'akheti. |
| | **Flight:** | Univoltine, May to early July depending on the altitude. |
| | **Habitat:** | Stony steppes, rocky slopes, semi-deserts. |
| 142*. *Satyrium lunulata* (Erschoff, 1874) | **English name:** | |
| | **Georgian name:** | |
| | **Russian name:** | |
| | **Distr. Caucasus:** | Unknown. |
| | **Distr. Georgia:** | |
| | **Flight:** | Unknown. |
| | **Habitat:** | Unknown. |
| 143*. *Callophrys mystaphia* Miller, 1913 | **English name:** | Miller's Green Hairstreak |
| | **Georgian name:** | |
| | **Russian name:** | |
| | **Distr. Caucasus:** | Unknown. |
| | **Distr. Georgia:** | Single record from K'akheti, requires verification. |
| | **Flight:** | Univoltine, from May to June. |
| | **Habitat:** | High mountain meadows. |
| 144. *Callophrys chalybeitincta* Sovinsky, 1905 | **English name:** | Sovinsky's Green Hairstreak |
| | **Georgian name:** | ქოლის ცისფერა |
| | **Russian name:** | Зеленушка стальная |
| | **Distr. Caucasus:** | Widespread. |
| | **Distr. Georgia:** | Widespread but not common. |
| | **Flight:** | Univoltine, early April to June. High in the mountains–May to July. |
| | **Habitat:** | Woodlands, woodland clearings and glades, bushes. |
| 145. *Thecla betulae* (Linnaeus, 1758) | **English name:** | Brown Hairstreak |
| | **Georgian name:** | არყის ზეფირი |
| | **Russian name:** | Зефир берёзовый |
| | **Distr. Caucasus:** | Great and Lesser Caucasus. |
| | **Distr. Georgia:** | Local and rare. |
| | **Flight:** | Univoltine, late June to late September. |
| | **Habitat:** | Deciduous and mixed forests, forest-steppes, gardens. |

| | | |
|---|---|---|
| **146. *Favonius quercus*** (Linnaeus, 1758) | **English name:** | Purple Hairstreak |
| | **Georgian name:** | მუხის ზეფირი |
| | **Russian name:** | Зефир дубовый |
| | **Distr. Caucasus:** | Widespread. |
| | **Distr. Georgia:** | Abkhazia and eastern part of the Lesser Caucasus, rare. |
| | **Flight:** | Univoltine, June to mid-August. |
| | **Habitat:** | Oak woodlands. |
| **147. *Tomares callimachus*** (Eversmann, 1848) | **English name:** | Eversmann's Hairstreak |
| | **Georgian name:** | |
| | **Russian name:** | Томарес Каллимах |
| | **Distr. Caucasus:** | Extreme north-western and south-eastern part of the Great Caucasus, Lesser Caucasus. |
| | **Distr. Georgia:** | Lesser Caucasus, extremely rare. |
| | **Flight:** | Univoltine, mid-March to mid-May depending on the altitude. |
| | **Habitat:** | Dry, stony calcareous hills, rocky stream valleys. |
| **148\*. *Tomares romanovi*** (Christoph, 1882) | **English name:** | Romanoff's Tomares |
| | **Georgian name:** | |
| | **Russian name:** | Томарес Романова |
| | **Distr. Caucasus:** | Djavakheti-Armenian plateau. |
| | **Distr. Georgia:** | Vashlovani National Park (K'akheti), extremely rare. |
| | **Flight:** | Univoltine, mid-April to late June. |
| | **Habitat:** | Stony gullies, hills and river valleys. |
| **Nymphalidae**, Swainson, 1827 | **Limenitidinae**, Behr, 1864 | |
| **149. *Limenitis populi*** (Linnaeus, 1758) | **English name:** | Poplar Admiral |
| | **Georgian name:** | |
| | **Russian name:** | Ленточница тополевая |
| | **Distr. Caucasus:** | North-Eastern foothills of Great Caucasus. |
| | **Distr. Georgia:** | Single record from vicinity of Sukhumi (Abkhasia). |
| | **Flight:** | Univoltine, from June to July. |
| | **Habitat:** | Deciduous forests glades and clearings. |
| **150. *Limenitis camilla*** (Linnaeus, 1764) | **English name:** | White Admiral |
| | **Georgian name:** | ლურჯი ლენტურა |
| | **Russian name:** | Ленточник Камилла |
| | **Distr. Caucasus:** | Great and Lesser Caucasus. |
| | **Distr. Georgia:** | Widespread and common. |
| | **Flight:** | Uni- or bivoltine, May to mid-August depending on the altitude. |
| | **Habitat:** | Woodland clearings and glades, forest-steppes. |

(Continued)

| | | |
|---|---|---|
| **151. *Limenitis reducta* Staudinger, 1901**  | **English name:** | Southern White Admiral |
| | **Georgian name:** | მცირე ლენტურა |
| | **Russian name:** | Ленточник голубоватый |
| | **Distr. Caucasus:** | Widespread. |
| | **Distr. Georgia:** | Widespread but not frequent. |
| | **Flight:** | Uni- or bivoltine, May to October. |
| | **Habitat:** | Deciduous forests glades and clearings, bushy and cultivated areas. |

| | | |
|---|---|---|
| **152. *Neptis rivularis* (Scopoli, 1763)**  | **English name:** | Hungarian Glider |
| | **Georgian name:** | გრაკლას ჭრელა ლენტურა |
| | **Russian name:** | Пеструшка таволговая |
| | **Distr. Caucasus:** | Widespread. |
| | **Distr. Georgia:** | Widespread but not common. |
| | **Flight:** | Univoltine, May to early September. |
| | **Habitat:** | Woodland clearings and glades. |

**Nymphalidae**, Swainson, 1827     **Heliconiinae**, Swainson, 1822

| | | |
|---|---|---|
| **153\*. *Boloria caucasica* (Lederer, 1852)**  | **English name:** | Caucasian Fritillary |
| | **Georgian name:** | |
| | **Russian name:** | Перламутровка кавказская |
| | **Distr. Caucasus:** | Great Caucasus and Lesser Caucasus. |
| | **Distr. Georgia:** | Widespread in the mountains range, common in the subalpine zone. |
| | **Flight:** | Univoltine, late June to early September depending on the altitude. |
| | **Habitat:** | Alpine and subalpine meadows. |

| | | |
|---|---|---|
| **154. *Boloria eunomia* (Esper, 1800)**  | **English name:** | Bog Fritillary |
| | **Georgian name:** | |
| | **Russian name:** | Перламутровка бледная |
| | **Distr. Caucasus:** | Western part of the Great Caucasus and Lesser Caucasus. |
| | **Distr. Georgia:** | Samtskhe-Javakheti, only a few individuals. |
| | **Flight:** | Univoltine, mid-May to July. |
| | **Habitat:** | Mountain meadows and swamps. |

| | | |
|---|---|---|
| **155. *Boloria euphrosyne* (Linnaeus, 1758)**  | **English name:** | Pearl-bordered Fritillary |
| | **Georgian name:** | |
| | **Russian name:** | Перламутровка Эвфросина |
| | **Distr. Caucasus:** | Great Caucasus, western and central part of the Lesser Caucasus. |
| | **Distr. Georgia:** | Widespread and common. |
| | **Flight:** | Bivoltine in the lowlands, late April to September. Univoltine in the mountains, May to July. |
| | **Habitat:** | Woodland clearings and glades. |

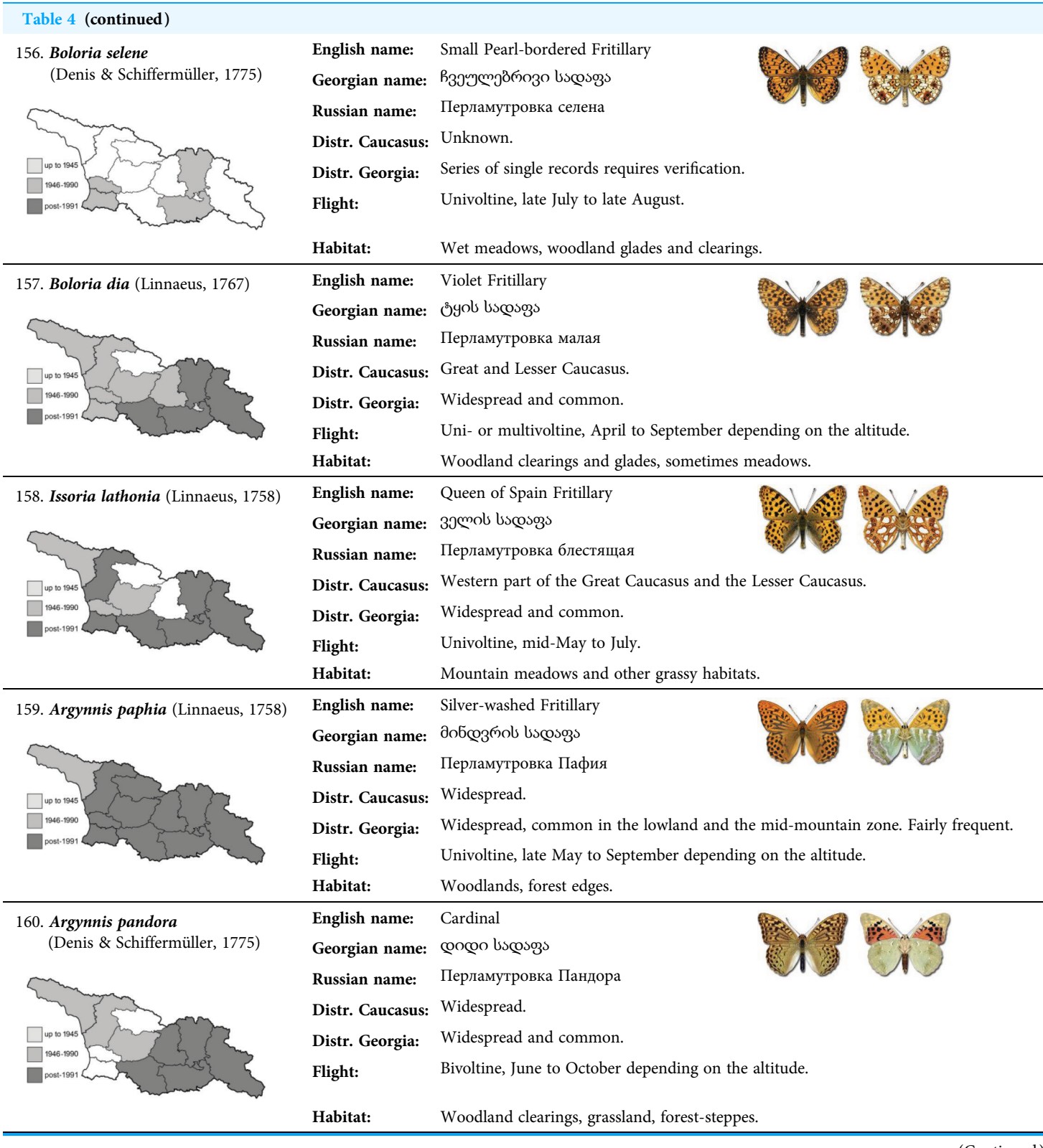

**156. *Boloria selene*** (Denis & Schiffermüller, 1775)

| | |
|---|---|
| English name: | Small Pearl-bordered Fritillary |
| Georgian name: | ჩვეულებრივი სადაფა |
| Russian name: | Перламутровка селена |
| Distr. Caucasus: | Unknown. |
| Distr. Georgia: | Series of single records requires verification. |
| Flight: | Univoltine, late July to late August. |
| Habitat: | Wet meadows, woodland glades and clearings. |

**157. *Boloria dia*** (Linnaeus, 1767)

| | |
|---|---|
| English name: | Violet Fritillary |
| Georgian name: | ტყის სადაფა |
| Russian name: | Перламутровка малая |
| Distr. Caucasus: | Great and Lesser Caucasus. |
| Distr. Georgia: | Widespread and common. |
| Flight: | Uni- or multivoltine, April to September depending on the altitude. |
| Habitat: | Woodland clearings and glades, sometimes meadows. |

**158. *Issoria lathonia*** (Linnaeus, 1758)

| | |
|---|---|
| English name: | Queen of Spain Fritillary |
| Georgian name: | ველის სადაფა |
| Russian name: | Перламутровка блестящая |
| Distr. Caucasus: | Western part of the Great Caucasus and the Lesser Caucasus. |
| Distr. Georgia: | Widespread and common. |
| Flight: | Univoltine, mid-May to July. |
| Habitat: | Mountain meadows and other grassy habitats. |

**159. *Argynnis paphia*** (Linnaeus, 1758)

| | |
|---|---|
| English name: | Silver-washed Fritillary |
| Georgian name: | მინდვრის სადაფა |
| Russian name: | Перламутровка Пафия |
| Distr. Caucasus: | Widespread. |
| Distr. Georgia: | Widespread, common in the lowland and the mid-mountain zone. Fairly frequent. |
| Flight: | Univoltine, late May to September depending on the altitude. |
| Habitat: | Woodlands, forest edges. |

**160. *Argynnis pandora*** (Denis & Schiffermüller, 1775)

| | |
|---|---|
| English name: | Cardinal |
| Georgian name: | დიდი სადაფა |
| Russian name: | Перламутровка Пандора |
| Distr. Caucasus: | Widespread. |
| Distr. Georgia: | Widespread and common. |
| Flight: | Bivoltine, June to October depending on the altitude. |
| Habitat: | Woodland clearings, grassland, forest-steppes. |

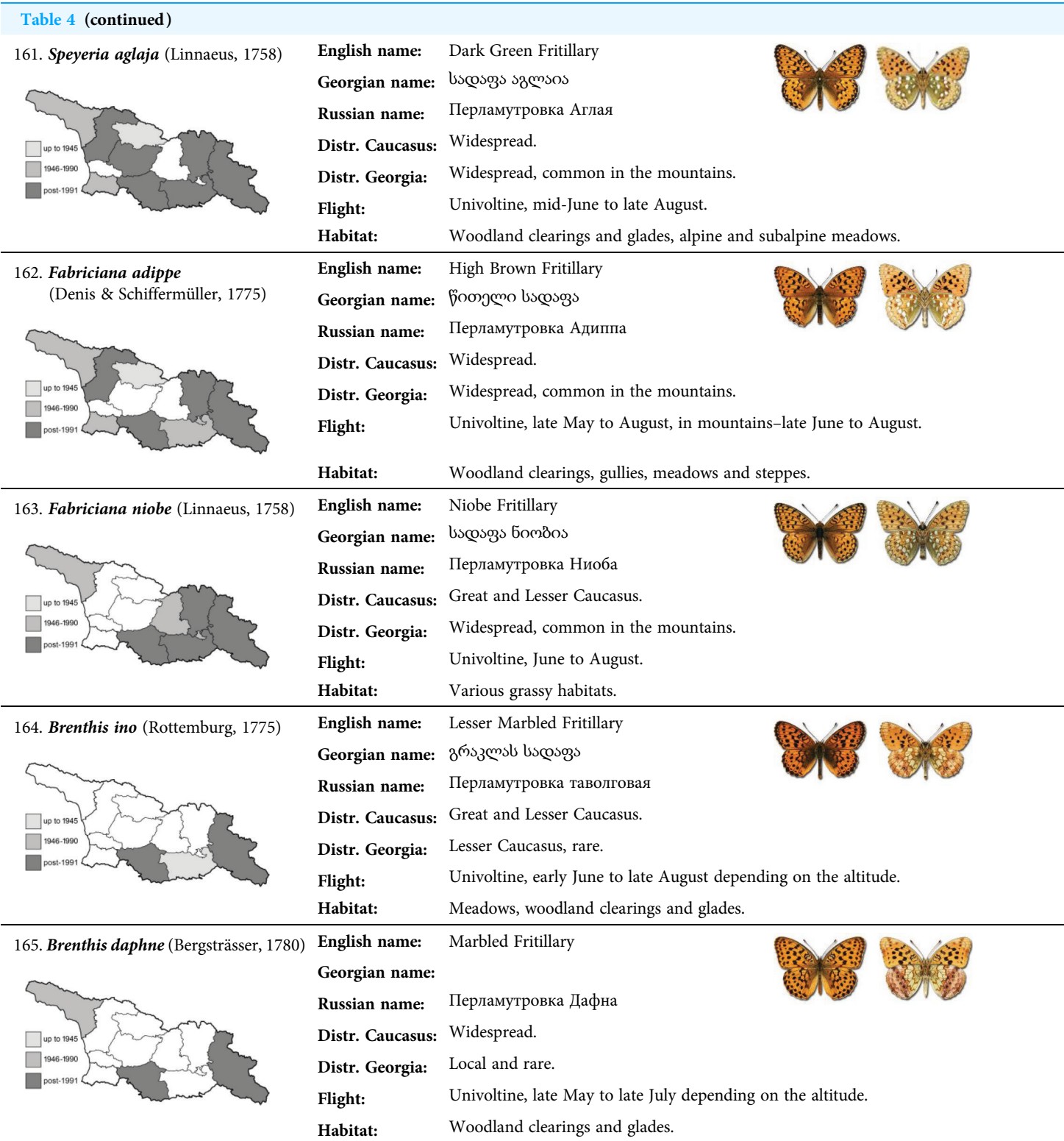

| | | |
|---|---|---|
| **161. *Speyeria aglaja* (Linnaeus, 1758)** | **English name:** | Dark Green Fritillary |
| | **Georgian name:** | სადაფა აგლაია |
| | **Russian name:** | Перламутровка Аглая |
| | **Distr. Caucasus:** | Widespread. |
| | **Distr. Georgia:** | Widespread, common in the mountains. |
| | **Flight:** | Univoltine, mid-June to late August. |
| | **Habitat:** | Woodland clearings and glades, alpine and subalpine meadows. |
| **162. *Fabriciana adippe* (Denis & Schiffermüller, 1775)** | **English name:** | High Brown Fritillary |
| | **Georgian name:** | წითელი სადაფა |
| | **Russian name:** | Перламутровка Адиппа |
| | **Distr. Caucasus:** | Widespread. |
| | **Distr. Georgia:** | Widespread, common in the mountains. |
| | **Flight:** | Univoltine, late May to August, in mountains–late June to August. |
| | **Habitat:** | Woodland clearings, gullies, meadows and steppes. |
| **163. *Fabriciana niobe* (Linnaeus, 1758)** | **English name:** | Niobe Fritillary |
| | **Georgian name:** | სადაფა ნიობია |
| | **Russian name:** | Перламутровка Ниоба |
| | **Distr. Caucasus:** | Great and Lesser Caucasus. |
| | **Distr. Georgia:** | Widespread, common in the mountains. |
| | **Flight:** | Univoltine, June to August. |
| | **Habitat:** | Various grassy habitats. |
| **164. *Brenthis ino* (Rottemburg, 1775)** | **English name:** | Lesser Marbled Fritillary |
| | **Georgian name:** | გრაკლას სადაფა |
| | **Russian name:** | Перламутровка таволговая |
| | **Distr. Caucasus:** | Great and Lesser Caucasus. |
| | **Distr. Georgia:** | Lesser Caucasus, rare. |
| | **Flight:** | Univoltine, early June to late August depending on the altitude. |
| | **Habitat:** | Meadows, woodland clearings and glades. |
| **165. *Brenthis daphne* (Bergsträsser, 1780)** | **English name:** | Marbled Fritillary |
| | **Georgian name:** | |
| | **Russian name:** | Перламутровка Дафна |
| | **Distr. Caucasus:** | Widespread. |
| | **Distr. Georgia:** | Local and rare. |
| | **Flight:** | Univoltine, late May to late July depending on the altitude. |
| | **Habitat:** | Woodland clearings and glades. |

| 166. *Brenthis hecate* (Denis & Schiffermüller, 1775) | | |
|---|---|---|
| | **English name:** | Twin-spot Fritillary |
| | **Georgian name:** | |
| | **Russian name:** | Перламутровка Геката |
| | **Distr. Caucasus:** | Widespread. |
| | **Distr. Georgia:** | Local and rare. |
| | **Flight:** | Univoltine, late May to mid-August depending on the altitude. |
| | **Habitat:** | Grassy slopes, meadows, woodland clearings. |

| **Nymphalidae**, Swainson, 1827 | **Apaturinae**, Boisduval, 1840 |
|---|---|

| 167. *Apatura ilia* (Denis & Schiffermüller, 1775) | | |
|---|---|---|
| | **English name:** | Lesser Purple Emperor |
| | **Georgian name:** | ილიას ფერცვალა |
| | **Russian name:** | Переливница малая |
| | **Distr. Caucasus:** | Western and central part of the Great Caucasus, Lesser Caucasus. |
| | **Distr. Georgia:** | Widespread but local, not rare. |
| | **Flight:** | Uni- or Bivoltine, late May to August depending on the altitude. |
| | **Habitat:** | Deciduous and mixed forests, sometimes gardens and parks. |

| 168. *Thaleropis ionia* Eversmann, 1851 | | |
|---|---|---|
| | **English name:** | Ionian Emperor |
| | **Georgian name:** | მზისავი იონია |
| | **Russian name:** | Переливница иония |
| | **Distr. Caucasus:** | Great Caucasus and Lesser Caucasus. |
| | **Distr. Georgia:** | Single record from vicinity of Kvariati (Ajaria). |
| | **Flight:** | Bi- or trivoltine, late May to September. |
| | **Habitat:** | Deciduous woodlands, river valleys up to 2,000 m. |

| **Nymphalidae**, Swainson, 1827 | **Nymphalinae**, Swainson, 1827 |
|---|---|

| 169. *Melitaea cinxia* (Linnaeus, 1758) | | |
|---|---|---|
| | **English name:** | Glanville Fritillary |
| | **Georgian name:** | ჩვეულებრივი კამათელა |
| | **Russian name:** | Шашечница обыкновенная |
| | **Distr. Caucasus:** | Widespread. |
| | **Distr. Georgia:** | Widespread and common. |
| | **Flight:** | Uni- or multivoltine, mid-May to late September depending on the altitude. |
| | **Habitat:** | Various grassy habitats. |

| 170. *Melitaea arduinna* (Esper, 1783) | | |
|---|---|---|
| | **English name:** | Freyer's Fritillary |
| | **Georgian name:** | |
| | **Russian name:** | Шашечница горная |
| | **Distr. Caucasus:** | Central part of the Great Caucasus, Lesser Caucasus. |
| | **Distr. Georgia:** | Local and rare. |
| | **Flight:** | Univoltine, May to August depending on the altitude. |
| | **Habitat:** | Woodland clearings and edges, dry meadows, slopes and gullies. |

*(Continued)*

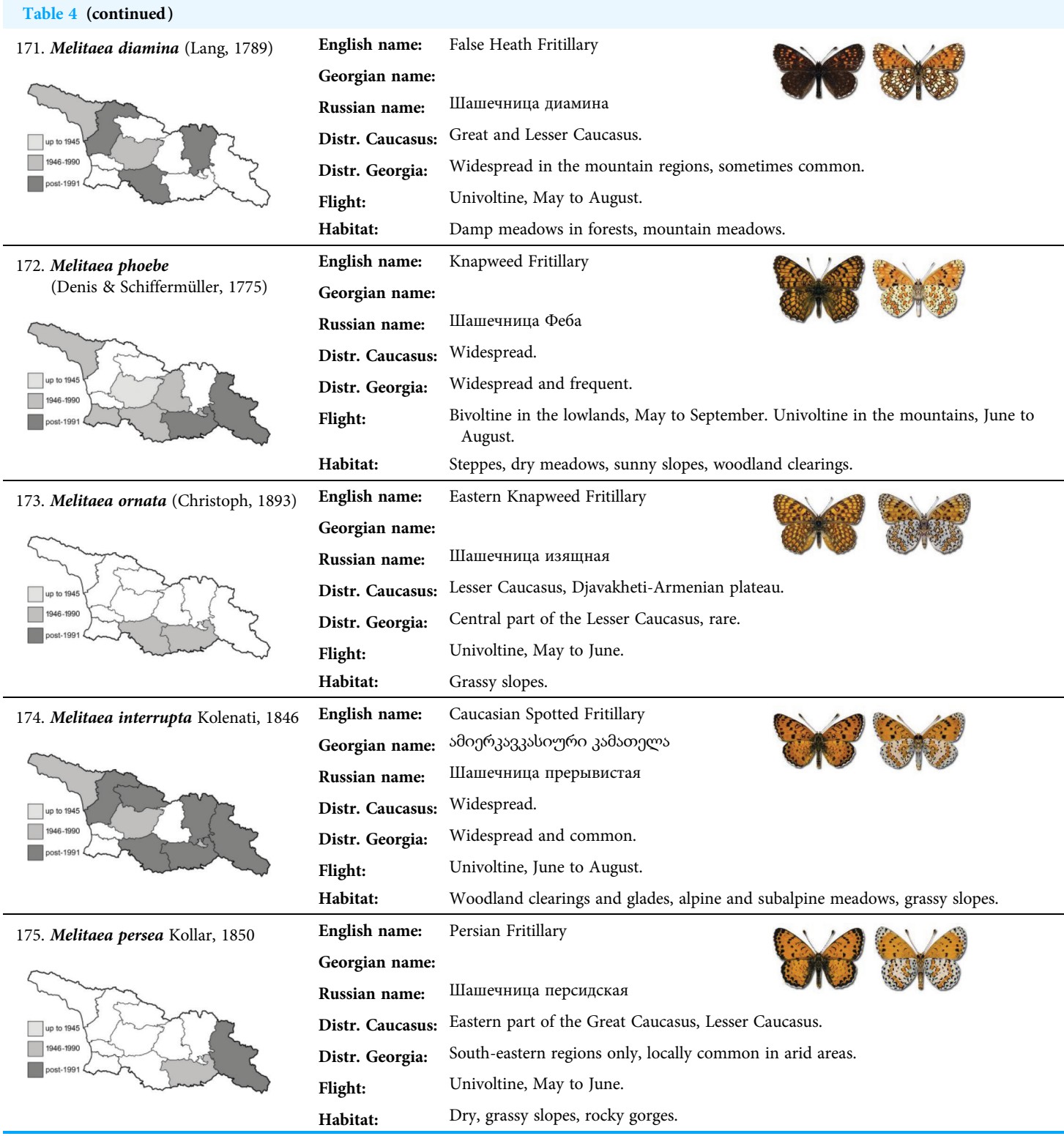

**171. *Melitaea diamina*** (Lang, 1789)

| | |
|---|---|
| **English name:** | False Heath Fritillary |
| **Georgian name:** | |
| **Russian name:** | Шашечница диамина |
| **Distr. Caucasus:** | Great and Lesser Caucasus. |
| **Distr. Georgia:** | Widespread in the mountain regions, sometimes common. |
| **Flight:** | Univoltine, May to August. |
| **Habitat:** | Damp meadows in forests, mountain meadows. |

**172. *Melitaea phoebe*** (Denis & Schiffermüller, 1775)

| | |
|---|---|
| **English name:** | Knapweed Fritillary |
| **Georgian name:** | |
| **Russian name:** | Шашечница Феба |
| **Distr. Caucasus:** | Widespread. |
| **Distr. Georgia:** | Widespread and frequent. |
| **Flight:** | Bivoltine in the lowlands, May to September. Univoltine in the mountains, June to August. |
| **Habitat:** | Steppes, dry meadows, sunny slopes, woodland clearings. |

**173. *Melitaea ornata*** (Christoph, 1893)

| | |
|---|---|
| **English name:** | Eastern Knapweed Fritillary |
| **Georgian name:** | |
| **Russian name:** | Шашечница изящная |
| **Distr. Caucasus:** | Lesser Caucasus, Djavakheti-Armenian plateau. |
| **Distr. Georgia:** | Central part of the Lesser Caucasus, rare. |
| **Flight:** | Univoltine, May to June. |
| **Habitat:** | Grassy slopes. |

**174. *Melitaea interrupta*** Kolenati, 1846

| | |
|---|---|
| **English name:** | Caucasian Spotted Fritillary |
| **Georgian name:** | ამიერკავკასიური კამათელა |
| **Russian name:** | Шашечница прерывистая |
| **Distr. Caucasus:** | Widespread. |
| **Distr. Georgia:** | Widespread and common. |
| **Flight:** | Univoltine, June to August. |
| **Habitat:** | Woodland clearings and glades, alpine and subalpine meadows, grassy slopes. |

**175. *Melitaea persea*** Kollar, 1850

| | |
|---|---|
| **English name:** | Persian Fritillary |
| **Georgian name:** | |
| **Russian name:** | Шашечница персидская |
| **Distr. Caucasus:** | Eastern part of the Great Caucasus, Lesser Caucasus. |
| **Distr. Georgia:** | South-eastern regions only, locally common in arid areas. |
| **Flight:** | Univoltine, May to June. |
| **Habitat:** | Dry, grassy slopes, rocky gorges. |

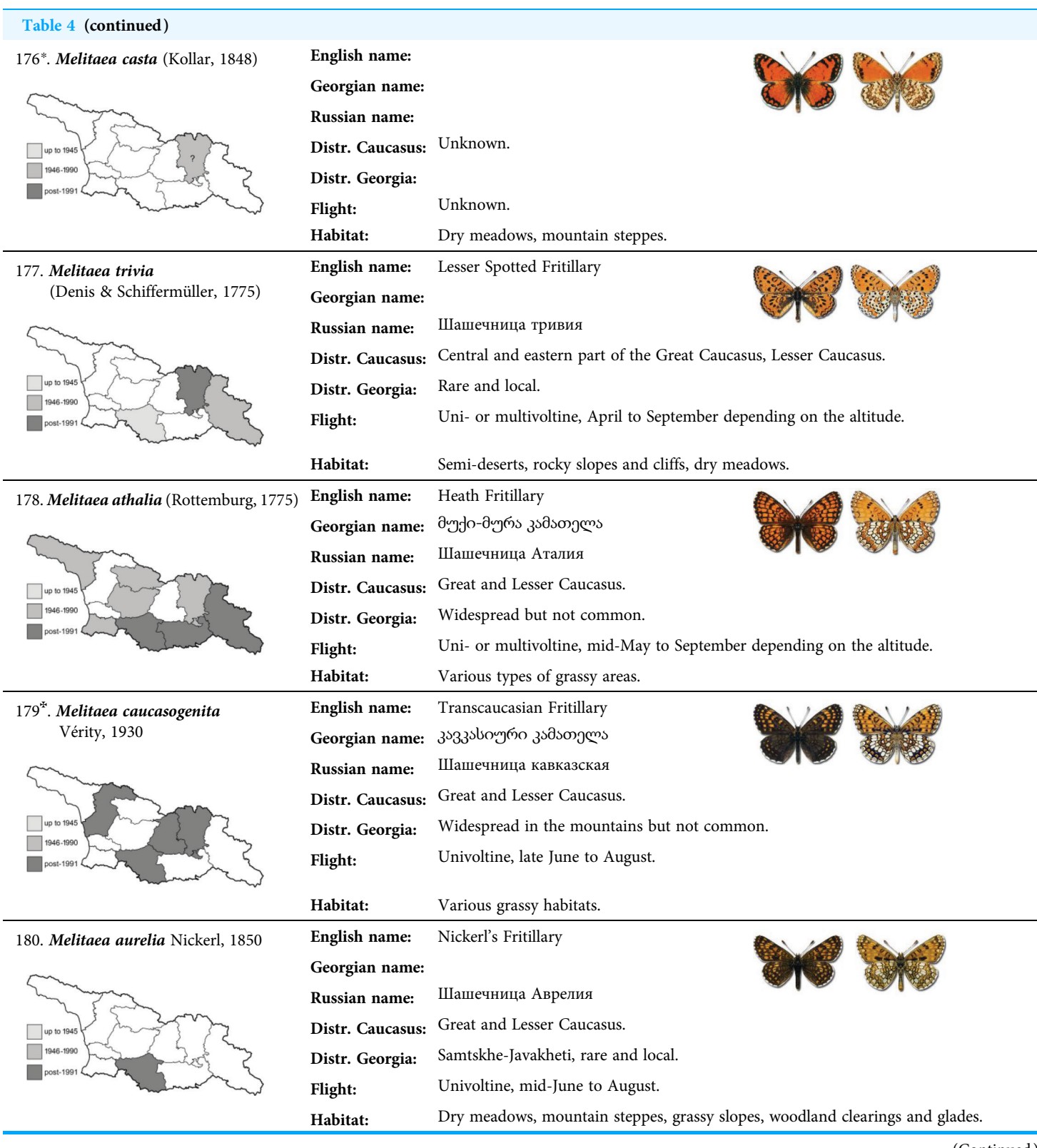

| | | |
|---|---|---|
| 176*. *Melitaea casta* (Kollar, 1848) | **English name:** | |
| | **Georgian name:** | |
| | **Russian name:** | |
| | **Distr. Caucasus:** | Unknown. |
| | **Distr. Georgia:** | |
| | **Flight:** | Unknown. |
| | **Habitat:** | Dry meadows, mountain steppes. |
| 177. *Melitaea trivia* (Denis & Schiffermüller, 1775) | **English name:** | Lesser Spotted Fritillary |
| | **Georgian name:** | |
| | **Russian name:** | Шашечница тривия |
| | **Distr. Caucasus:** | Central and eastern part of the Great Caucasus, Lesser Caucasus. |
| | **Distr. Georgia:** | Rare and local. |
| | **Flight:** | Uni- or multivoltine, April to September depending on the altitude. |
| | **Habitat:** | Semi-deserts, rocky slopes and cliffs, dry meadows. |
| 178. *Melitaea athalia* (Rottemburg, 1775) | **English name:** | Heath Fritillary |
| | **Georgian name:** | მუქი-მურა კამათელა |
| | **Russian name:** | Шашечница Аталия |
| | **Distr. Caucasus:** | Great and Lesser Caucasus. |
| | **Distr. Georgia:** | Widespread but not common. |
| | **Flight:** | Uni- or multivoltine, mid-May to September depending on the altitude. |
| | **Habitat:** | Various types of grassy areas. |
| 179*. *Melitaea caucasogenita* Vérity, 1930 | **English name:** | Transcaucasian Fritillary |
| | **Georgian name:** | კავკასიური კამათელა |
| | **Russian name:** | Шашечница кавказская |
| | **Distr. Caucasus:** | Great and Lesser Caucasus. |
| | **Distr. Georgia:** | Widespread in the mountains but not common. |
| | **Flight:** | Univoltine, late June to August. |
| | **Habitat:** | Various grassy habitats. |
| 180. *Melitaea aurelia* Nickerl, 1850 | **English name:** | Nickerl's Fritillary |
| | **Georgian name:** | |
| | **Russian name:** | Шашечница Аврелия |
| | **Distr. Caucasus:** | Great and Lesser Caucasus. |
| | **Distr. Georgia:** | Samtskhe-Javakheti, rare and local. |
| | **Flight:** | Univoltine, mid-June to August. |
| | **Habitat:** | Dry meadows, mountain steppes, grassy slopes, woodland clearings and glades. |

*(Continued)*

| Table 4 (continued) | | | |
|---|---|---|---|

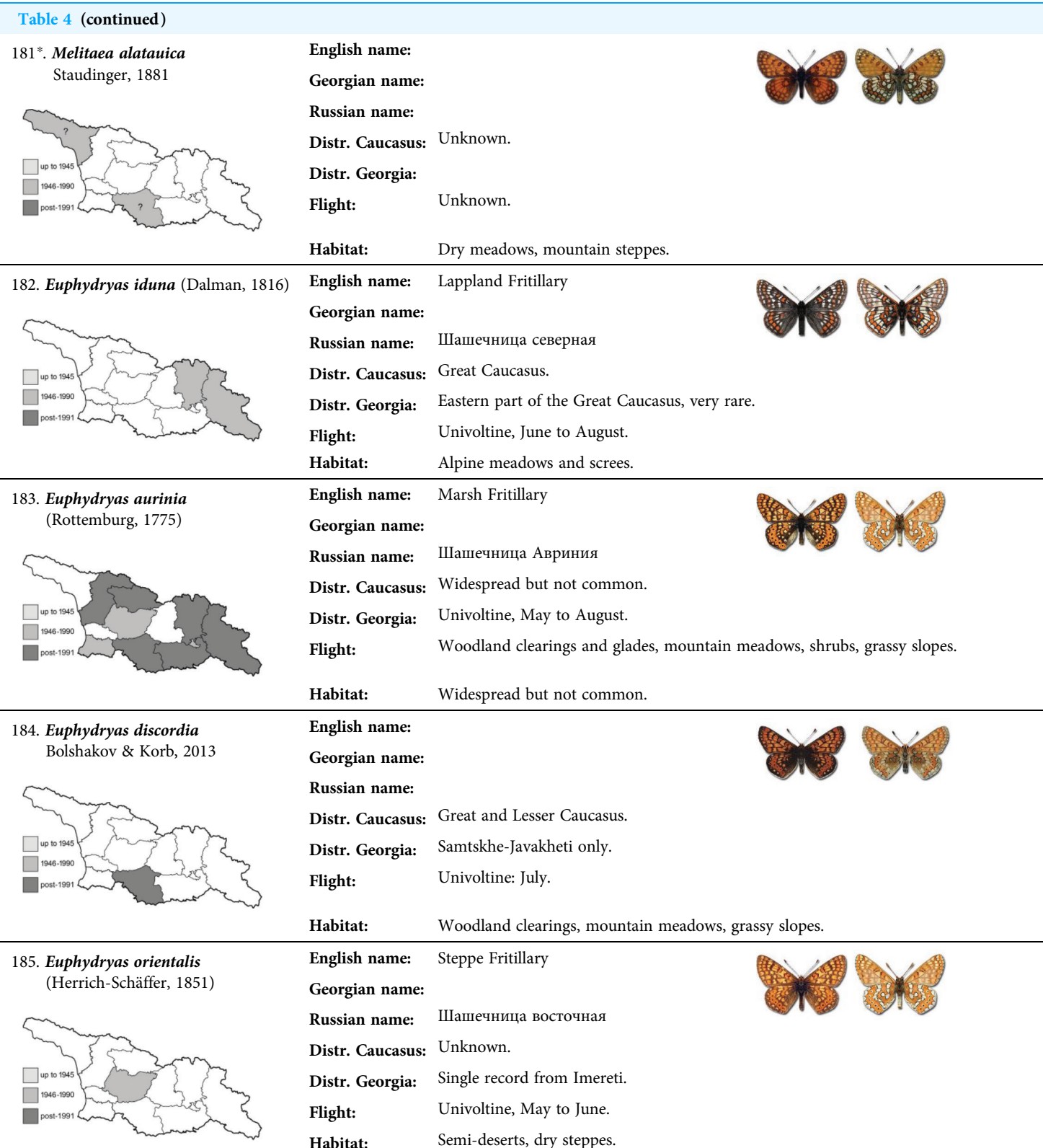

**181\*. *Melitaea alatauica*** Staudinger, 1881

English name:
Georgian name:
Russian name:
Distr. Caucasus: Unknown.
Distr. Georgia:
Flight: Unknown.
Habitat: Dry meadows, mountain steppes.

**182. *Euphydryas iduna*** (Dalman, 1816)

English name: Lappland Fritillary
Georgian name:
Russian name: Шашечница северная
Distr. Caucasus: Great Caucasus.
Distr. Georgia: Eastern part of the Great Caucasus, very rare.
Flight: Univoltine, June to August.
Habitat: Alpine meadows and screes.

**183. *Euphydryas aurinia*** (Rottemburg, 1775)

English name: Marsh Fritillary
Georgian name:
Russian name: Шашечница Авриния
Distr. Caucasus: Widespread but not common.
Distr. Georgia: Univoltine, May to August.
Flight: Woodland clearings and glades, mountain meadows, shrubs, grassy slopes.
Habitat: Widespread but not common.

**184. *Euphydryas discordia*** Bolshakov & Korb, 2013

English name:
Georgian name:
Russian name:
Distr. Caucasus: Great and Lesser Caucasus.
Distr. Georgia: Samtskhe-Javakheti only.
Flight: Univoltine: July.
Habitat: Woodland clearings, mountain meadows, grassy slopes.

**185. *Euphydryas orientalis*** (Herrich-Schäffer, 1851)

English name: Steppe Fritillary
Georgian name:
Russian name: Шашечница восточная
Distr. Caucasus: Unknown.
Distr. Georgia: Single record from Imereti.
Flight: Univoltine, May to June.
Habitat: Semi-deserts, dry steppes.

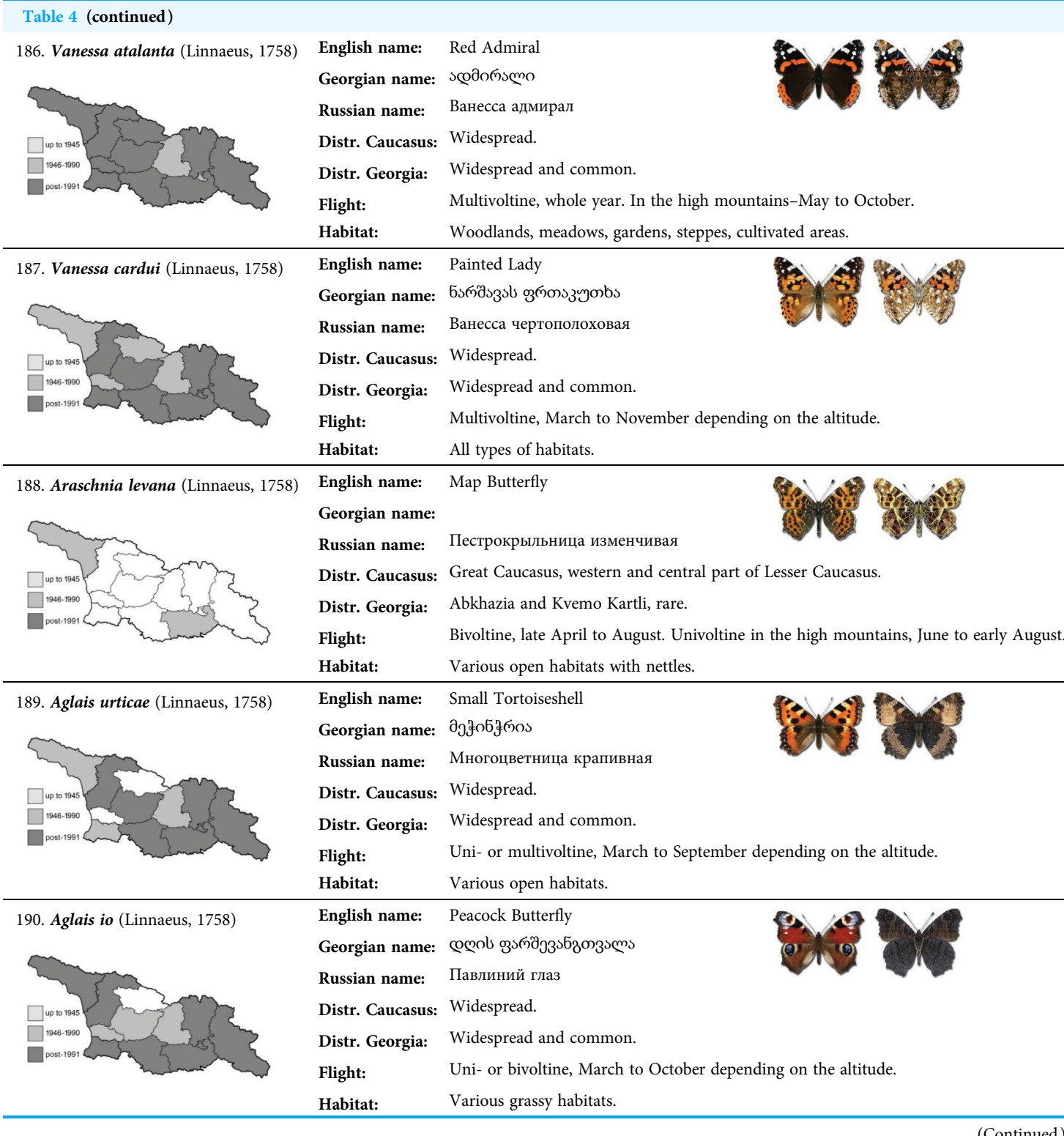

| 186. *Vanessa atalanta* (Linnaeus, 1758) | | |
|---|---|---|
| **English name:** | Red Admiral | |
| **Georgian name:** | ადმირალი | |
| **Russian name:** | Ванесса адмирал | |
| **Distr. Caucasus:** | Widespread. | |
| **Distr. Georgia:** | Widespread and common. | |
| **Flight:** | Multivoltine, whole year. In the high mountains–May to October. | |
| **Habitat:** | Woodlands, meadows, gardens, steppes, cultivated areas. | |
| **187. *Vanessa cardui* (Linnaeus, 1758)** | | |
| **English name:** | Painted Lady | |
| **Georgian name:** | ნარშავას ფრთაკუთხა | |
| **Russian name:** | Ванесса чертополоховая | |
| **Distr. Caucasus:** | Widespread. | |
| **Distr. Georgia:** | Widespread and common. | |
| **Flight:** | Multivoltine, March to November depending on the altitude. | |
| **Habitat:** | All types of habitats. | |
| **188. *Araschnia levana* (Linnaeus, 1758)** | | |
| **English name:** | Map Butterfly | |
| **Georgian name:** | | |
| **Russian name:** | Пестрокрыльница изменчивая | |
| **Distr. Caucasus:** | Great Caucasus, western and central part of Lesser Caucasus. | |
| **Distr. Georgia:** | Abkhazia and Kvemo Kartli, rare. | |
| **Flight:** | Bivoltine, late April to August. Univoltine in the high mountains, June to early August. | |
| **Habitat:** | Various open habitats with nettles. | |
| **189. *Aglais urticae* (Linnaeus, 1758)** | | |
| **English name:** | Small Tortoiseshell | |
| **Georgian name:** | მეჯინჭრია | |
| **Russian name:** | Многоцветница крапивная | |
| **Distr. Caucasus:** | Widespread. | |
| **Distr. Georgia:** | Widespread and common. | |
| **Flight:** | Uni- or multivoltine, March to September depending on the altitude. | |
| **Habitat:** | Various open habitats. | |
| **190. *Aglais io* (Linnaeus, 1758)** | | |
| **English name:** | Peacock Butterfly | |
| **Georgian name:** | დღის ფარშევანგთვალა | |
| **Russian name:** | Павлиний глаз | |
| **Distr. Caucasus:** | Widespread. | |
| **Distr. Georgia:** | Widespread and common. | |
| **Flight:** | Uni- or bivoltine, March to October depending on the altitude. | |
| **Habitat:** | Various grassy habitats. | |

| 191. *Nymphalis antiopa* (Linnaeus, 1758) | | |
|---|---|---|
| | **English name:** | Camberwell Beauty |
| | **Georgian name:** | ძაძანა |
| | **Russian name:** | Многоцветница траурная |
| | **Distr. Caucasus:** | Widespread. |
| | **Distr. Georgia:** | Widespread, common in coniferous forest range. |
| | **Flight:** | Univoltine, April to September. |
| | **Habitat:** | Deciduous and mixed forests clearings and edges, river banks. |

| 192. *Nymphalis polychloros* (Linnaeus, 1758) | | |
|---|---|---|
| | **English name:** | Large Tortoiseshell |
| | **Georgian name:** | მსხლის მრავალფერა |
| | **Russian name:** | Многоцветница обыкновенная |
| | **Distr. Caucasus:** | Widespread. |
| | **Distr. Georgia:** | Widespread and common. |
| | **Flight:** | Univoltine, May to October. |
| | **Habitat:** | Deciduous and mixed forests, river banks, sometimes gardens. |

| 193. *Nymphalis xanthomelas* (Denis & Schiffermüller, 1775) | | |
|---|---|---|
| | **English name:** | Yellow-legged-Tortoiseshell |
| | **Georgian name:** | შავყვითელა ვანესა |
| | **Russian name:** | Многоцветница чёрно-рыжая |
| | **Distr. Caucasus:** | Great and Lesser Caucasus. |
| | **Distr. Georgia:** | Local and rare. |
| | **Flight:** | Univoltine, May to October. |
| | **Habitat:** | Deciduous and mixed forests, river banks. |

| 194. *Nymphalis vaualbum* (Denis & Schiffermüller, 1775) | | |
|---|---|---|
| | **English name:** | False Comma |
| | **Georgian name:** | ფრთაკუთხა L-თეთრი |
| | **Russian name:** | Многоцветница v-белое |
| | **Distr. Caucasus:** | Western and central part of the Great Caucasus, Lesser Caucasus. |
| | **Distr. Georgia:** | Extremely rare. |
| | **Flight:** | Univoltine, from March to September. |
| | **Habitat:** | Deciduous and mixed forests. |

| 195. *Polygonia egea* (Cramer, 1775) | | |
|---|---|---|
| | **English name:** | Southern Comma |
| | **Georgian name:** | |
| | **Russian name:** | Углокрыльница южная |
| | **Distr. Caucasus:** | Widespread. |
| | **Distr. Georgia:** | Local and rare. |
| | **Flight:** | Uni- or bivoltine, March to October. |
| | **Habitat:** | Deciduous and mixed forests clearings and glades, scrubs. |

| 196. *Polygonia c-album* (Linnaeus, 1758) | | |
|---|---|---|
| | **English name:** | Comma |
| | **Georgian name:** | ფრთაკუთხა C-თეთრი |
| | **Russian name:** | Углокрыльница с-белое |
| | **Distr. Caucasus:** | Widespread. |
| | **Distr. Georgia:** | Widespread and common. |
| | **Flight:** | Uni- or multivoltine, March to October depending on the altitude. |
| | **Habitat:** | Woodland, gardens. |

**Nymphalidae**, Swainson, 1827 — **Satyrinae**, Boisduval, 1833

| 197. *Coenonympha pamphilus* (Linnaeus, 1758) | | |
|---|---|---|
| | **English name:** | Small Heath |
| | **Georgian name:** | მეთივია პამფილი |
| | **Russian name:** | Сенница памфил |
| | **Distr. Caucasus:** | Widespread. |
| | **Distr. Georgia:** | Widespread and common. |
| | **Flight:** | Multivoltine, March to November. |
| | **Habitat:** | Various grassy habitats. |

| 198. *Coenonympha tullia* (Müller, 1764) | | |
|---|---|---|
| | **English name:** | Large Heath |
| | **Georgian name:** | |
| | **Russian name:** | Сенница туллия |
| | **Distr. Caucasus:** | Great Caucasus. |
| | **Distr. Georgia:** | Abkhazia, very rare. |
| | **Flight:** | Univoltine, late June to August. |
| | **Habitat:** | Alpine and subalpine meadows from forest belt to nival zone. |

| 199. *Coenonympha glycerion* (Borkhausen, 1788) | | |
|---|---|---|
| | **English name:** | Chestnut Heath |
| | **Georgian name:** | |
| | **Russian name:** | Сенница глицерион |
| | **Distr. Caucasus:** | Great and Lesser Caucasus. |
| | **Distr. Georgia:** | Eastern regions only, not common. |
| | **Flight:** | Univoltine, late May to August. |
| | **Habitat:** | Various grassy habitats: meadows, forest edges and clearings. |

| 200. *Coenonympha arcania* (Linnaeus, 1760) | | |
|---|---|---|
| | **English name:** | Pearly Heath |
| | **Georgian name:** | თეთრზოლიანი მეთივია |
| | **Russian name:** | Сенница аркания |
| | **Distr. Caucasus:** | Widespread. |
| | **Distr. Georgia:** | Widespread but not common. |
| | **Flight:** | Univoltine, mid-May to mid-August depending on the altitude. |
| | **Habitat:** | Clearings and glades in deciduous and mixed forests, grassy slopes, scrubs. |

(Continued)

| 201. *Coenonympha leander* (Esper, 1784) | | |
|---|---|---|
| **English name:** | Russian Heath | |
| **Georgian name:** | | |
| **Russian name:** | Сенница леандр | |
| **Distr. Caucasus:** | Central and eastern part of the Great Caucasus, Lesser Caucasus. | |
| **Distr. Georgia:** | Local and rare. | |
| **Flight:** | Univoltine, late May to August. | |
| **Habitat:** | Various grassy habitats: meadows, forest edges and clearings. | |

| 202*. *Coenonympha saadi* Kollar, 1849 | | |
|---|---|---|
| **English name:** | Saadi's Heath | |
| **Georgian name:** | | |
| **Russian name:** | Сенница Саади | |
| **Distr. Caucasus:** | Central part of southern slopes of the Great Caucasus, southern part of the Lesser Caucasus. | |
| **Distr. Georgia:** | Local and rare. | |
| **Flight:** | Univoltine, May to July. | |
| **Habitat:** | Deserts and semi-deserts, arid foothills. | |

| 203*. *Coenonympha symphita* Lederer, 1870 | | |
|---|---|---|
| **English name:** | Caucasian Heath | |
| **Georgian name:** | | |
| **Russian name:** | Сенница альпийская | |
| **Distr. Caucasus:** | Lesser Caucasus and Djavakheti-Armenian plateau. | |
| **Distr. Georgia:** | Lesser Caucasus, rare. | |
| **Flight:** | Univoltine, mid-June to mid-August. | |
| **Habitat:** | Meadows and damp steppes in forest belt, alpine and subalpine meadows. | |

| 204. *Kirinia climene* (Esper, 1783) | | |
|---|---|---|
| **English name:** | Lesser Lattice Brown | |
| **Georgian name:** | | |
| **Russian name:** | Бархатница Климена | |
| **Distr. Caucasus:** | Widespread. | |
| **Distr. Georgia:** | Local and extremely rare. | |
| **Flight:** | Univoltine, June to August. | |
| **Habitat:** | Forest edges and clearings, scrubby gullies, river valleys. | |

| 205. *Pararge aegeria* (Linnaeus, 1758) | | |
|---|---|---|
| **English name:** | Speckled Wood | |
| **Georgian name:** | ტყის მურათვალა | |
| **Russian name:** | Краеглазка эгерия | |
| **Distr. Caucasus:** | Widespread. | |
| **Distr. Georgia:** | Widespread and common. | |
| **Flight:** | Bi- or trivoltine, April to October depending on the altitude. | |
| **Habitat:** | Woodlands, parks, gardens. | |

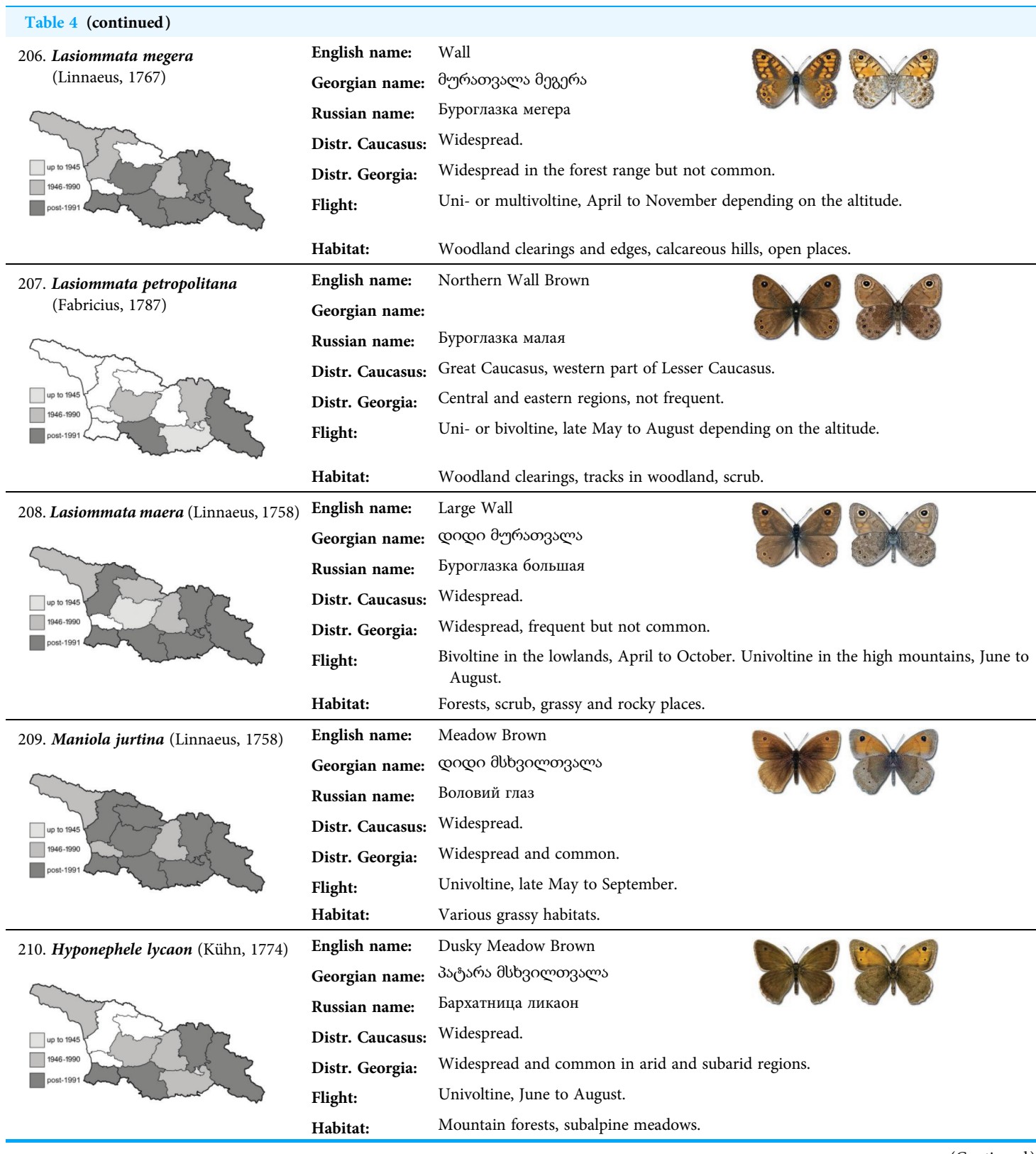

| | | |
|---|---|---|
| 206. *Lasiommata megera* (Linnaeus, 1767) | **English name:** | Wall |
| | **Georgian name:** | მურათვალა მეგერა |
| | **Russian name:** | Буроглазка мегера |
| | **Distr. Caucasus:** | Widespread. |
| | **Distr. Georgia:** | Widespread in the forest range but not common. |
| | **Flight:** | Uni- or multivoltine, April to November depending on the altitude. |
| | **Habitat:** | Woodland clearings and edges, calcareous hills, open places. |
| 207. *Lasiommata petropolitana* (Fabricius, 1787) | **English name:** | Northern Wall Brown |
| | **Georgian name:** | |
| | **Russian name:** | Буроглазка малая |
| | **Distr. Caucasus:** | Great Caucasus, western part of Lesser Caucasus. |
| | **Distr. Georgia:** | Central and eastern regions, not frequent. |
| | **Flight:** | Uni- or bivoltine, late May to August depending on the altitude. |
| | **Habitat:** | Woodland clearings, tracks in woodland, scrub. |
| 208. *Lasiommata maera* (Linnaeus, 1758) | **English name:** | Large Wall |
| | **Georgian name:** | დიდი მურათვალა |
| | **Russian name:** | Буроглазка большая |
| | **Distr. Caucasus:** | Widespread. |
| | **Distr. Georgia:** | Widespread, frequent but not common. |
| | **Flight:** | Bivoltine in the lowlands, April to October. Univoltine in the high mountains, June to August. |
| | **Habitat:** | Forests, scrub, grassy and rocky places. |
| 209. *Maniola jurtina* (Linnaeus, 1758) | **English name:** | Meadow Brown |
| | **Georgian name:** | დიდი მსხვილთვალა |
| | **Russian name:** | Воловий глаз |
| | **Distr. Caucasus:** | Widespread. |
| | **Distr. Georgia:** | Widespread and common. |
| | **Flight:** | Univoltine, late May to September. |
| | **Habitat:** | Various grassy habitats. |
| 210. *Hyponephele lycaon* (Kühn, 1774) | **English name:** | Dusky Meadow Brown |
| | **Georgian name:** | პატარა მსხვილთვალა |
| | **Russian name:** | Бархатница ликаон |
| | **Distr. Caucasus:** | Widespread. |
| | **Distr. Georgia:** | Widespread and common in arid and subarid regions. |
| | **Flight:** | Univoltine, June to August. |
| | **Habitat:** | Mountain forests, subalpine meadows. |

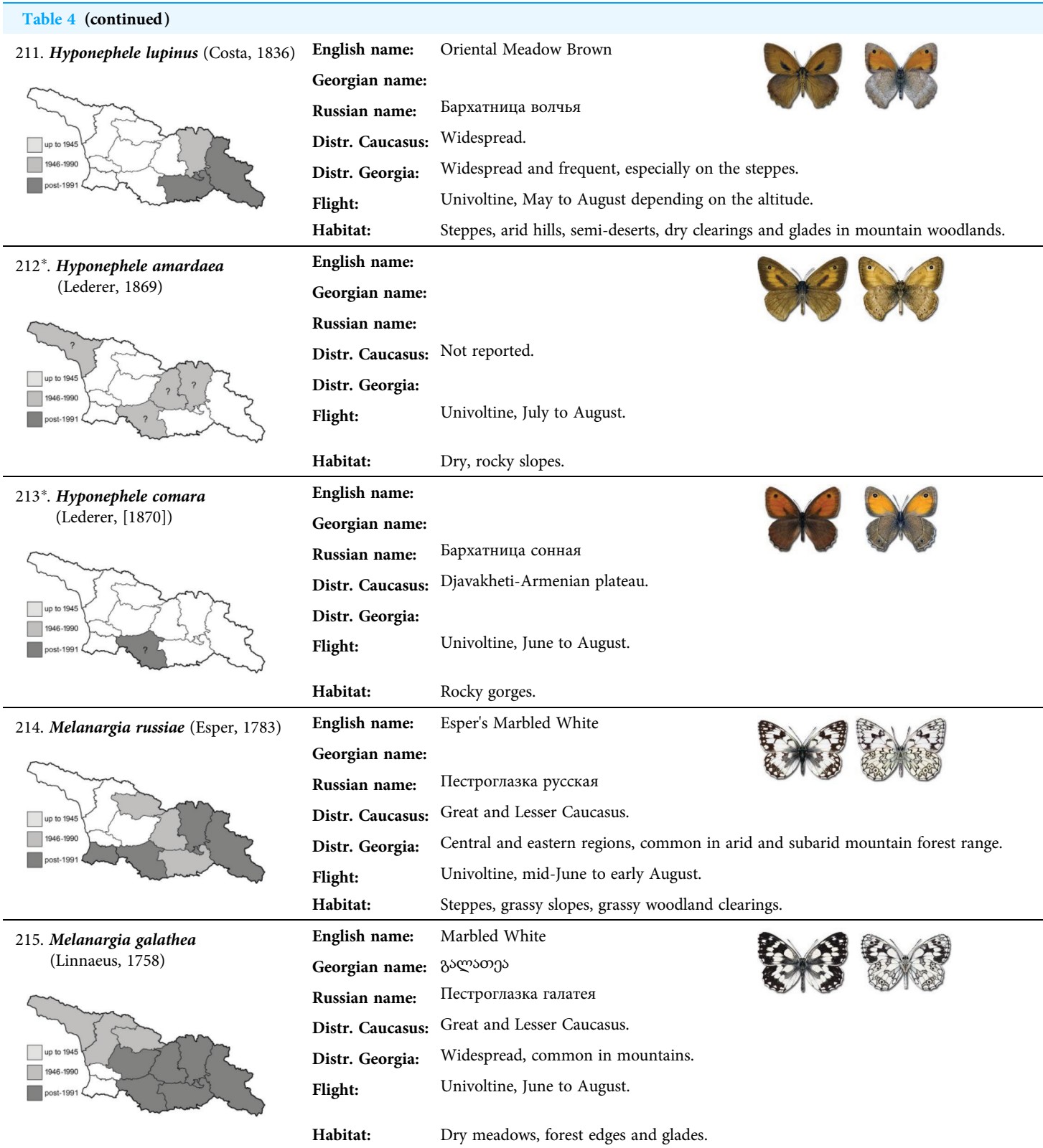

| 211. *Hyponephele lupinus* (Costa, 1836) | | |
|---|---|---|
| **English name:** | Oriental Meadow Brown |
| **Georgian name:** | |
| **Russian name:** | Бархатница волчья |
| **Distr. Caucasus:** | Widespread. |
| **Distr. Georgia:** | Widespread and frequent, especially on the steppes. |
| **Flight:** | Univoltine, May to August depending on the altitude. |
| **Habitat:** | Steppes, arid hills, semi-deserts, dry clearings and glades in mountain woodlands. |

| 212*. *Hyponephele amardaea* (Lederer, 1869) | | |
|---|---|---|
| **English name:** | |
| **Georgian name:** | |
| **Russian name:** | |
| **Distr. Caucasus:** | Not reported. |
| **Distr. Georgia:** | |
| **Flight:** | Univoltine, July to August. |
| **Habitat:** | Dry, rocky slopes. |

| 213*. *Hyponephele comara* (Lederer, [1870]) | | |
|---|---|---|
| **English name:** | |
| **Georgian name:** | |
| **Russian name:** | Бархатница сонная |
| **Distr. Caucasus:** | Djavakheti-Armenian plateau. |
| **Distr. Georgia:** | |
| **Flight:** | Univoltine, June to August. |
| **Habitat:** | Rocky gorges. |

| 214. *Melanargia russiae* (Esper, 1783) | | |
|---|---|---|
| **English name:** | Esper's Marbled White |
| **Georgian name:** | |
| **Russian name:** | Пестроглазка русская |
| **Distr. Caucasus:** | Great and Lesser Caucasus. |
| **Distr. Georgia:** | Central and eastern regions, common in arid and subarid mountain forest range. |
| **Flight:** | Univoltine, mid-June to early August. |
| **Habitat:** | Steppes, grassy slopes, grassy woodland clearings. |

| 215. *Melanargia galathea* (Linnaeus, 1758) | | |
|---|---|---|
| **English name:** | Marbled White |
| **Georgian name:** | გალათეა |
| **Russian name:** | Пестроглазка галатея |
| **Distr. Caucasus:** | Great and Lesser Caucasus. |
| **Distr. Georgia:** | Widespread, common in mountains. |
| **Flight:** | Univoltine, June to August. |
| **Habitat:** | Dry meadows, forest edges and glades. |

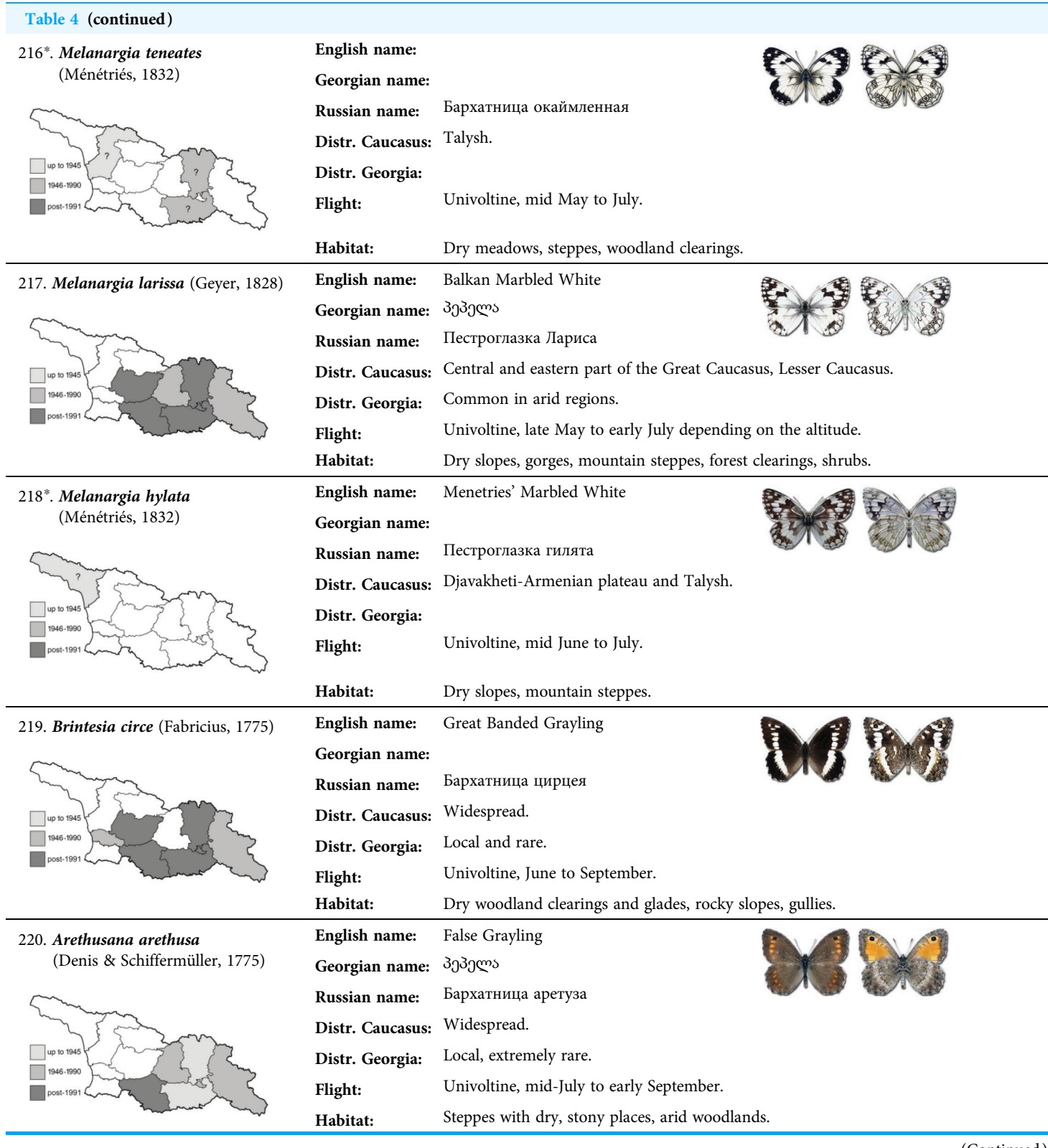

| 216*. *Melanargia teneates* (Ménétriés, 1832) | English name: | |
|---|---|---|
| | Georgian name: | |
| | Russian name: | Бархатница окаймленная |
| | Distr. Caucasus: | Talysh. |
| | Distr. Georgia: | |
| | Flight: | Univoltine, mid May to July. |
| | Habitat: | Dry meadows, steppes, woodland clearings. |

| 217. *Melanargia larissa* (Geyer, 1828) | English name: | Balkan Marbled White |
|---|---|---|
| | Georgian name: | პეპელა |
| | Russian name: | Пестроглазка Лариса |
| | Distr. Caucasus: | Central and eastern part of the Great Caucasus, Lesser Caucasus. |
| | Distr. Georgia: | Common in arid regions. |
| | Flight: | Univoltine, late May to early July depending on the altitude. |
| | Habitat: | Dry slopes, gorges, mountain steppes, forest clearings, shrubs. |

| 218*. *Melanargia hylata* (Ménétriés, 1832) | English name: | Menetries' Marbled White |
|---|---|---|
| | Georgian name: | |
| | Russian name: | Пестроглазка гилята |
| | Distr. Caucasus: | Djavakheti-Armenian plateau and Talysh. |
| | Distr. Georgia: | |
| | Flight: | Univoltine, mid June to July. |
| | Habitat: | Dry slopes, mountain steppes. |

| 219. *Brintesia circe* (Fabricius, 1775) | English name: | Great Banded Grayling |
|---|---|---|
| | Georgian name: | |
| | Russian name: | Бархатница цирцея |
| | Distr. Caucasus: | Widespread. |
| | Distr. Georgia: | Local and rare. |
| | Flight: | Univoltine, June to September. |
| | Habitat: | Dry woodland clearings and glades, rocky slopes, gullies. |

| 220. *Arethusana arethusa* (Denis & Schiffermüller, 1775) | English name: | False Grayling |
|---|---|---|
| | Georgian name: | პეპელა |
| | Russian name: | Бархатница аретуза |
| | Distr. Caucasus: | Widespread. |
| | Distr. Georgia: | Local, extremely rare. |
| | Flight: | Univoltine, mid-July to early September. |
| | Habitat: | Steppes with dry, stony places, arid woodlands. |

| 221. *Chazara briseis* (Linnaeus, 1764) | | |
|---|---|---|
| | **English name:** | The Hermit |
| | **Georgian name:** | |
| | **Russian name:** | Бархатница бризеида |
| | **Distr. Caucasus:** | Southern slopes of the Great Caucasus, Lesser Caucasus. |
| | **Distr. Georgia:** | Widespread and frequent. |
| | **Flight:** | Univoltine, June to September. |
| | **Habitat:** | Various grassy and stony habitats. |

| 222. *Chazara persephone* (Hübner, 1805) | | |
|---|---|---|
| | **English name:** | Dark Rockbrown |
| | **Georgian name:** | |
| | **Russian name:** | Бархатница Персефона |
| | **Distr. Caucasus:** | Widespread. |
| | **Distr. Georgia:** | Local and rare. |
| | **Flight:** | Univoltine, early June to September. |
| | **Habitat:** | Rocky places on steppes and mountain forests, sometimes semi-deserts. |

| 223. *Pseudochazara alpina* (Staudinger, 1878) | | |
|---|---|---|
| | **English name:** | |
| | **Georgian name:** | |
| | **Russian name:** | Бархатница альпийская |
| | **Distr. Caucasus:** | Great Caucasus, western part of the Lesser Caucasus. |
| | **Distr. Georgia:** | Lesser Caucasus, eastern part of the Great Caucasus, rare. |
| | **Flight:** | Univoltine, late June to September. |
| | **Habitat:** | Various stony and rocky places. |

| 224. *Pseudochazara mniszechii* (Herrich-Schäffer, 1851) | | |
|---|---|---|
| | **English name:** | Dark Grayling |
| | **Georgian name:** | |
| | **Russian name:** | Бархатница Мнишека |
| | **Distr. Caucasus:** | Western part of the Lesser Caucasus. |
| | **Distr. Georgia:** | Local and not frequent. |
| | **Flight:** | Univoltine, mid-June to early August. |
| | **Habitat:** | Rocky slopes with sparse vegetation, sometimes cliffs, screes. |

| 225. *Pseudochazara geyeri* (Herrich-Schäffer, 1846) | | |
|---|---|---|
| | **English name:** | Grey Asian Grayling |
| | **Georgian name:** | |
| | **Russian name:** | Бархатница Гейера |
| | **Distr. Caucasus:** | Western part of the Lesser Caucasus, northern part of Djavakheti-Armenian plateau. |
| | **Distr. Georgia:** | Kvemo Kartli and Samtskhe-Javakheti, extremely rare. |
| | **Flight:** | Univoltine, July to September. |
| | **Habitat:** | Various stony and rocky places, sometimes screes. |

| 226. *Pseudochazara beroe* (Herrich-Schäffer, 1844) | English name: | Freyer's Tawny Rockbrown |
|---|---|---|
| | Georgian name: | |
| | Russian name: | Бархатница бероя |
| | Distr. Caucasus: | Western part of the Lesser Caucasus, Djavakheti-Armenian plateau. |
| | Distr. Georgia: | Javakheti mountains. |
| | Flight: | Univoltine, June to September. |
| | Habitat: | Various stony and rocky places, sometimes screes. |

| 227. *Pseudochazara pelopea* (Klug, 1832) | English name: | Klug's Tawny Rockbrown |
|---|---|---|
| | Georgian name: | |
| | Russian name: | Бархатница Пелопея |
| | Distr. Caucasus: | Central and eastern part of the Great Caucasus, Lesser Caucasus. |
| | Distr. Georgia: | Widespread, fairly frequent. |
| | Flight: | Univoltine, mid-June to mid-August. |
| | Habitat: | Stony slopes and gorges, arid hills, dry, stony woodland glades and clearings. |

| 228. *Satyrus iranica* Schwingenschuss, 1939 | English name: | |
|---|---|---|
| | Georgian name: | |
| | Russian name: | |
| | Distr. Caucasus: | Central and eastern part of the Great Caucasus, Lesser Caucasus. |
| | Distr. Georgia: | Lesser Caucasus, rare. |
| | Flight: | Univoltine, June to August. |
| | Habitat: | Rocky steppes, woodlands clearings and glades, scrubs. |

| 229. *Satyrus amasina* Staudinger, 1861 | English name: | Amasian Satyr |
|---|---|---|
| | Georgian name: | |
| | Russian name: | Бархатница амазийская |
| | Distr. Caucasus: | Central and eastern part of the Great Caucasus, Lesser Caucasus. |
| | Distr. Georgia: | Lesser Caucasus, rare. |
| | Flight: | Univoltine, June to August. |
| | Habitat: | Rocky steppes, woodlands clearings and glades, scrubs. |

| 230. *Minois dryas* (Scopoli, 1763) | English name: | Dryad |
|---|---|---|
| | Georgian name: | პეპელა |
| | Russian name: | Бархатница дриада |
| | Distr. Caucasus: | Widespread. |
| | Distr. Georgia: | Widespread, common in the mountain forest range. |
| | Flight: | Univoltine, late June to August. |
| | Habitat: | Woodland clearings and glades, rocky steppes, scrub. |

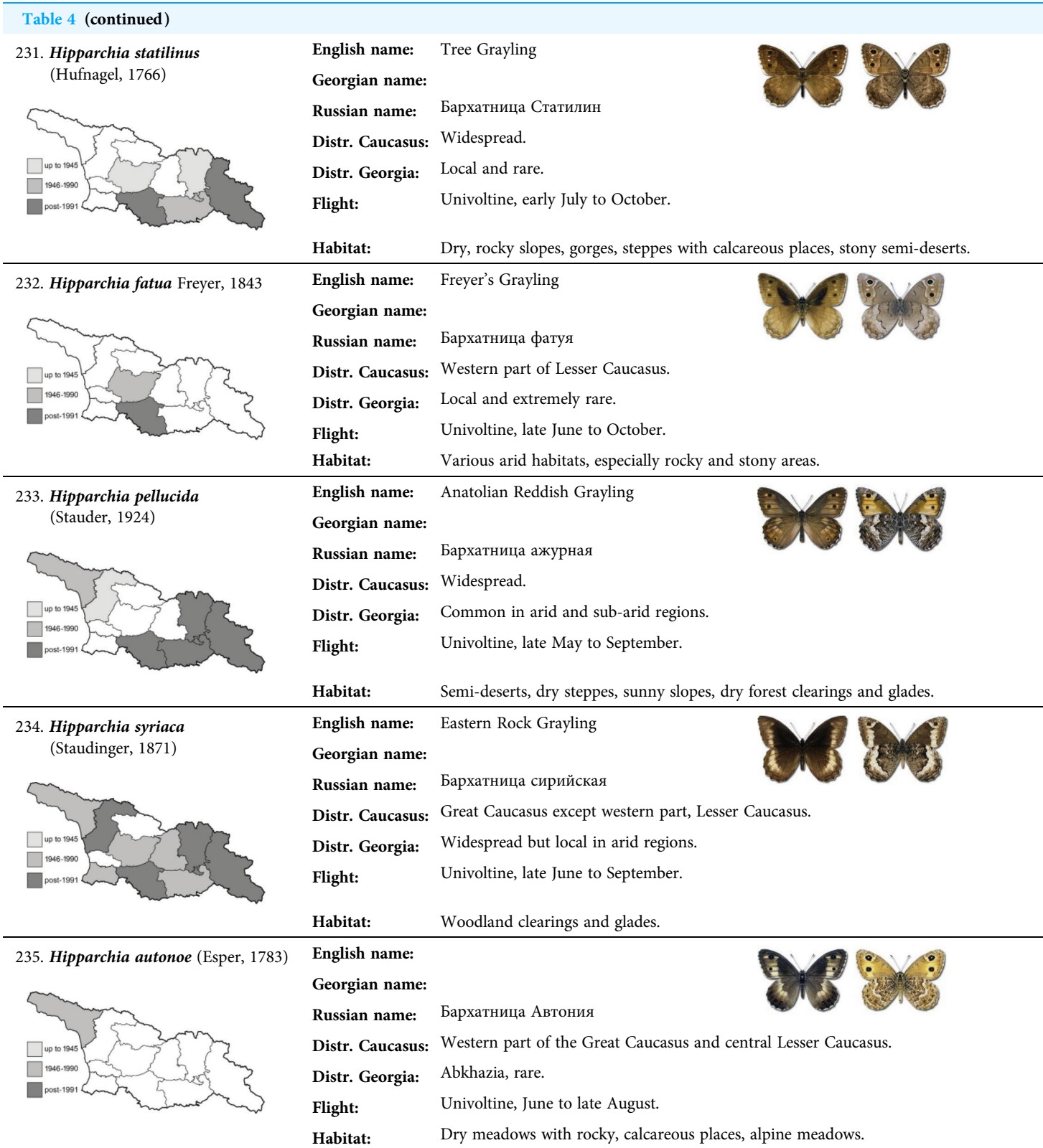

| | | |
|---|---|---|
| **231.** *Hipparchia statilinus* (Hufnagel, 1766) | **English name:** | Tree Grayling |
| | **Georgian name:** | |
| | **Russian name:** | Бархатница Статилин |
| | **Distr. Caucasus:** | Widespread. |
| | **Distr. Georgia:** | Local and rare. |
| | **Flight:** | Univoltine, early July to October. |
| | **Habitat:** | Dry, rocky slopes, gorges, steppes with calcareous places, stony semi-deserts. |
| **232.** *Hipparchia fatua* Freyer, 1843 | **English name:** | Freyer's Grayling |
| | **Georgian name:** | |
| | **Russian name:** | Бархатница фатуя |
| | **Distr. Caucasus:** | Western part of Lesser Caucasus. |
| | **Distr. Georgia:** | Local and extremely rare. |
| | **Flight:** | Univoltine, late June to October. |
| | **Habitat:** | Various arid habitats, especially rocky and stony areas. |
| **233.** *Hipparchia pellucida* (Stauder, 1924) | **English name:** | Anatolian Reddish Grayling |
| | **Georgian name:** | |
| | **Russian name:** | Бархатница ажурная |
| | **Distr. Caucasus:** | Widespread. |
| | **Distr. Georgia:** | Common in arid and sub-arid regions. |
| | **Flight:** | Univoltine, late May to September. |
| | **Habitat:** | Semi-deserts, dry steppes, sunny slopes, dry forest clearings and glades. |
| **234.** *Hipparchia syriaca* (Staudinger, 1871) | **English name:** | Eastern Rock Grayling |
| | **Georgian name:** | |
| | **Russian name:** | Бархатница сирийская |
| | **Distr. Caucasus:** | Great Caucasus except western part, Lesser Caucasus. |
| | **Distr. Georgia:** | Widespread but local in arid regions. |
| | **Flight:** | Univoltine, late June to September. |
| | **Habitat:** | Woodland clearings and glades. |
| **235.** *Hipparchia autonoe* (Esper, 1783) | **English name:** | |
| | **Georgian name:** | |
| | **Russian name:** | Бархатница Автония |
| | **Distr. Caucasus:** | Western part of the Great Caucasus and central Lesser Caucasus. |
| | **Distr. Georgia:** | Abkhazia, rare. |
| | **Flight:** | Univoltine, June to late August. |
| | **Habitat:** | Dry meadows with rocky, calcareous places, alpine meadows. |

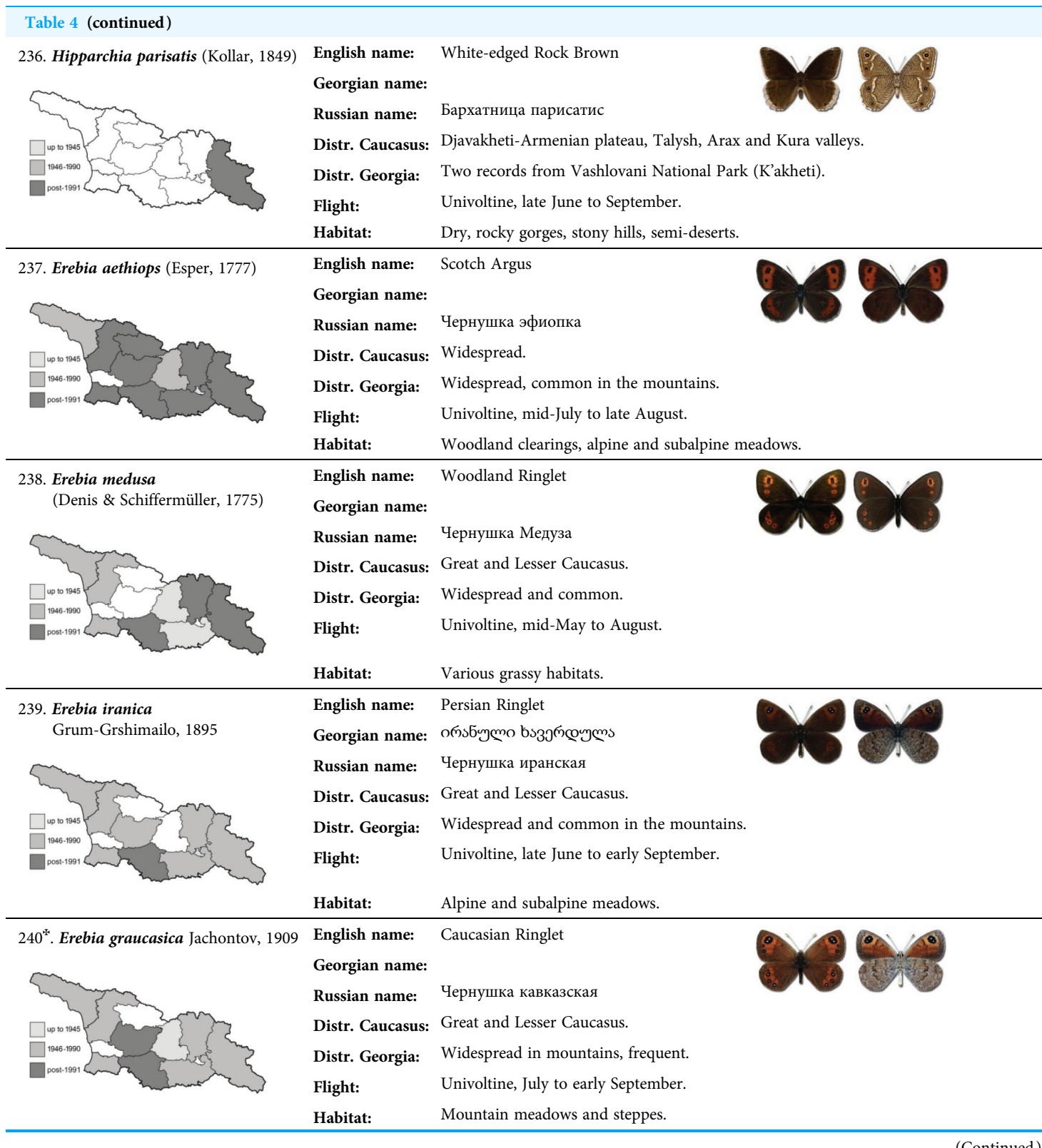

| 236. *Hipparchia parisatis* (Kollar, 1849) | | |
|---|---|---|
| | English name: | White-edged Rock Brown |
| | Georgian name: | |
| | Russian name: | Бархатница парисатис |
| | Distr. Caucasus: | Djavakheti-Armenian plateau, Talysh, Arax and Kura valleys. |
| | Distr. Georgia: | Two records from Vashlovani National Park (K'akheti). |
| | Flight: | Univoltine, late June to September. |
| | Habitat: | Dry, rocky gorges, stony hills, semi-deserts. |
| 237. *Erebia aethiops* (Esper, 1777) | | |
| | English name: | Scotch Argus |
| | Georgian name: | |
| | Russian name: | Чернушка эфиопка |
| | Distr. Caucasus: | Widespread. |
| | Distr. Georgia: | Widespread, common in the mountains. |
| | Flight: | Univoltine, mid-July to late August. |
| | Habitat: | Woodland clearings, alpine and subalpine meadows. |
| 238. *Erebia medusa* (Denis & Schiffermüller, 1775) | | |
| | English name: | Woodland Ringlet |
| | Georgian name: | |
| | Russian name: | Чернушка Медуза |
| | Distr. Caucasus: | Great and Lesser Caucasus. |
| | Distr. Georgia: | Widespread and common. |
| | Flight: | Univoltine, mid-May to August. |
| | Habitat: | Various grassy habitats. |
| 239. *Erebia iranica* Grum-Grshimailo, 1895 | | |
| | English name: | Persian Ringlet |
| | Georgian name: | ირანული ხავერდულა |
| | Russian name: | Чернушка иранская |
| | Distr. Caucasus: | Great and Lesser Caucasus. |
| | Distr. Georgia: | Widespread and common in the mountains. |
| | Flight: | Univoltine, late June to early September. |
| | Habitat: | Alpine and subalpine meadows. |
| 240*. *Erebia graucasica* Jachontov, 1909 | | |
| | English name: | Caucasian Ringlet |
| | Georgian name: | |
| | Russian name: | Чернушка кавказская |
| | Distr. Caucasus: | Great and Lesser Caucasus. |
| | Distr. Georgia: | Widespread in mountains, frequent. |
| | Flight: | Univoltine, July to early September. |
| | Habitat: | Mountain meadows and steppes. |

(Continued)

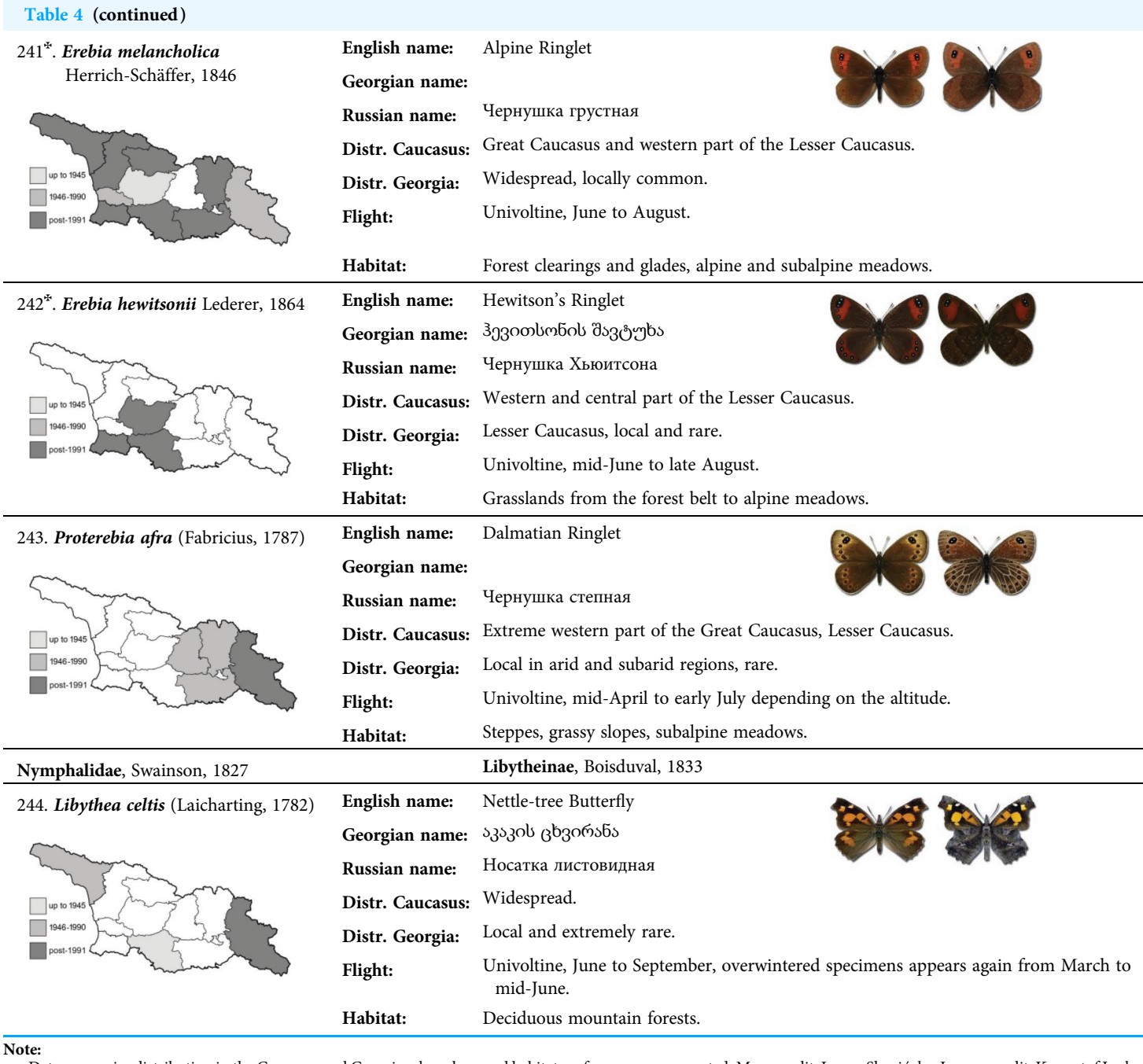

| | | |
|---|---|---|
| **241*.** *Erebia melancholica* Herrich-Schäffer, 1846 | **English name:** | Alpine Ringlet |
| | **Georgian name:** | |
| | **Russian name:** | Чернушка грустная |
| | **Distr. Caucasus:** | Great Caucasus and western part of the Lesser Caucasus. |
| | **Distr. Georgia:** | Widespread, locally common. |
| | **Flight:** | Univoltine, June to August. |
| | **Habitat:** | Forest clearings and glades, alpine and subalpine meadows. |
| **242*.** *Erebia hewitsonii* Lederer, 1864 | **English name:** | Hewitson's Ringlet |
| | **Georgian name:** | ჰევითსონის შავტუხა |
| | **Russian name:** | Чернушка Хьюитсона |
| | **Distr. Caucasus:** | Western and central part of the Lesser Caucasus. |
| | **Distr. Georgia:** | Lesser Caucasus, local and rare. |
| | **Flight:** | Univoltine, mid-June to late August. |
| | **Habitat:** | Grasslands from the forest belt to alpine meadows. |
| **243.** *Proterebia afra* (Fabricius, 1787) | **English name:** | Dalmatian Ringlet |
| | **Georgian name:** | |
| | **Russian name:** | Чернушка степная |
| | **Distr. Caucasus:** | Extreme western part of the Great Caucasus, Lesser Caucasus. |
| | **Distr. Georgia:** | Local in arid and subarid regions, rare. |
| | **Flight:** | Univoltine, mid-April to early July depending on the altitude. |
| | **Habitat:** | Steppes, grassy slopes, subalpine meadows. |
| **Nymphalidae**, Swainson, 1827 | **Libytheinae**, Boisduval, 1833 | |
| **244.** *Libythea celtis* (Laicharting, 1782) | **English name:** | Nettle-tree Butterfly |
| | **Georgian name:** | აკაკის ცხვირანა |
| | **Russian name:** | Носатка листовидная |
| | **Distr. Caucasus:** | Widespread. |
| | **Distr. Georgia:** | Local and extremely rare. |
| | **Flight:** | Univoltine, June to September, overwintered specimens appears again from March to mid-June. |
| | **Habitat:** | Deciduous mountain forests. |

**Note:**
Data on species distribution in the Caucasus and Georgia, phenology and habitat preferences are presented. Maps credit: Iwona Słowińska. Images credit: Krzysztof Jonko.

The arrangement of sampling points on the map (Fig. 3) indicates that the Lesser Caucasus is relatively well explored in comparison to the Great Caucasus. The Great Caucasus, although it borders Lesser Caucasus, remains poorly investigated with regard to butterflies. More species may be expected in this area, but it will depend on the intensity of the future field research.

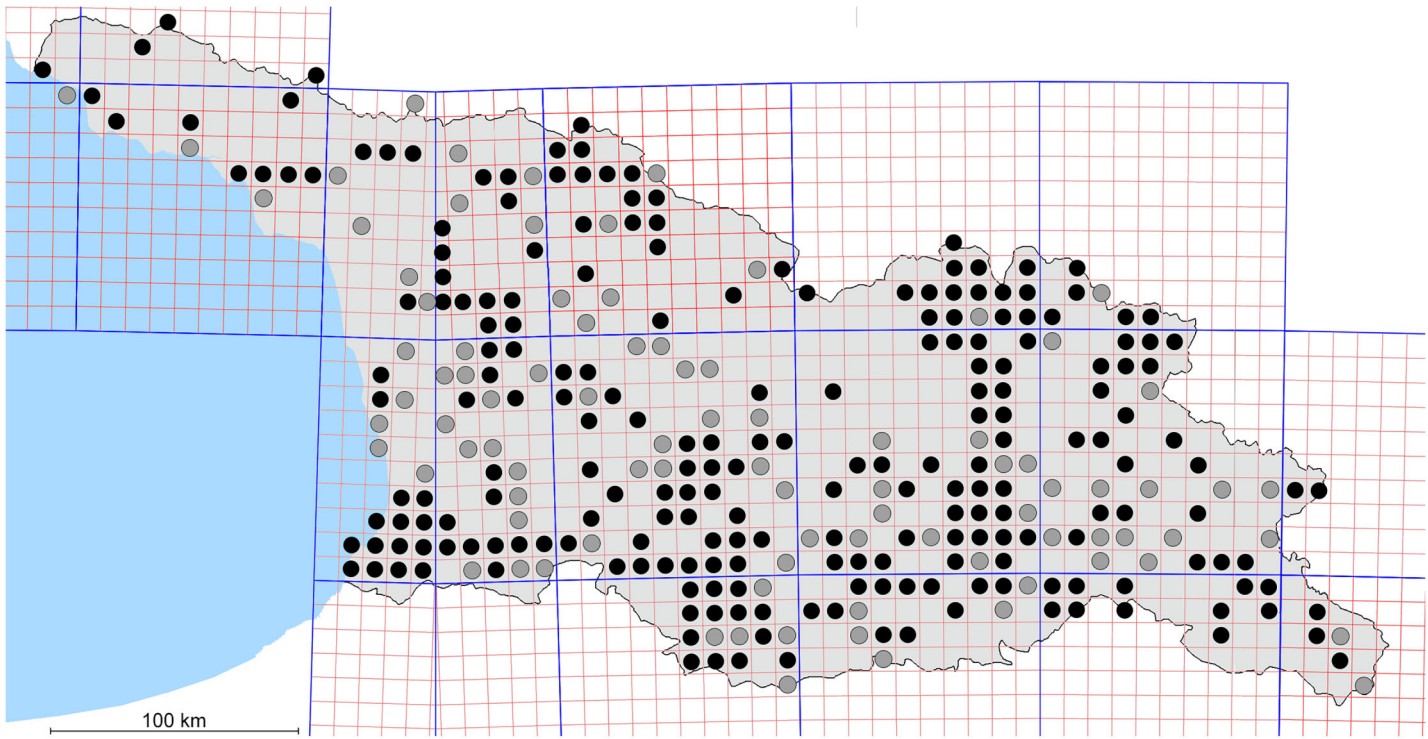

**Figure 3 Map of studied localities in UTM projection (grid 10 × 10 km).** (○) One or two record(s). (•) Three or more records. Map credit: Krzysztof Jonko.

| Region | Code[1] | Records | Localities | Species |
|---|---|---|---|---|
| Abkhazia | AB | 461 | 30 | 131 |
| Ajaria | AJ | 794 | 56 | 90 |
| Guria | GU | 44 | 12 | 30 |
| Imereti | IM | 316 | 37 | 106 |
| K'akheti | KA | 924 | 85 | 159 |
| Kvemo Kartli | KK | 626 | 70 | 151 |
| Mtskheta-Mtianeti | MM | 1,245 | 96 | 145 |
| Rach'a-Lechkhumi-Kvemo Svaneti | RL | 105 | 25 | 43 |
| Samegrelo-Zemo Svaneti | SZ | 808 | 74 | 98 |
| Samtskhe-Javakheti | SJ | 1,713 | 81 | 191 |
| Shida Kartli | SK | 100 | 15 | 70 |
| Tbilisi | TB | 559 | 11 | 119 |
| Total in Georgia | | 7,695 | 592 | 244 |

**Table 5 Comparison of number of records, localities and butterfly species in Georgia regions.**

**Note:**
[1] ISO 3166-2 region code.

## DISCUSSION

The territory of Georgia is located between two mountain ranges: the Great Caucasus to the north and the Lesser Caucasus to the south. The inter-montane plains of Central

Georgia are located between them. It is necessary to remember that a major part of this country is occupied by mountains with 55% of its surface area located at altitudes higher than 1,000 m. Such topographic heterogeneity and habitat diversity favours high species diversity and endemism visible in different taxonomical groups in the entire area of the Caucasus (*e.g.*, *Eliava et al., 2007*; *Sendra & Reboleira, 2012*; *Zazanashvili et al., 2012*; *Mumladze, Cameron & Pokryszko, 2014*; *Konstantinov & Simov, 2018*; *Grego et al., 2020*; *Kokhia & Golovatch, 2020*).

After including information given by *Tshikolovets & Nekrutenko (2012)*, we updated the list of Georgian butterflies by adding 39 species which increases the total number of species to 244 (belonging to five families). Quite a large number of species new to Georgia compared to the publication mentioned above is a result of three factors: changes in systematics, new discoveries and description of species new to science. It is important to note that, to date, no species have been identified as endemic to Georgia. However, what distinguishes the Georgian butterfly fauna is the presence of rare and extremely rare species (comprising almost 25% of the butterfly fauna), as well as the species endemic to the Caucasus. Our data shows that butterfly fauna is very diverse, despite the fact that Georgia is a relatively small country. It is worth mentioning that Georgia is located at a biogeographical crossroads where at least three biogeographical provinces converge. For this reason, butterfly fauna of this country expresses the mixed character of its biogeographic connections with European, Asian and Middle Eastern elements, as well as Caucasian endemics. Notably, the presence of 16 endemic species to the Caucasus occurring in Georgia indicates a relatively high level of endemism in this country. It is important to note that these species play a crucial role in maintaining biodiversity, not only in Georgia but throughout the entire Caucasus region. Due to their limited geographic range, endemic species are often more vulnerable to extinction. This makes their conservation a priority in light of habitat loss and climate change.

Many records of over a dozen species in Georgia are probably erroneous, although these species can be expected there as they are noted in the surrounding countries. We assumed that the number of species reported in Georgia and its neighbouring countries was comparable.

Consequently, we compared our list of the butterfly fauna with existing checklists (or papers) from neighbouring countries of Armenia and Azerbaijan (belonging to the Transcaucasia) as well as Russia and Turkey (*Tshikolovets & Nekrutenko, 2012*; *Koçak & Kemal, 2018*; *Langourov, 2019*; *Sinev, 2019*; *Snegovaya & Petrov, 2019*). In the case of Russia, we focused only the regions bordering Georgia: Krasnodar and Stavropol Krai, the Republic of Adygea, Kabardino-Balkaria, Karachay-Cherkess, North Ossetia-Alania, Ingushetia, referred to as the West Caucasian region and Chechen Republic and the Republic of Dagestan referred to as the East Caucasian region. In the case of Turkey, we concentrated only on two provinces—Artvin and Ardahan.

We found that the number of species recorded for Georgia is comparable to the butterfly fauna of other republics of the Transcaucasia, while it is higher in relation to neighbouring provinces of Turkey and Russia (Table 6).

**Table 6 Comparison of number of butterfly species of Georgia with surrounding countries.**

| Family | Georgia | Azerbaijan | Armenia | Turkey[1] | Russia[2] |
|---|---|---|---|---|---|
| Hesperiidae | 29 | 35 | 31 | 24 | 28 |
| Lycaenidae | 83 | 101 | 87 | 82 | 74 |
| Nymphalidae | 96 | 104 | 88 | 83 | 93 |
| Papilionidae | 8 | 10 | 5 | 6 | 7 |
| Pieridae | 28 | 28 | 27 | 24 | 23 |
| Riodinidae | 0 | 1 | 0 | 0 | 1 |
| Total | 244 | 279 | 238 | 219 | 226 |

Notes:
[1] Only Artvin & Ardahan provinces of Turkey.
[2] Only East & West Caucasus regions of Russia.

The high number of species recorded for Azerbaijan should be treated with caution; we are convinced that several species require independent confirmation, like in the case of Georgia.

Even though Georgia has a long history of butterfly research, the fauna is still poorly researched. It is worth mentioning that, since the collapse of the Soviet Union, research on Georgian butterflies and moths has primarily been conducted by *Didmanidze (2004)*, and it has been 20 years since the last inventory was published.

The updated checklist presented in this article and data containing thumbnails, vernacular names and species regional distribution maps may serve as a baseline for further exploration of Georgian butterflies. We hope that our work will be useful for scientists and will also promote interest in butterflies among communities, local entomologists and citizen scientists. They may use this checklist to document the presence or absence of butterfly species in specific regions of Georgia. Citizen participation in data collection may be one of the steps towards conservation of butterflies and/or their habitats in that country.

## CONCLUSIONS

Our study supplements the knowledge of butterflies from Georgia. We added 39 new species found or described in the last decade. Among 244 recorded species almost 25% of Georgian species are categorised as rare and extremely rare, and these species may be at the highest risk of potential extinction. The number of recorded species in Georgia is comparable to the butterfly fauna of other republics of the Transcaucasia; however, it is higher than neighbouring provinces of Turkey and Russia. The occurrence of over a dozen species is questioned by us due to potential misidentification or ambiguous data. That is why we consider it essential to verify determination of several species listed by Didmanidze or confirm their occurrence in Georgia with independent studies. The data presented in this article may serve as a baseline for further exploration of the diversity of butterfly fauna of Georgia. This is crucial to better management of conservation in the face of loss habitats and changes resulting from climate change.

## ACKNOWLEDGEMENTS

We would like to thank all the V4 & Eastern Partnership team members (mentioned in alphabetical order): Ani Bikashvili, Sofia Gabelashvili, Ľuboš Hrivniak, Bella Japoshvili, Ilhama Kerimova, Peter Manko, Levan Mumladze, David Murányi, Josef Oboňa, Michal Rendoš, Nataly Snegovaya and Matej Žiak for their kindness and help during the field work. We would like to thank Mateusz Płóciennik for his valuable comments on the article. We would like to express our gratitude to Jarosław Buszko (Nicolaus Copernicus University in Toruń) for his invaluable and constructive feedback on the manuscript. We also thank two anonymous reviewers for their comments and suggestions on the first version of the manuscript.

### Funding

This work was supported by the International Visegrád Fund (Project No. 21810533). The funding for the Article Publication Charge (APC) has been provided by the University of Lodz Statutory Funds. The funders had no role in study design, data collection and analysis, decision to publish, or preparation of the manuscript.

### Grant Disclosures

The following grant information was disclosed by the authors:
International Visegrád Fund: 21810533.
University of Lodz Statutory Funds.

### Competing Interests

The authors declare that they have no competing interests.

### Author Contributions

- Iwona Słowińska conceived and designed the experiments, performed the experiments, analyzed the data, prepared figures and/or tables, authored or reviewed drafts of the article, and approved the final draft.
- Krzysztof Jonko conceived and designed the experiments, performed the experiments, analyzed the data, prepared figures and/or tables, authored or reviewed drafts of the article, and approved the final draft.

### Data Availability

The raw data with location names in English and Georgian and GPS coordinates are available in the Supplemental File.

### Supplemental Information

Supplemental information for this article can be found online at http://dx.doi.org/10.7717/peerj.18720#supplemental-information.

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
