# Peer review of "Butterflies (Lepidoptera: Papilionoidea) of Georgia (Caucasus): annotated review of regional butterfly fauna with vernacular names index, notes on distribution and phenology"

_PeerJ, doi:10.7717/peerj.18720_

## Round 0.1 · original submission · Major Revisions

Dear Dr. Słowińska,

After this review round, the reviewers believe that your manuscript may be accepted for publication in PeerJ after a thorough review is performed. Please take special care with the issues reported by reviewer #2.

Sincerely,
Daniel Silva

Reviewer 1 ·

Basic reporting

it seems the introduction part is too long and includes of unnecessary lines.
better to be more brief.
the Method part is mixed with the Conclusion section. all comparison between fauna of neighboring area should be gathered in Conclusion section.

in Result section , all materials and Methods would be written.

Experimental design

Nice and excellent

Validity of the findings

1- Melitaea casta is an endemic species of Zagros mountains in Iran.
it is surely a mistake for Georgia.

2- It is proposed the Authors use the latest Checklist of Lepidoptera of Iran to stablish correct and valid names for taxa.
https://www.researchgate.net/publication/369299416_Catalogue_of_the_Lepidoptera_of_Iran

3- It is also proposed that the arrangement of taxa would be on the basis of subgenera taxonomic names.
e.g.in Polyommatus Genus subgenus Agrodiaetus is between other Polyommatus species.

4- the author can also use the books of other countries in the same ecoregion like Iran. ( in north of Iran there is the most southern part of Caucasus ecoregion with the same species.

Additional comments

overall it is interesting study and worthy to publish after minor correction.

Reviewer 2 ·

Basic reporting

This paper basically presents an updated distributional checklist of Georgian butterflies. Such an initiative is definitely welcome. The main problem now is that the original contribution made by this paper is not sufficiently clear, the authors must make it explicit. See Additional comments for some specific points.

As the authors appear not to have much experience with publishing similar papers in international journals, I strongly suggest that they ask some butterfly experts experienced in publishing to comment on their text before resubmission. I am sorry but currently the structure of the text and the use of the language do not look very professional, and should be improved. Also, while I am not a native user of English myself, I nevertheless dare to say that the text would benefit from linguistic proofreading.

Experimental design

Not applicable.

Validity of the findings

The authors should make their original contribution explicit after which the validity of the novel conclusions could be fully evaluated.

Additional comments

While I sincerely appreciate the work done, I must say that I am not particularly impressed by this manuscript. Above all else, it is not sufficiently clear what is the novelty of this study. The original contribution made by the authors must be made explicit. I strongly suggest that instead of Didmanidze (2004) you take the thorough monographic work by Tshikolovets and Nekrutenko (2012) (TN, hereafter) as your main reference point (even if that work does not include an explicit list of Georgian butterflies, relevant information can be easily retrieved from distribution maps). I strongly suggest your Table 3 should explicitly compare your conclusions to the conclusions by TN and then you should explain, case by case, why does your opinion differ from that of TN. In fact, in numerous cases where you state that Didmanidze’s (2004) records are doubtful, such doubts have already been expressed by TN (which you must not hide), so that you do not add anything new. After such a change, the novelty of your contribution will be much easier to evaluate. Also, I strongly recommend that before resubmitting your work you ask one or more butterfly experts experienced in international publishing to comment on your text. Also, I think that some Georgian biologists should see it as well.

Specific:
Line 1. Do not say “Lepidoptera (Papilionoidea) of Georgia”, this may give the wrong impression that you deal with all lepidopterans, please write “Butterflies (Papilionoidea) of Georgia” !
Line 23. Unfortunately, this does not lead to minimisation of biodiversity losses, please reformulate.
Line 24. Unfortunately?! Is it unfortunate that science has progressed?
Line 32. I am sorry but the fact that you have visited Georgia twice for a few days is probably not important enough go be mentioned in the Abstract.
Lines 37, 340. This 25% does not say anything before you tell how exactly do you define and measure being “rare”.
Line 43. “In comparison to the other republics of Transcaucasia (Armenia, Azerbaijan), we noted a similar number of species” – this is trivial and not worth of mentioning.
Line 65. Administrative division of Georgia is irrelevant here. Instead, I miss a much more comprehensive physical geographical and biogeographical description of the country. Which habitats are present?
In the Introduction, there is far too much about insect declines and the fact that faunistic inventories are important for nature conservation. All your potential readers know all this, please reduce this kind of text to a few sentences.
Tell us more about the current situation in Georgian lepidopterology. Can we say that there are neither professional nor amateur lepidopterists in Georgia currently? Any research carried on by foreigners?
Line 139. The merits of the Georgian alphabet are irrelevant here.
Line 168. TN is based on a huge number of (frequently, old and obscure) literature sources. Did you really review all this? Probably not, so please tell us clearly and honestly which literature did you actually work with.
Line 179. Tell us how many records in GBIF were there for Georgia.
Line 195. Once again, I feel that your short visits are overemphasized here.
Line 231. Not Lepidoptera, just butterflies!
Line 278. What is the source of Georgian and Russian vernacular names, how widely are they used? Should one propose Georgian names for species which do not have ones?
Line 282. To the best of my understanding, Georgian language was no way replaced by Russian in Soviet times. I think that any statements like that would be very painful to read for any Georgian so that you must be careful. It is true of course that Russian was and still is widely known in Georgia but this is a different thing, and Abkhazia is a very special case.
Line 303. I strongly feel that this analysis of altitudinal zonation does not belong in this paper. This should be a separate study, with the results being compared to similar studies elsewhere. Moreover, I also strongly doubt that you results carry any novelty.
Line 350. I find this comparison with neighbouring areas trivial. Of course, we can expect that butterfly diversity of Georgia is comparable to that of neighbouring areas, how could it be otherwise?
Lines 377-406. This is more Results than Discussion, and does not fit in this paper in my mind (see above).
Table 2. What notations have you used if both old and recent records are available? Please clarify.

Table 3. I did not check everything carefully but you should do so, I noticed the following problems, and there may be more:
** item 8. The understanding that the traditional L. sinapis is a complex of sibling species is older than Dinca et al (2011). Reference for both species occurring in Georgia?
** item 18. There is lots of recent work on the A. glandon complex, the reference from 1970 is certainly outdated.
** item 26. “Trialetsky and Mesketsky” are likely Russian modifications of these toponyms, to be avoided, perhaps.
** item 30. Tzarskije Kolodtzy (a 19th century name) is Dedoplistskaro nowadays.
** item 38. “subspecies was elevated to species level” – reference missing.

---

## Round 0.2 · Minor Revisions

Dear Dr. Słowińska,

After this new review round, the review suggested several minor changes to your manuscript. In the next version of your manuscript, if you perform the proposed changes and convince the reviewer about the final version of your manuscript, I am certain your manuscript will be accepted for publication in PeerJ.

Sincerely,
Daniel SIlva

Reviewer 2 ·

Basic reporting

Check English in Table 3.

Experimental design

N/A

Validity of the findings

Much improved.

Additional comments

The authors have addressed most of my concerns, the following points remain:

+1. Perhaps you should specify in the title “…. Georgia (Caucasus)” as there are also other Georgias in the world.
+2. You consider many of the records of Didmanidze doubtful. In this context, it is important to know whether respective specimens are still present in some collections and can be checked. Also, more generally, please tell in the text whether there are butterfly collections in Georgia one can work with.
+3. I keep insisting that you add a sentence describing the current situation with lepidopterological knowledge in Georgia, perhaps around line 99. It is relevant in the present context whether there is someone working on butterflies in Georgia or not.
+4. Line 71. I would not say that Georgia covers a SMALL part of the Caucasus region.
+5. Thank you for defining “rare” but “extremely rare” still remains to be defined, you use it below.
+6. Where is the data base, is it available in some way? Please specify in the text.
+7. I think that the species endemic to Georgia deserve to be listed, and endemism discussed somehow.
+8. “List of necessary comments” sounds strange.
Line 238. I still insist that you mention the sources of Georgian and Russian names in the text. In response to your response, I feel that I disagree with “In our opinion, artificially assigning vernacular names to many or even all species is not appropriate”. I am sure that in all languages in which (all) butterflies (of the region) have official names, most of these have been “artificially assigned”, traditional vernacular names are available for just a handful of species, perhaps in any language. Nevertheless, this is just a comment; of course, I do not mean that you should create new Georgian names here.
Line 325. May Michal Rendoš be Michal Rindoš if we mean the same person?
Table 2. No, you have not stated that if both ‘old’ and ‘new’ records are available, you use the symbol for ‘new’ records. Please do so.
Table 3. It is usual to abbreviate genus names if several species of the same genus are listed. You do it sometimes (e.g. Pontia) but not always (eg Colias chrysotheme), please unify.
Was Table 3 also subjected to linguistic proofreading? The language in the table looks little more careless than the rest of the text.

---

## Round 0.3 · Minor Revisions

Dear Dr. Słowińska,

The review raised minor but significant issues that require your attention before the acceptance of the manuscript. Please take a thorough look at what has been criticized and address the issues appropriately.

Sincerely,

Daniel Silva

Reviewer 2 ·

Basic reporting

OK

Experimental design

N/A

Validity of the findings

See below.

Additional comments

I appreciate the work the authors have done, most of my concerns have been adequately addressed.

HOWEVER, I noticed a major problem which escaped my attention previously (my fault!). I now understood how endemic species have been indicated in Table 2 but this cannot be correct! Just starting from the beginning of the list: Pyrgus melotis, P. jupei and Parnassius nordmanni are no way endemic to Georgia! See for example distribution maps in Tshikolovets and Nekrutenko 2012. Or do you mean "endemic to the Caucasus region" and not "endemic to Georgia"? If so, this must be stated clearly everywhere you talk about endemism. However, also if you mean "endemic to Caucasus", there must be mistakes as for example Lycaena candens also occurs in Europe. Please check everything carefully and tell in the text how did you decide if a species should be considered endemic for Georgia or not.

Also, please clearly tell in the text if and how your faunistic data base is accessible. If not otherwise, write the standard "The data base of faunistic records is available from the authors upon request", if this is the case. If you plan to publish the data base later on, please tell how.

---

## Round 0.4 · Minor Revisions

Dear Dr. Słowińska,

Your manuscript has received a minor decision and will be accepted for publication after you resubmit the version with the corrected issues.

Sincerely,

Daniel Silva

Reviewer 2 ·

Basic reporting

OK

Experimental design

N/A

Validity of the findings

OK

Additional comments

Thanks for the corrections but I would suggest that you still make the following changes related to the endemism business:

Line 284: “its endemic species” – I think that everybody will understand it so that you are talking about species endemic to Georgia but you are not! Please reformulate.

Line 289: perhaps “endemic to Caucasus”, not “endemic from Caucasus”

Table 1. Also here, you must say that you talk about species endemic to the Caucasus region, otherwise everybody will think that you talk about species endemic to Georgia.

Are there any species endemic to Georgia? Probably not. Please say this somewhere in the text.

---

## Round 0.5 · accepted · Accept

Dear Dr. Słowińska,

I am pleased to accept your manuscript for publication in PeerJ. Congratulations on your hard work.

Sincerely,
Daniel Silva